# Research

behaviour/ecology/evolution

burying beetle, carry-over effects, communal breeding, parental investment, previous breeding experience, sexual conflict

**Author for correspondence:**
Long Ma
e-mail: long.ma@rug.nl

# Sex-specific influence of communal breeding experience on parenting performance and fitness in a burying beetle

Long Ma[1], Maaike A. Versteegh[1], Martijn Hammers[1,2] and Jan Komdeur[1]

[1]Groningen Institute for Evolutionary Life Sciences (GELIFES), University of Groningen, 9712 CP Groningen, The Netherlands
[2]Aeres University of Applied Sciences, Arboretum West 98, 1325 WB Almere, The Netherlands

LM, 0000-0002-6911-0684; MH, 0000-0002-6638-820X; JK, 0000-0002-9241-0124

Communal breeding, wherein multiple conspecifics live and reproduce together, may generate short-term benefits in terms of defence and reproduction. However, its carry-over effects remain unclear. We experimentally tested the effects of communal breeding on parental care and reproduction in burying beetles (*Nicrophorus vespilloides*), which use carcasses as breeding resources and provide parental care to offspring. We subjected individuals to communal or non-communal breeding (i.e. pair breeding) during their first breeding event and to non-communal breeding during their second breeding event. We measured the parental care of individuals and of groups and the reproductive success of groups during both breeding events. In communal groups, large individuals became dominant and largely monopolized the carcass, whereas small individuals (i.e. subordinates) had restricted access to the carcass. At the first breeding event, large males in communal groups spent more time providing care than large males in non-communal groups, whereas such an effect was not observed for large females and small individuals. Reproductive successes were similar in communal and non-communal groups, indicating no short-term benefits of communal breeding in terms of reproduction. Compared with males from non-communal groups, males originating from communal groups produced a larger size of brood during their second breeding event, whereas such an effect was not observed for females. Our results demonstrate the sex-specific effects of communal breeding experience on parenting performance and fitness.

# 1. Introduction

Social groups of conspecifics occur in most animal species, for example individuals form temporary associations due to aggregated resources or forage as groups (i.e. group hunting) [1,2]. More advanced forms of social groups may occur among conspecifics, for example in cooperatively and communally breeding systems (hereafter group breeding), where multiple individuals live and reproduce together to rear a single brood or litter [3–5]. In such breeding groups, members may gain kin-selected benefits (i.e. the indirect fitness benefits individuals gain through helping their relatives reproduce) and various aspects of direct fitness benefits (e.g. non-kin reciprocity and mutualism) from group living [1,6–8], although they may or may not benefit in terms of direct reproductive success [9–12]. A large number of studies have focused on the short-term fitness benefits of kin-selected and direct benefits of group breeding for each individual involved, such as the joint defence of territories and resources against intruders [8,13–15], reduced workloads during parental care [16–20], higher reproductive success [2,21,22] and higher survival and longevity [23]. Furthermore, events and processes occurring in group breeding may affect individual performance in subsequent periods (i.e. carry-over effects), and these may translate into delayed or long-term fitness benefits for the individuals involved [18,24,25]. Nevertheless, the carry-over effects of group breeding on behaviour and fitness are not well-known in animal species [26–28]. Studying the short- and long-term fitness benefits for individuals that live and reproduce in such groups contributes towards a nuanced understanding of the evolutionary mechanisms that shape helping behaviour and cooperation in animals [10,12,23,29].

Carry-over effects refer to a potential consequence wherein an individual's previous conditions (e.g. physical state and previous experience) could exert an impact on its subsequent performance, such as parenting behaviour and reproductive success. Such effects can be due to differences in access to resources or variation in the resource allocation at one stage in life to another stage, e.g. within and across breeding seasons or between reproductive events [30–33]. Under these circumstances, individuals may trade off their efforts in self-maintenance and reproduction between life cycle stages due to variation in physiological condition (e.g. body weight and immunocompetence) [27,34,35]. Generally, asymmetries in physical state are driven by ecological conditions, most notably habitat quality [36–38]. For example, in some migratory birds, bad weather conditions during migration can lead to a reduction in body condition. Subsequently, this may affect later events of the life cycle, such as the arrival time at the breeding site or reproductive timing and success [39–41]. In some breeding groups, dominance hierarchy may generate individual differences in body condition [42,43]. For example, high-ranked, dominant individuals often monopolize high-quality resources or the access to resources in social groups. This monopolization of resources by dominants may result in lower-ranked individuals (i.e. subordinates) using lower-quality resources or having restricted access to resources. This differential access to resources may therefore result in carry-over effects for dominants and subordinates, e.g. subordinates suffering low survival rates and low reproductive success in the future [42–44]. Moreover, previous breeding experience and changes in physiological states may have carry-over effects on fitness and reproductive performance for individuals [45,46]. For example, in the northern goshawk (*Accipiter gentilis*), experienced individuals with previous breeding provided more parental care towards the current brood compared with inexperienced individuals [45].

In group breeding systems, the presence of subordinates can reduce the costs of reproductive efforts for dominants, which may then have a positive effect on the dominants' future survival and reproduction [23,24,27,47]. The impact of the allocation of parental effort to the current reproduction on subsequent reproductive performance and survival can also be viewed as a type of carry-over effect [32,44,48]. Compared with non-group breeding species, group breeding species are particularly interesting to study trade-offs between current and future reproduction, because parenting behaviour and allocation of resources by breeding individuals can be adjusted by their relative position in a dominance hierarchy [19,24,47,49]. The dominance status of individuals is thought to determine not only their reproductive success in current reproductive attempts, i.e. high parentage in a shared brood [4,50,51], but also their future mating opportunities and reproduction [8,52–54]. As such, breeding in groups may impact reproductive success not only through short-term effects but also through carry-over effects on future reproductive allocation [31,48,52,55]. In the context of group breeding, females and males may be able to differentially respond to the uncertainty of parentage, as well as resource availability, by adjusting their investment to current reproductive attempts. This can lead to a sex difference in the allocation of parental resources for future reproduction and fitness [16,49,56,57]. As

such, it is expected that a sex-specific difference in parental allocation is more pronounced during group breeding, compared with pair breeding [16,56]. Hitherto, the carry-over effects of group breeding on parenting behaviour and future fitness on reproduction of group members has been little studied [3,27,48,58].

Here, we experimentally investigate the short-term and carry-over effects of communal breeding on parental behaviour and fitness in burying beetles (*Nicrophorus vespilloides*). Sexually mature beetle adults search for carcasses of small vertebrates (e.g. mice and birds), help each other in pairs or groups to bury the carcass, lay eggs and then raise the larvae on the buried carcass until developmental independence [59–62]. Although these beetles can breed in pairs, more than two individuals of the same-sex may form groups, cooperate in carcass burial, and breed communally by sharing a carcass, especially when the carcass is large [15,49,50,63]. In such associations, social dominance is largely established through several rounds of fights, with larger individuals being more likely to win fights and become dominant compared with smaller individuals [35,63–65]. Specifically, larger individuals become dominants that can largely monopolize the carcass, while subordinates are smaller individuals and have limited access to the carcass [35,64,65]. Some studies on burying beetles have demonstrated that the decision to breed as a pair or as a group (i.e. communally) is largely determined by the extent to which ecological constraints can be mitigated through mutualistic benefits, e.g. an improved group performance in carcass burial and more effective defence of a large carcass against intruders in a group [15,50,66]. Such mutualistic benefits of group breeding may outweigh the costs that female breeders suffer (e.g. fewer offspring produced) when breeding communally compared with breeding in a pair [63,66–68]. For burying beetles, breeding resources (i.e. carcasses) are ephemeral and limited, which largely determine an individual's breeding opportunity during a given time frame, as well as its reproductive strategy [65,69]. While breeding communally may decrease fitness benefits in reproduction compared with breeding alone or in pairs, breeding communally is still a better option for burying beetles than not breeding at all (i.e. 'the-best-of-a-bad-job' strategy), if they have no other breeding opportunity because of the constraints of breeding resources [59,63,69]. Burying beetles are opportunistic breeders and may have multiple breeding events over their lifetime in a benign environment (e.g. a high resource availability and a low interspecific competition pressure) [69,70]. It can be hypothesized that breeding communally among individuals evolves as an adaptive strategy, which may largely reduce the reproductive costs in adverse environmental conditions and potentially improve the overall fitness benefits across breeding events and over an individual's lifetime. Thus, it is worth studying the carry-over effects of communal breeding on parental behaviour and fitness across breeding events in burying beetles. It is also important to examine whether and how carry-over effects, concurrent with short-term effects, shape the evolution of group breeding and its adaptation to adverse and rapidly changing environments.

The aims of our study are twofold. First, we investigate whether individuals adjust their parental care behaviour and reduce parental investment in communal groups compared with non-communal groups (pair breeding). Second, we examine the short-term fitness implications of communal breeding and its carry-over effects on parental care and reproduction. To investigate the short-term fitness implications of communal breeding on parental care and reproduction, we measured the parental care of each individual (i.e. time spent providing care on the carcass and weight change) and of groups (i.e. burial degree of carcass and the total amount of parental investment by groups), as well as the reproductive success of groups (i.e. brood size and larval weight) in communal and non-communal breeding. We predict that communal groups produce larger brood sizes compared with non-communal groups. To examine the carry-over effects of communal breeding, we experimentally investigated the impact of previous breeding experience, i.e. communal versus non-communal breeding, on future parental care (e.g. burial degree of carcass for each pair, an individual's time spent providing care and weight change) and reproductive success for each pair in a subsequent pair breeding event. Given that individuals have been shown to reduce their parental investment (e.g. shared investment in carcass burial and provide less post-hatching care towards offspring) during communal breeding [49,55], we predict that individuals with a previous communal breeding experience may be able to allocate more resources to reproduction and gain higher reproductive fitness benefits during a subsequent pair breeding treatment than individuals with a previous pair breeding experience. Additionally, we predict that these positive carry-over effects of communal breeding on reproduction and fitness for individuals and groups are governed through the direct effect of the reduced parental investment during previous breeding events. If sex-specific differences in individual resource allocation during communal breeding are more pronounced than during pair breeding, we also expect sex differences in carry-over effects of communal breeding history on future fitness.

# 2. Material and methods

## 2.1. Set-up of laboratory population of burying beetles

Wild beetles were caught during the reproductive season (from late April to September) in 2016 and in 2017 in pitfall traps buried in the forest soil. Beetles were caught at two locations: the estate 'Vosbergen', Eelde (53°08′ N, 06°35′ E), and the University of Groningen Zernike campus, Groningen (53°14′ N, 06°32′ E), both in The Netherlands. These wild beetles were transplanted to the Animal Facility at University of Groningen under laboratory conditions, and their descended, outbreed, second-generation offspring were used in experiments 1 and 2. During the entire rearing period, four to six adult beetles of the same sex were kept in plastic boxes ($23 \times 19$ cm and 12.5 cm high), and fed mealworms twice a week. All beetles were reared in small and similarly sized groups in the absence of a breeding resource (e.g. mouse carcass), and thus, the social environment in early life was similar for all beetles. During the entire rearing and experimental period, all beetles were maintained at 20°C with a 16 : 8 h light to dark photoperiod.

## 2.2. Experimental protocols

### 2.2.1. Experiment 1: Short-term implications of communal versus non-communal breeding on parental care and reproduction

**Communal versus non-communal breeding**
To investigate the immediate implications of communal breeding on parental care and reproductive success, we set up double-pair (consisting of two pairs, i.e. one large pair and one small pair) and single-pair (one male and one female) treatments to create communal and non-communal breeding (i.e. pair breeding) events, respectively [63,71] (figure 1). We selected sexually mature adult beetles aged between 10 and 14 days at post-eclosion for our experiments. Each individual was sexed according to morphological traits, and body size was measured (protonum width; accuracy: 0.01 mm) [72]. Before the experiment, similarly sized, unrelated male and female individuals (mean$_{\,|\,\text{female}-\text{male}\,|}$ ± s.e. = 0.11 mm ± 0.02) were paired and kept for 8–12 h in the same box ($10 \times 5$ cm and 8.5 cm high) filled with 1 cm of clean peat, to ensure female insemination and partner recognition [49,63]. Then, each pair of beetles was classified as large- and small-pair groups according to their size, and pairs within groups were similar in body size (mean size ± s.e. for large pair: 4.92 mm ± 0.04; for small pair: 4.37 mm ± 0.05). In burying beetles, adult body size determines individual competitive ability, such that larger individuals are more likely to monopolize a carcass after fights and to have a higher reproductive success compared with smaller individuals [64,73,74]. To create the communally breeding treatment, one large pair (dominants) and one small pair (subordinates) of beetles ($n = 19$) were selected, and we ensured that the two pairs of beetles were unrelated and differed by approximately 10% in body size, because a stable dominance hierarchy in carcass use is more likely when the size difference between opponents is larger [63,71]. Previous studies indicated that the double-pair breeding treatment could create a communal breeding event [75,76]. For these beetles, an individual's body size has also been found to influence its parental investment towards the current brood, i.e. larger individuals are likely to invest more in the current brood than smaller individuals [77]. Thus, we randomly selected some pairs of beetles from small-pair ($n = 10$) or large-pair ($n = 10$) groups to create the non-communal breeding treatment, which could control for the effect of body size on individual behaviour (i.e. parental investment) [77,78]. We weighed individuals (accuracy: 0.0001 g) immediately prior to the onset of the experiment. We chose a large mouse carcass ($25.0 \pm 2.0$ g) as the breeding resource, because communal breeding is more likely to take place on carcasses larger than 25 g in *Nicrophorus vespilloides* [49,63]. In another study, we tested the parentage of offspring produced in double-pair breeding groups using five microsatellite loci [79] and found that small pairs of individuals produced a relatively small proportion of offspring in a shared brood (9 out of 10 small females reproduced; 5 out of 10 small males reproduced; $N_{\text{brood}} = 10$, $N_{\text{offspring}} = 204$) [75], indicating that communal breeding events (i.e. one shared brood) were indeed induced in such double-pair treatment. All breeding events took place in a transparent box ($23 \times 19$ cm and 12.5 cm high) filled with 3 cm of moist peat. At the onset of each treatment, beetles were placed in breeding boxes, where a thawed mouse was introduced as the breeding resource. To recognize individuals during observation, each individual was marked by making small holes in the elytra with a size 00 insect pin.

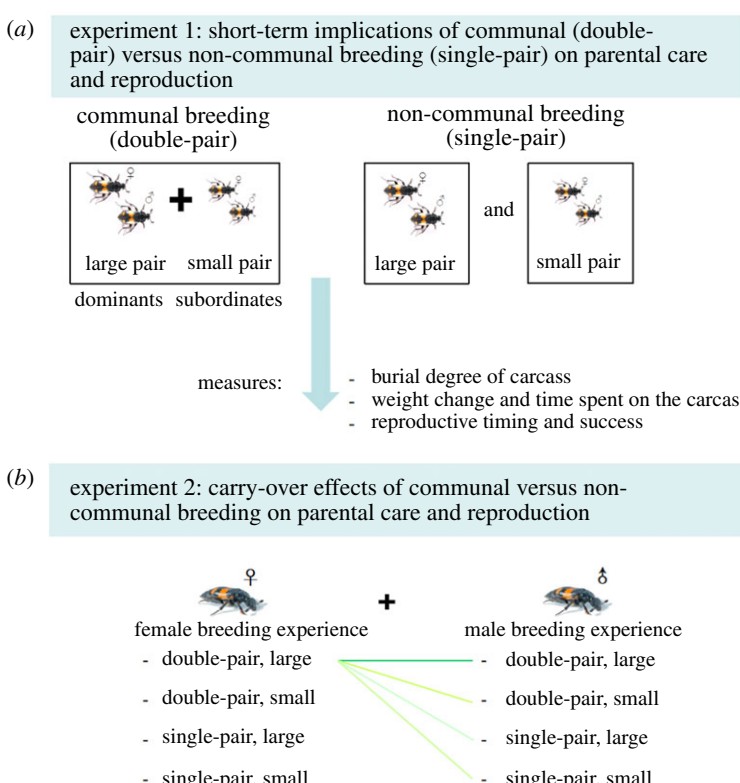

**Figure 1.** Schematic overview of the experimental set-up. (*a*) Experiment 1: Short-term implications of communal (double-pair) versus non-communal breeding (single-pair) on parental care and reproduction. To investigate the short-term implications of communal breeding on parental investment and reproduction, we performed double-pair (consisting of one large pair and one small pair) and single-pair (one male and one female) treatments to create communal and non-communal breeding events, respectively. (*b*) Experiment 2: Carry-over effects of communal versus non-communal breeding on parental care and reproduction. To examine the effects of previous breeding experience (communal versus non-communal breeding) on parental investment and reproduction in the subsequent breeding event (i.e. non-communal breeding), orthogonal experiments were performed using beetles from experiment 1.

### Measurement of parental care and reproductive success

During the entire reproductive period (from the onset of experimentation until larval dispersal), beetle activity on and around the carcass was checked twice daily by visual inspection at 9.00 and 17.00. For each check, beetles were observed for 30 s per group continuously to ascertain whether or not they were providing parental care on the carcass. Such continuous observations could exclude any cases where beetles were present on the carcass for other reasons (e.g. wandering on the carcass) than to provide parental care [69]. We recorded parental care behaviour when an individual provided indirect parental care (i.e. carcass guarding and maintenance on the surface of the carcass), and/or provided direct parental care (i.e. larvae provisioning inside the carcass) [50,80,81]. For each individual, we defined individual parental investment time by calculating the proportion of time that each individual was observed providing parental care on the carcass (i.e. carcass preparation and offspring provisioning) during the entire observation period [49,82,83]. We also defined the total amount of time spent on parental care by groups (including communal and non-communal groups) by calculating the proportion of total times that all individuals in groups were found to provide care on the carcass. As a proxy for the cooperation of breeding groups or breeding pairs in parental care, the degree of carcass burial was estimated according to the fraction of the mouse above the ground and carcass roundness [84]. In burying beetles, the degree of carcass burial over time may be associated with resource protection, i.e. a faster rate in carcass burial is likely to reduce the probability of being found or usurped by other intruders (e.g. flies) [63,66].

In communal breeding events, dominant individuals largely monopolize the carcass, while both dominant and subordinate females are able to reproduce offspring by laying eggs surrounding the carcass [64,65,74]. This results in females laying their eggs earlier (which may hatch earlier) and synchronously laying eggs with other female cobreeders [65,71,76]. Therefore, we examined the timing

of egg laying and larvae hatching in communal and non-communal groups. Egg-laying and larvae-hatching time was recorded as the time from the start of the experiment until the onset of egg laying and larvae hatching, respectively. We defined 'the onset of egg laying' as the onset observational time when some eggs were found at the bottom of the box, and we defined 'the onset of larvae hatching' as the onset observational time when some newly hatched larvae were found. We also calculated the egg-laying period (the period of time from the onset of egg laying until the onset of larvae hatching). As measures for offspring development and reproductive output of groups, we used the larvae-dispersing time (the time from the start of the experiment until the onset of larvae dispersing) and the brood size (number of larvae) and the average larval weight (total weight of larvae/brood size) of groups at larval dispersal (i.e. larvae dispersed from the carcass), respectively. We recorded weight change during breeding ([final weight – initial weight]/initial weight) and survival of adult individuals at the end of the experiment, as parameters for parental investment during the entire breeding period and reproductive costs for individuals, respectively [49,57]. For burying beetles, an individual's weight change during the entire breeding period is a mixture of the costs of providing parental care and the benefits of consuming parts of the carcass [49]. After the first breeding event, surviving beetles were kept individually for 5 days in rearing boxes (10 × 5 cm and 8.5 cm high) and were not fed with any food (i.e. mealworms) prior to subsequent experimentation. We did not feed beetles during this period to avoid the potential effect of food consumption and weight change on an individual's behaviour in the subsequent period.

### 2.2.2. Experiment 2: Carry-over effects of communal versus non-communal breeding on parental care and reproduction

*Communal versus non-communal breeding experience*
To examine the effects of previous breeding experience (communal versus non-communal breeding) on parental care and reproductive success in the subsequent breeding event, orthogonal experiments were performed using beetles from experiment 1 (figure 1). In contrast to the first breeding events, the second breeding events were carried out solely as pair breeding (one pair of individuals bred on the carcass). For this experiment, beetles that had not formed pairs with each other in the previous experiment, and originated from communal (i.e. double-pair treatment) or non-communal groups (i.e. single-pair treatment), were paired randomly; e.g. large females from communal groups were paired with either large or small males from communal or non-communal groups, and small females from communal groups were paired with either large or small males from communal or non-communal groups (figure 1). For newly formed breeding pairs in the second breeding event (without other conspecifics), all males may experience some paternity uncertainty because females mated in the first experiment. However, in the second experiment, the uncertainty of paternity was unlikely to have an effect on an individual's parental care [85–87]. Then, each pair of individuals was left to induce a pair breeding event on a small mouse carcass ($15.0 \pm 1.0$ g), because single pairs are more likely to use small-sized carcasses [60,88]. Just prior to the experiment, all individuals were weighed.

*Measurement of parental care and reproductive success*
During the entire reproductive period of this second experiment, burial degree, time spent providing care on the carcass for each individual, and reproductive and developmental timing were recorded following the same protocol as in the first experiment. For the second breeding event, we also defined the total parental investment time of pairs by calculating the proportion of total times that both female and male individuals were found providing care on the carcass. At larval dispersal, brood size (number of larvae) and average larval weight (total weight of larvae/brood size) were measured as indicators of reproductive success of pairs. We recorded weight change and survival of individuals (described below), as parameters for parental investment and reproductive costs for individuals, respectively.

## 2.3. Statistical analyses

All analyses were performed using R v. 3.6.3 (R Core Team 2018). The best-fitting models with the lowest AIC values were selected using the stepAIC function, and only the statistics for the terms that were included in these models are reported.

In experiment 1, the degree of carcass burial was analysed using a linear mixed model (LMM) fitted with a normally distributed error structure using observational time, breeding group (communal versus non-communal), and their interaction as fixed factors, and brood identity as a random factor. Time spent

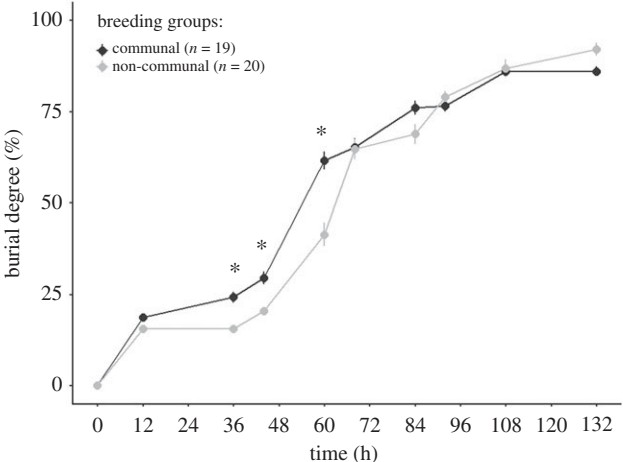

**Figure 2.** Mean (±s.e.) burial degree of the mouse carcass in communal and non-communal groups of burying beetles over time. See electronic supplementary material, table S1 for statistical analysis. Asterisks indicate significance $p < 0.05$.

providing care on the carcass by each individual was analysed with a generalized mixed linear model (GLMM) with a binomial error structure using body size (large versus small), breeding group (communal versus non-communal), sex and their interactions as fixed factors, and brood identity as a random factor. We ran a GLMM with a binomial error structure to analyse the total amount of time spent by breeding groups, using breeding group as a fixed factor, and brood identity as a random factor. We used linear models (LMs) or generalized linear models (GLMs) with Poisson error structures to analyse developmental and reproductive timing (including egg-laying, larvae-hatching and larvae-dispersing time), and reproductive success (including brood size and averaged larval weight), using breeding group as a fixed factor.

In experiment 2, we used generalized linear models (GLMs with Poisson or binomial error structures) or linear models (LMs with a Gaussian error structure) to analyse time spent providing parental care on the carcass, weight change and mortality of each individual, using an individual's own and its partner's previous breeding experience (communal versus non-communal) and size (large versus small), and their interactions as fixed factors. We also ran a GLM with a binomial error structure to analyse the total amount of time spent by each pair of individuals in pair breeding events, using previous breeding experience and size of females and males and their interactions as fixed factors. We analysed the degree of carcass burial ($n = 237$) using observational time, previous breeding experience (communal versus non-communal) and size (large versus small) of females and males and their interactions as fixed factors, and analysed developmental and reproductive timing and reproductive success using previous breeding experience and body size of females and males and their interactions as fixed factors. For burying beetles, the previous parental investment of individuals (e.g. parental investment time and weight change during previous breeding events) may influence their subsequent parental behaviour [54,89]. To further test whether the potential carry-over effects of communal and non-communal breeding on an individual's parental performance in the future breeding events are manifested through the direct effect of the previous parental investment, we also analysed an individual's parenting investment time and weight change at the second breeding event, using GLMs or LMs fitted with binomial or Gaussian error structures. In these models, we included sex, size (large versus small), previous breeding experience (communal versus non-communal), parental investment time and weight change of each individual at experiment 1, parental investment time by its partner and their interactions as fixed factors.

# 3. Results

## 3.1. Experiment 1: Short-term implications of communal versus non-communal breeding on parental performance and fitness

Our results showed that communal groups performed better in carcass burial at the initial stage of the experiment, since communal groups were significantly faster in burying carcasses than non-communally breeding groups at 36, 44 and 60 h (36 h: $t = 3.16$, $p = 0.02$; 44 h: $t = 3.25$, $p = 0.01$ and 60 h:

$t = 7.28$, $p < 0.001$; figure 2; electronic supplementary material, table S1). After around 68 h, there was no difference in the degree of carcass burial between communal and non-communal groups (figure 2; electronic supplementary material, table S1). In the communal groups, the larger pairs of individuals became dominant and spent significantly more time on the carcass during the entire breeding period than the smaller pairs of individuals, which were subordinates (mean ± s.e. of large versus small individuals: 58.96 ± 2.95 versus 7.22 ± 1.33%; $\chi^2 = 315.65$, $p < 0.001$; table 1). At the individual level and within size classes, large females spent a similar amount of time providing care on the carcass in communal and non-communal groups (mean ± s.e. in communal versus non-communal groups: 64.79 ± 4.64 versus 74.89 ± 4.07%; hereafter mean ± s.e. shown as communal versus non-communal groups), while large males in communal groups spent significantly more time on the carcass than large males in non-communal groups (mean ± s.e.: 53.43 ± 3.23 versus 34.24 ± 3.79%; figure 3a and table 1a and electronic supplementary material, table S2). Both large females and large males spent a similar amount of time on the carcass in communal and non-communal groups, while small females and small males spent significantly less time on the carcass in communal groups than in non-communal groups (mean ± s.e. of small females: 10.34 ± 2.22 versus 68.37 ± 6.80%; of small males: 4.09 ± 1.13 versus 41.11 ± 6.08%; figure 3a and table 1a and electronic supplementary material, table S2). Moreover, at the group level, communal groups spent a significantly higher total amount of time on the carcass than non-communal groups (mean ± s.e.: 33.09 ± 1.78 versus 27.33 ± 1.57%; table 1a).

On average, individuals lost weight during breeding (mean ± s.e. = −24.11 ± 2.74%, $n = 78$; table 1a), which indicates that parental investment during breeding was costly. No difference in weight change was observed for large females, small females and large males in communal and non-communal groups, whereas small males in communal groups lost significantly less weight than small males in non-communal groups (figure 3b and table 1a and electronic supplementary material, table S2). Mortality at larval dispersal did not differ for (i) large females, (ii) small females and (iii) larger males in communal and non-communal groups; however, mortality was higher for small males in communal groups than for small males in non-communal groups (see electronic supplementary material, figure S1).

Breeding group (communal versus non-communal) had no effect on brood size but had a significant effect on average larval weight (electronic supplementary material, table S1). Communal groups laid their eggs significantly earlier and produced lighter larvae compared with non-communal groups (mean ± s.e. of egg-laying time: 77.00 ± 5.16 h, $n = 16$ versus 93.33 ± 5.34 h, $n = 18$, $\chi^2 = 4.75$, $p = 0.03$; of average larval weight: 0.1160 ± 0.0090 g, $n = 13$ versus 0.1457 ± 0.0070 g, $n = 11$, $F = 6.39$, $p = 0.02$; electronic supplementary material, table S1). Both groups had a similar egg-laying period and larvae-hatching time (mean ± s.e. of egg-laying period: 94.67 ± 4.78 h, $n = 15$ versus 83.14 ± 6.28 h, $n = 14$, $\chi^2 = 2.88$, $p = 0.09$; of larvae-hatching time: 143.59 ± 7.58 h, $n = 15$ versus 149.14 ± 7.30 h, $n = 14$, $\chi^2 = 1.18$, $p = 0.28$), and at larval dispersal, brood size of offspring was also similar (mean ± s.e.: 15.85 ± 1.65, $n = 13$ versus 14.73 ± 1.61, $n = 11$; $F = 0.23$, $p = 0.64$; electronic supplementary material, table S1).

## 3.2. Experiment 2: Carry-over effects of communal versus non-communal breeding on parental performance and fitness

Previous breeding experience of individuals had a significant effect on parental performance in carcass burial (table 2 and electronic supplementary material, table S3). Females originating from communal groups had lower rates of carcass burial than females from non-communal groups (experience: $F = 26.41$, $p < 0.001$; experience × time: $F = 7.67$, $p = 0.01$; figure 4a and table 2 and electronic supplementary material, table S3), whereas males originating from communal and non-communal groups had similar rates of carcass burial (experience: $F = 9.09$, $p = 0.005$; experience × time: $F = 2.34$, $p = 0.13$; figure 4b and table 2 and electronic supplementary material table S3). Moreover, the interaction of previous breeding experience of females and males had a significant influence on the rates of carcass burial, since the rate of carcass burial was higher in pairs with females originating from non-communal groups and males originating from communal groups than in other pairs (female experience × male experience × time: $F = 4.84$, $p = 0.03$; electronic supplementary material, figure S2 and table S3).

The previous breeding experience of individuals in a pair influenced their parental investment time (table 1b). At the individual level, females and males originating from communal or non-communal groups spent a similar parental investment time on the carcass (mean ± s.e. of females: 61.15 ± 5.93%, $n = 21$ versus 78.92 ± 6.84%, $n = 13$; of males: 38.56 ± 7.50%, $n = 20$ versus 42.45 ± 5.91%, $n = 14$; figure 4c and table 1b and electronic supplementary material, table S2) and had similar weight

**Table 1.** Factors with short-term and carry-over effects on parental investment and mortality. Sample sizes are shown in this table. Data in bold (including explanatory variables and *p*-values) indicate statistically significant results ($p < 0.05$).

| response variables | explanatory variables | estimate | s.e. | $\chi^2$/F-value | *p*-value |
|---|---|---|---|---|---|
| *(a)* short-term effects | | | | | |
| individual time spent providing care on the carcass | | | | | |
| (d.f. = 7, 115) | intercept | 0.62 | 0.16 | | |
| | **body size**[a] | **−2.83** | 0.24 | 315.65 | **<0.001** |
| | **breeding group**[b] | **0.49** | 0.29 | 10.98 | **<0.001** |
| | **sex**[f] | **−0.50** | 0.18 | 71.13 | **<0.001** |
| | **body size: breeding group** | 2.51 | 0.41 | 58.32 | **<0.001** |
| | body size: sex | −0.52 | 0.41 | 0.58 | 0.44 |
| | **breeding group: sex** | −1.29 | 0.32 | 12.38 | **<0.001** |
| | **body size: breeding group: sex** | 1.13 | 0.55 | 4.27 | **0.04** |
| individual weight change during reproduction | | | | | |
| (d.f. = 7, 78) | intercept | −34.28 | 5.66 | | |
| | **body size**[a] | 19.90 | 9.44 | 9.66 | **0.003** |
| | breeding group[b] | −7.71 | 9.44 | 0.61 | 0.44 |
| | sex[c] | 6.96 | 8.46 | 2.01 | 0.16 |
| | body size: breeding group | 8.15 | 15.21 | 1.04 | 0.31 |
| | body size: sex | −0.04 | 13.41 | 1.92 | 0.17 |
| | breeding group: sex | 18.85 | 14.20 | 0.06 | 0.81 |
| | body size: breeding group: sex | −36.57 | 21.27 | 2.96 | 0.09 |
| total amount of time spent by all individuals | | | | | |
| (d.f. = 1, 38) | intercept | −0.71 | 0.08 | | |
| | **breeding group**[b] | −0.28 | 0.11 | 6.28 | **0.01** |
| *(b)* carry-over effects | | | | | |
| individual time spent providing care on the carcass | | | | | |
| (d.f. = 8, 67) | intercept | 1.32 | 0.38 | | |
| | **sex**[f] | **−1.35** | 0.29 | 20.98 | **<0.001** |
| | experience[d] | −0.23 | 0.44 | 3.34 | 0.07 |
| | **partner's experience**[e] | **−0.93** | 0.44 | 9.44 | **0.003** |
| | **body size**[f] | **−0.30** | 0.38 | 4.34 | **0.04** |
| | partner's body size | 0.01 | 0.38 | 0.94 | 0.34 |
| | experience: partner's body size | 0.80 | 0.66 | 0.56 | 0.46 |
| | partner's experience: body size | −1.44 | 0.71 | 3.47 | 0.07 |
| | **experience: partner's experience** | 1.41 | 0.64 | 5.00 | **0.03** |
| individual weight change during reproduction | | | | | |
| (d.f. = 7, 36) | intercept | 21.80 | 12.59 | | |
| | sex[c] | −6.17 | 11.09 | 0.99 | 0.32 |
| | experience[d] | 1.34 | 13.44 | 1.76 | 0.19 |
| | partner's experience[e] | −10.12 | 16.73 | 0.005 | 0.94 |
| | body size[a] | 24.52 | 18.01 | 0.02 | 0.88 |
| | partner's body size[f] | −18.90 | 13.53 | 1.05 | 0.31 |
| | experience: body size | −54.46 | 27.68 | 2.41 | 0.13 |
| | partner's experience: partner's body size | 40.26 | 26.52 | 2.30 | 0.14 |

(*Continued.*)

**Table 1.** (*Continued.*)

| response variables | explanatory variables | estimate | s.e. | $\chi^2$/F-value | p-value |
|---|---|---|---|---|---|
| individual mortality | | | | | |
| (d.f. = 4, 67) | intercept | −1.57 | 0.61 | | |
| | sex[c] | 0.82 | 0.59 | 1.03 | 0.31 |
| | **body size**[a] | 1.48 | 0.61 | 5.60 | **0.02** |
| | **experience**[d] | −1.68 | 0.68 | 7.00 | **0.01** |
| | partner's experience[e] | 1.05 | 0.59 | 3.28 | 0.07 |

[a]Small individuals relative to large individuals.

[b]Non-communal breeding relative to communal breeding.

[c]Males relative to females.

[d]Individuals with non-communal breeding experience relative to individuals with communal breeding experience.

[e]Partners with non-communal breeding experience relative to partners with communal breeding experience.

[f]Small partners relative to large partners.

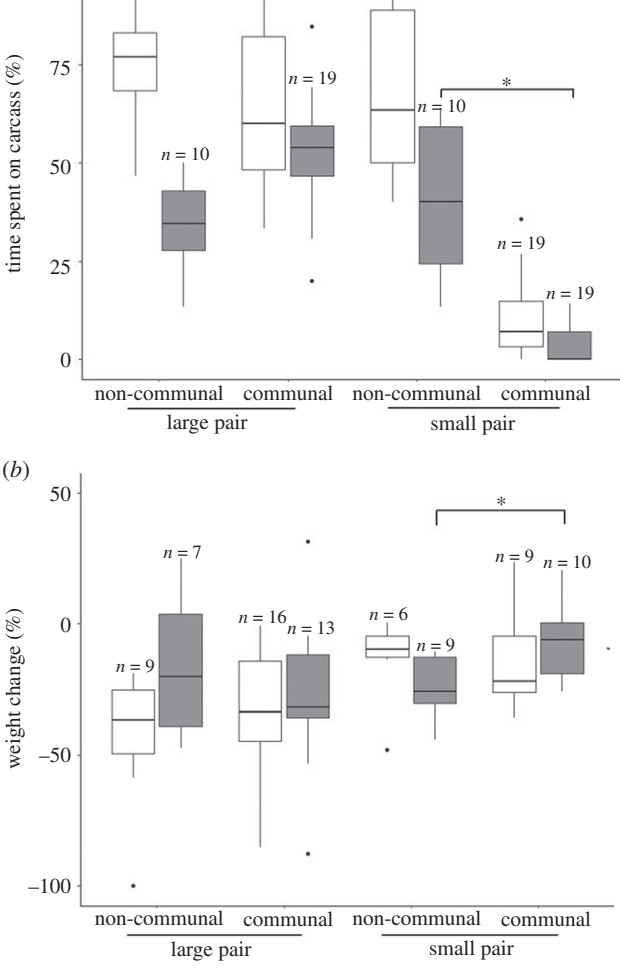

**Figure 3.** Short-term implications of communal versus non-communal breeding on individual parental investment. (*a*) Time spent on the carcass and (*b*) weight change for individuals, for pairs of different size and breeding groups. In (*a*) and (*b*), white boxes are females and grey boxes are males. Sample sizes are shown in graphics. Table 1*a* is for statistical analysis. Asterisks indicate significance $p < 0.05$.

**Table 2.** Parental investment in current reproduction and carry-over effects on fitness benefits.

| | | parental investment in current reproduction (communal versus non-communal breeding) | | | carry-over effects on fitness benefits (communal versus non-communal breeding) | | reproductive timing | | reproductive success | |
| | | time spent providing | | | | | | | | |
| sex | body size | care | weight change | mortality | burial degree | mortality | larvae hatching | larvae dispersing | brood size | average larval weight |
| --- | --- | --- | --- | --- | --- | --- | --- | --- | --- | --- |
| female | large | n.s. | n.s. | n.s. | − | + | − | − | n.s. | n.s. |
| | small | − | n.s. | n.s. | − | + | − | − | n.s. | n.s. |
| male | large | + | n.s. | n.s. | n.s. | + | n.s. | n.s. | + | + |
| | small | − | + | + | n.s. | + | n.s. | n.s. | + | n.s. |

Notes: Significant influences of breeding group (communal versus non-communal breeding), and previous breeding experience (communal versus non-communal breeding) on parental investment and future fitness benefits for females and males are shown as: positive (+), negative (−) and no significance (n.s.). Mortality: increased (+) or decreased (−) risk of mortality. Reproductive timing: earlier (+) or later (−) onset time of larvae hatching and larvae dispersing.

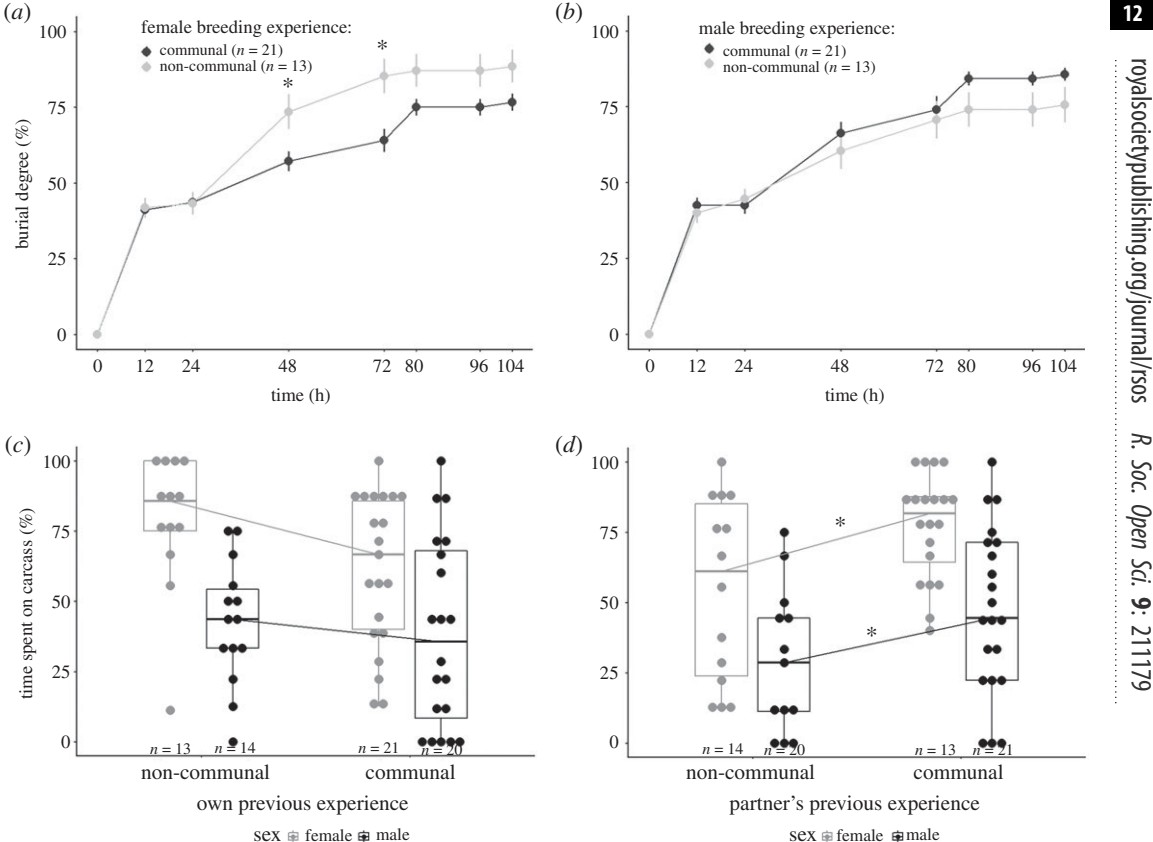

**Figure 4.** Carry-over effects of communal breeding on parental investment. Effects of previous breeding experience (communal versus non-communal breeding) on mean (±s.e.) burial degree by (a) females and (b) males, time spent on carcasses by females and males (c,d). In (c) and (d), grey (females) and black (males) dots indicate raw data. Sample sizes are shown in graphics. Table 1b and electronic supplementary material, tables S2 and S3 are for statistical analysis. Asterisks indicate significance $p < 0.05$.

changes during breeding (table 1b and electronic supplementary material, table S2). Regardless of their own previous breeding experience, female and male individuals spent significantly more time on the carcass when their partners originated from communal groups than from non-communal groups (mean ± s.e. of females' partner: $77.19 \pm 4.15\%$, $n = 13$ versus $54.75 \pm 8.74\%$, $n = 14$; of males' partner: $46.99 \pm 6.44\%$, $n = 21$ versus $29.12 \pm 7.10\%$, $n = 20$; figure 4c,d and table 1b and electronic supplementary material, table S2). In addition, when their partners originated from non-communal groups, both female and male individuals originating from communal groups spent significantly less time on the carcass than those from non-communal groups (mean ± s.e. of females: $44.88 \pm 9.45\%$, $n = 9$ versus $72.50 \pm 15.85\%$, $n = 5$; of males: $20.41 \pm 8.64\%$, $n = 8$ versus $43.06 \pm 10.24\%$, $n = 5$; $z = -3.01$, $p = 0.005$; table 1b and electronic supplementary material, table S2). However, when their partners originated from communal groups, individuals originating from communal or non-communal groups spent similar times providing care on the carcass (mean ± s.e. of females: $73.36 \pm 5.62\%$, $n = 12$ versus $82.94 \pm 5.90\%$, $n = 8$; of males: $50.65 \pm 11.38\%$, $n = 12$ versus $42.11 \pm 7.68\%$, $n = 9$; $z = -0.43$, $p = 0.89$; table 1b and electronic supplementary material, table S2). At the pair level, the interaction of female and male breeding experience had a significant effect on the total parenting investment time by pairs (table 1b). When females originated from communal groups, there was a higher total parental investment time in the presence of males originating from communal groups than from non-communal groups (mean ± s.e.: $62.00 \pm 5.82\%$, $n = 12$ versus $43.50 \pm 5.50\%$, $n = 9$; $z = 3.04$, $p = 0.005$; table 1b and electronic supplementary material, table S2), whereas in groups with females originating from non-communal groups, the total investment time by pairs was similar when males originated from communal or non-communal groups (mean ± s.e.: $51.67 \pm 3.94\%$, $n = 8$ versus $57.78 \pm 9.39\%$, $n = 5$; $z = -0.82$, $p = 0.65$; table 1b and electronic supplementary material, table S2). For both females and males, we found that individuals originating from communal groups suffered higher mortality after breeding than those from non-communal groups (19 out of 41 versus 5 out of 27; table 1b).

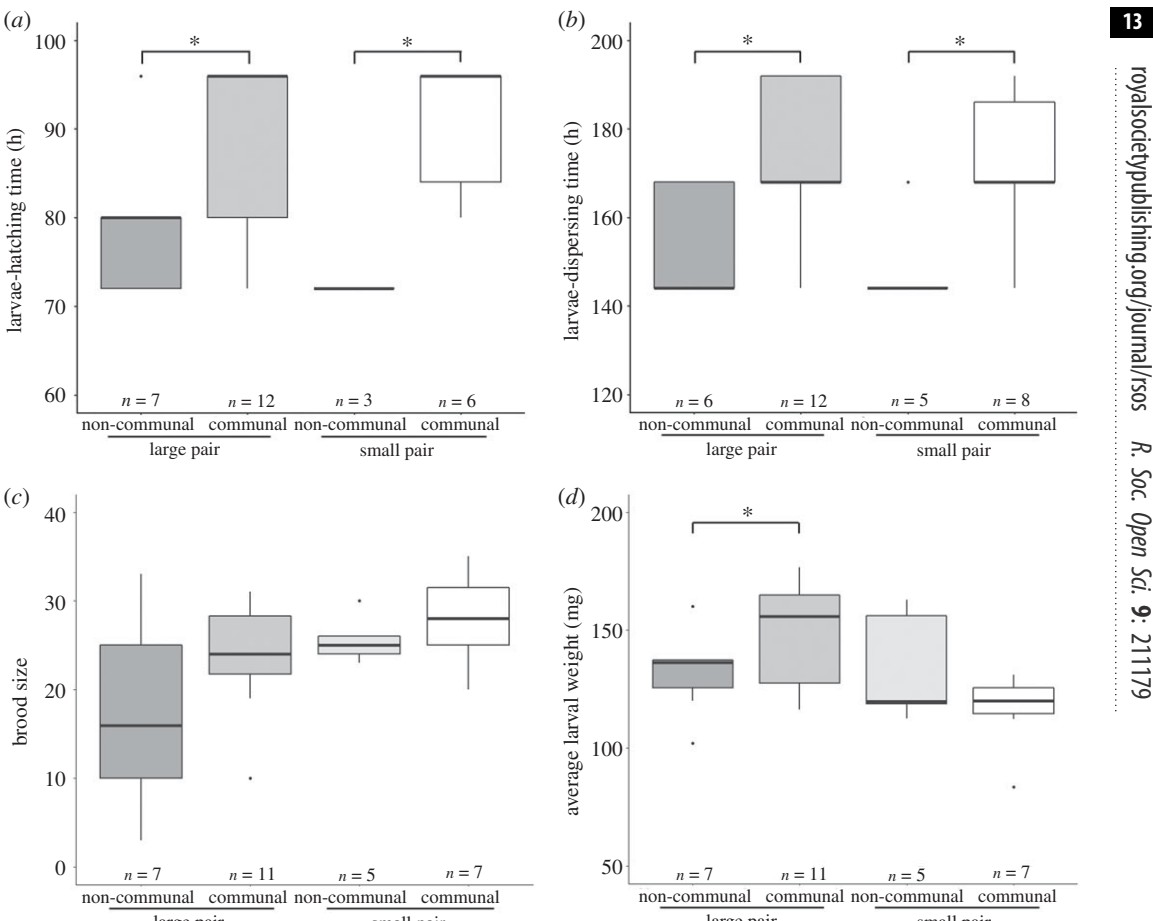

**Figure 5.** Carry-over effects of communal breeding on future fitness. Effects of previous breeding experience on (*a*) larvae-hatching time, (*b*) larvae-dispersing time for large and small females from communal or non-communal groups, (*c*) brood size and (*d*) average larval weight for large and small males from communal or non-communal groups in burying beetles. Table 1*b* and electronic supplementary material table S3 are for statistical analysis. Asterisks indicate significance *p* < 0.05.

Additionally, an individual's parental investment (i.e. parenting investment time and weight change of each individual) was not associated with its weight change during the previous communal breeding events (electronic supplementary material, table S4).

Females originating from communal groups produced larvae that hatched and dispersed later from the carcass compared with females originating from non-communal groups (mean ± s.e. of larvae-hatching time: 88.89 ± 2.23 h, *n* = 18 versus 76.80 ± 2.44 h, *n* = 10; of larvae-dispersing time: 172.00 ± 4.00 h, *n* = 18 versus 151.40 ± 3.20 h, *n* = 13; figure 5*a,b* and table 2 and electronic supplementary material, table S3), whereas no difference was observed for males originating from communal and non-communal groups in the timing of offspring development (table 2 and electronic supplementary material, table S3). Regardless of male's previous breeding experience, females originating from communal and non-communal groups had similar reproductive success (mean ± s.e. of brood size: 22.61 ± 2.15, *n* = 18 versus 25.08 ± 1.28, *n* = 13; of average larval weight: 0.1389 ± 0.0067 g, *n* = 18 versus 0.1333 ± 0.0053 g, *n* = 13; table 2 and electronic supplementary material, table S3). However, females originating from communal groups produced smaller brood size than females originating from non-communal groups when males originated from non-communal groups (mean ± s.e.: 17.57 ± 3.93, *n* = 7 versus 25.40 ± 2.56, *n* = 5; *z* = −2.36, *p* = 0.04; electronic supplementary material, table S3). Males originating from communal groups produced larger brood size than males originating from non-communal groups (mean ± s.e.: 25.42 ± 1.33, *n* = 19 versus 20.83 ± 2.70, *n* = 12), whereas the average larval weight was similar for males originating from communal and non-communal groups (mean ± s.e.: 0.1389 ± 0.0006 g, *n* = 19 versus 0.1328 ± 0.0056 g, *n* = 12; electronic supplementary material, table S2). Additionally, the size of males originating from communal and non-communal groups significantly influenced the reproductive success (electronic supplementary material, table S3). In particular, large males originating from communal groups produced heavier larvae than large males

from non-communal groups (figure 5*d* and table 2 and electronic supplementary material, table S3), whereas the reproductive success was similar for small males originating from communal and non-communal groups (figure 5*c,d* and table 2 and electronic supplementary material, table S3 and figure S3).

# 4. Discussion

## 4.1. Short-term and carry-over effects of communal breeding on parenting performance and fitness

We found that communal breeding appeared to have no short-term effects on parental investment and reproductive success. However, we found carry-over effects on parental investment and reproduction. Our results indicated that communally breeding in groups had improved carcass burial performance and increased total time spent on parental investment, whereas individual parental investment time was similar in communal and non-communal groups. Our findings on the short-term effects of communal breeding are consistent with previous studies on burying beetles, which indicates that individuals reproducing in communal groups do not have a higher reproductive success than individuals breeding in pairs [59,63,64,66,90,91]. Some studies on burying beetles suggest possible mutualistic benefits in communal breeding for some aspects of grouping performance, such as an enhanced performance of carcass burial and the improved capacity of defending carcasses against interspecific competitors [15,66,68,91]. However, such benefits seem not to compensate for the reproductive costs of each individual in benign environmental conditions, since females reproducing in such groups may have fewer offspring than females breeding in pairs [59,63,64]. Our results further suggest that such reproduction costs have a negative influence on offspring fitness, i.e. lighter larvae produced in communal breeding, which may be due to a high level of group conflict over reproduction between individuals [65,67,71]. In burying beetles, the weight of larvae dispersing from the carcass was positively associated with their body size when they emerged as adults, and adult body size subsequently determines an individual's competitive ability (i.e. fighting ability) and its advantages in resource monopolization and reproduction [64,73,74]. Even though the presence of other members is suggested to benefit dominant breeders' fitness traits in some burying beetle species [16,23,24,47], this scenario is absent for other burying beetle species, since a single, limited breeding resource cannot support the reproduction of all members [15,50,92]. Dominant individuals monopolizing a carcass could often reproduce and have a large proportion of offspring in communal groups [76]. Such dominance may play a limited role in controlling the allocation of reproduction due to the effect of carcass size and individual competitive ability [15,61,69]. Taken together, our findings further support the mutual tolerance hypothesis in communal groups of burying beetles, which suggests that a best-of-the-bad-job strategy deployed by communal breeders benefits the individuals involved, as well as group productivity, because of the rarity of breeding opportunities due to ephemeral and limited resources for breeding and the high costs of expelling other beetles from a large-sized carcass [15,63,65,67,69,93].

Our study showed experimental evidence for carry-over effects of communal breeding on some fitness traits, which were different for males and females. During the subsequent reproduction as pairs, males originating from communal groups gained enhanced benefits in reproduction (i.e. larger brood size) than males originating from non-communal groups. In contrast, females originating from communal groups showed worse parental care as they had lower rates of carcass burial and produced larvae that hatched and dispersed later than larvae produced by females originating from non-communal breeding. Such a low performance in carcass burial was closely linked with a delayed developmental period for offspring, which could then passively influence offspring fitness. Moreover, our results revealed that these carry-over effects on individual parental care and fitness may not be solely determined by the previous parental investment (i.e. weight changes during previous breeding events). Both males and females originating from communal breeding experienced higher mortality compared with individuals originating from non-communal breeding. This may be due to individuals originating from communal breeding incurring higher reproduction and survival costs [57,62,89]. In communal groups, the establishment of a dominance hierarchy is suggested to reduce the extent of aggressive interaction, while injuries are often observed [63,69]. It is likely that these injuries affect an individual's future reproduction and increase the risk of death. In this scenario, a microbe-rich soil environment might increase the risk of infections associated with fight wounds, thereby increasing the rate of mortality [62]. For each individual, dominance status is closely associated with fecundity and nutritional state [35,50]. For example, in *N. vespilloides*, limited access to the carcass by subordinate

females leads to low fecundity caused by nutritional deficiencies [50]. Such effects in fecundity and nutritional state are also accompanied with some negative changes in physiological state and immunity [35,50,62], which may have long-lasting influences on individual fitness benefits in the future, e.g. high mortality and short lifespan for individuals that originate from communal groups. To what extent the high mortality for males and females originating from communal groups may offset their reproductive benefits remains to be investigated [49,54,94]. For males, we also found that the carry-over effects of communal breeding were dependent on their body size, i.e. the enhanced benefits in future reproduction for large males, but not for small males. In particular, dominant males originating from communal groups produced heavier larvae than large males originating from non-communal groups, whereas no difference in reproduction was observed for small males originating from communal and non-communal groups. Dominant males originating from communal groups spent more time on the carcass than large males originating from non-communal groups. These results may suggest that the improved future benefit for dominant males is due to dominant males being able to save more resources and gain more experience than males breeding in pairs [49,52,70,81,95]. Burying beetles could gain fitness benefits from access to carcasses during breeding, because access to resources (e.g. by spending more time on the carcass, and by consuming more of the carcass) while breeding might alleviate the energetic costs of reproduction. As such, individuals with access to additional resources may improve their physical conditions (e.g. body mass and immunity) and sexual attractiveness, which may associate positively with enhanced reproductive success [50,57,59,89,96]. When multiple males compete for one carcass, uncertain and reduced paternity may affect an individual's parental care. For example, when reproductive competition is high, burying beetle males spent more time providing parental care [85]. However, such paternity uncertainty did not affect inferences about carry-over effects from different previous breeding experiences in our study. Females have mated and may have stored sperm from the first breeding event (i.e. communal and non-communal groups), with which she could fertilize eggs produced in the second breeding event (i.e. pair breeding). However, males largely prevent this and thus dilute the sperm of earlier competitors by copulating repeatedly with the same female [85–87].

For burying beetles, an individual's investment in the current reproduction is influenced by resource availability, as well as its previous experience [57,89]. For example, in *Nicrophorus orbicollis*, females reproducing on low-quality carcasses invest less in current reproduction and allocate more to future reproduction compared with females given high-quality carcasses. As such a change in carcass quality may reflect reproductive restraints by females decreasing the amount of resources in the current reproduction and increasing the probability of future reproductive opportunities [57,89]. As such, we suggest that individuals reproducing in communal groups may have enhanced physical conditions and allocate more resources and parental efforts for future reproduction, because of the limited resource availability for each individual and the high pressure of intraspecific competition. As argued above, we lack evidence for the effect of the energetic savings (i.e. weight gains) during previous reproductive events on an individual's subsequent behaviour and future reproduction. This might mean that the carry-over effects of communal breeding on individual parenting behaviour and fitness is not simply determined by the parental investment to previous reproduction, but instead is associated with the individual's previous experience [57,94]. Prior studies have demonstrated age-related differences in reproductive effort and reproductive allocation between breeding events [89,97–99]. In *N. orbicollis*, older individuals have a higher level of investment in the current reproduction (e.g. produce larger broods and consume less of the carcass) compared with younger individuals, because older individuals with a low residual reproductive value have limited future reproductive opportunities [89,98]. These previous studies also highlight the impact of individual reproductive restraint and senescence on individual reproductive effort [89,97]. Moreover, more recent work revealed an interplay between age and previous breeding experience on reproductive investment in burying beetles [62,99,100]. For example, *N. vespilloides* females that invested more in reproduction during earlier reproductive events show a decline in reproductive investment, but this negative effect of previous reproductive investment is observed only in older females [62]. These state-associated changes that may influence individual reproductive residual value should be considered, as they could determine an individual's reproductive effort for future reproduction, probably due to physiological constraints and the lack of experience. Thus, we suggest that the effects of communal breeding on fitness are associated not only with short-term costs and benefits but also with carry-over effects on future fitness [32,59,66,101], albeit such effects are more pronounced for dominant males than for dominant females [49,70,81,95].

Our results also demonstrate that previous experience as communal breeders affects the reproductive cost in terms of the joint parenting investment for females but not for males in pairs. These results

demonstrate that females may have a reduced investment in care and perform less well during reproduction (e.g. slow rates of carcass burial) because of their experience as communal breeders, which may be due to females incurring more costs in reproduction from breeding in communal groups than breeding in non-communal groups [35,59,89]. Female and male individuals did not adjust their level of parental care (i.e. parental investment time on the carcass and weight change during breeding) based on their own previous breeding experience; however, both increased their level of care when their partners had prior experience as communal breeders. For breeding pairs, the parental investment by males does not seem to buffer this negative effect of communal breeding experience on female fitness, as well as on the joint parental investment of pairs. This may be because males may have partial or no compensation for a reduction in parental investment by females depending on their own and their partners' history of previous breeding, which is driven by sexual conflict over parental investment between parents [57,80,102,103]. In burying beetles, although parents often cooperate to provide care towards their offspring, females always share the majority of parental care [104,105]. The quantity of female care is not associated with the presence of a male, whereas males may show highly flexible parental behaviour and adjust the amount of care in response to the reproductive state of their partners [80,103,106]. In support of this, pairs with both females and males originating from communal groups provided more care than pairs with females and males originating from communal and non-communal groups, respectively. It is well known that hormonal levels (e.g. juvenile hormones) rapidly change during breeding in burying beetles, and these changes may differ with the social environments (e.g. dominance status) [52,107]. Moreover, these physiological changes may intrinsically mediate the emission of pheromones involved and other physiological conditions, which may have influences on individual reproductive states and parental behaviour in the subsequent period. During breeding, two parents are able to effectively communicate with each other via the emission of pheromones, which could help them to simply recognize their partner's reproductive states and then benefits for coordinating parental and mating effort [108]. We suggest that such adjustments in parental investment depend on their partner's previous breeding experience and may be mediated by pheromone-dependent recognition. That is, each individual is likely to recognize its partner's previous breeding experience via the emission of pheromones, and then strategically adjust its own parental investment towards the current brood [52,108]. We suggest that future work should consider the flexible adjustment of females and males in parental investment based on the previous breeding experience of their partners and the recognition system involved, and its subsequent influence on common benefits for pairs.

## 4.2. Sex differences in the allocation of parental investment may lead to different benefits of communal breeding for males and females

Our experiments indicate that the carry-over effects of communal breeding on fitness benefits were more pronounced for males than for females, suggesting that these may be associated with sex differences in the allocation of parental investment between current and future reproduction [49,54,95,109]. During communal breeding, both females and males are able to shift their parental investment in brood care according to their perceived share in parentage of the brood [49,110–112]. Females that breed in communal groups can shift their resource allocation towards pre-hatching investment (i.e. lay more and larger eggs) and reduce their post-hatching care to larvae compared with females that breed alone. However, there is no evidence that females save more resources (e.g. gain more weight) during communal breeding to allocate more resources towards a future reproduction [49,54]. When multiple males compete for one carcass (e.g. in communal breeding), males may adjust the level of parental care in response to the uncertainty of paternity by reducing their investment in the current brood and allocating more resources towards a future reproduction [49,56]. Previous study shows that *N. vespilloides* males increase their parental investment when same-sex conspecifics are present and reproductive competition is high [63,85]. Males could probably gain compensatory benefits by spending more time on a carcass or consuming more of a carcass [85,113]. Dominant males produce a large proportion of offspring in communal groups because of their monopoly in carcass access. Dominant males are also able to maximize their paternity in the current brood by frequently copulating with dominant females and subordinate females prior to and during the period of egg laying [113,114]. However, dominant males cannot fully suppress the reproduction of males, since subordinate males may occasionally sire a proportion of the offspring by sneakily mating with all females [87,114]. While subordinate males usually leave the carcass earlier and do not provide post-hatching care when the

dominant male is present, the opportunity to access the carcass can motivate male parental behaviour [87,113]. These results indicate that access to the carcass is beneficial for males, because they are able to save resources for themselves or compensate for a reproductive cost due to the uncertainty of paternity. Hence, we suggest that each individual adjusts its allocation of parental care between reproductive events in order to maximize its fitness benefits over its lifetime. Such an allocation of parental care is affected by the interplay of sexual conflict and intraspecific competition over resources and reproduction during communal breeding [49,70,76,81], for the following reasons. First, sexual conflict can lead to a trade-off between parental investment in current and future reproduction for males and females. During communal breeding, females can increase their investment in current broods when intraspecific competition over reproduction is high, while males decrease their investment in current broods and save more parental resources for future reproduction [54,70,115–117]. Second, communal breeding may affect the future fitness more for males than for females, because males, but not females, in communal groups were able to save resources that could be allocated to enhance benefits in future reproduction [54,70,82,95,118,119]. Therefore, we suggest that this sex difference in the allocation of parental investment may affect future fitness benefits differently in the sexes, which is jointly influenced by both external (e.g. the presence of another breeder) and internal factors (e.g. previous breeding experience) [49,89,120]. Furthermore, such difference in a trade-off between current and future reproduction may give rise to sexual conflict in communal breeding [49,84,120]. Future studies with larger sample sizes should investigate the effect of communal breeding experience on the allocation of parental investment and fitness between sexes.

## 5. Conclusion

Our study on burying beetles offers a novel perspective for our understanding of communal breeding and shows no experimental evidence for short-term effects of communal breeding on parental investment. Our study found significant effects of communal breeding on future fitness and sex differences in these carry-over effects. Studies based on the short-term effects are insufficient for the advancement of our understanding of the evolution of group breeding. Living and breeding in social groups may generate positive or negative effects for each individual involved in the short term. Some of these effects (e.g. physiological changes and previous breeding experience) arising from group breeding can mask some direct, short-term fitness benefits. Such effects may also have a long-lasting impact on an individual's future fitness and its adaptation to complex environmental conditions. Future research on the evolution of social behaviour should investigate the importance of the carry-over effects of group breeding, and sex differences in these carry-over effects. Further study should investigate whether and how such carry-over effects on fitness select for variation in parental behaviour and reproductive strategy.

Ethics. The study was reviewed and approved by the Ethics Committee of the University of Groningen.

Data accessibility. Data available from the Dryad Digital Repository: https://doi.org/10.5061/dryad.4tmpg4fb0.

The data are provided in electronic supplementary material [121].

Authors' contributions. L.M.: conceptualization, data curation, formal analysis, investigation, methodology, writing—original draft, writing—review and editing; M.A.V.: formal analysis, methodology, writing—review and editing; M.H.: conceptualization, supervision, writing—review and editing; J.K.: conceptualization, funding acquisition, project administration, supervision, writing—review and editing.

All authors gave final approval for publication and agreed to be held accountable for the work performed therein.

Competing interests. We declare we have no competing interests.

Funding. This work was supported by a grant of The Netherlands Organization for Scientific Research (NWO-TOP-854.11.003) to J.K., and by an Ecology Fund Grant of the Royal Netherlands Academy of Arts and Sciences (KNAWWF/DA/973/Eco2018) to L.M. L.M. was supported by a PhD scholarship from the China Scholarship Council (CSC-201506230168). M.H. was funded by a NWO VENI Fellowship (863.15.020).

Acknowledgements. We thank Sjouke Kingma, Sudeshna Chakraborty and Sacha C. Engelhardt for their helpful comments on the manuscript. We also thank Martijn Salomons, the laboratory manager of Animal Facilities, University of Groningen, for organizing climate rooms for housing burying beetle colony.

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
