## [Peer Review File · Royal Society Open Science]

Review History

RSOS-211179.R0 (Original submission)

Review form: Reviewer 1

Is the manuscript scientifically sound in its present form?

Yes

Are the interpretations and conclusions justified by the results?

No

Is the language acceptable?

No

Do you have any ethical concerns with this paper?

No

Have you any concerns about statistical analyses in this paper?

Yes

Recommendation?

Major revision is needed (please make suggestions in comments)

Comments to the Author(s)

Letter to Authors:

The social environment that an individual experiences may have profound effects on individual reproductive success within and across breeding attempts. Here, the authors argue that early breeding experience has the potential to impact lifetime fitness due to carry-over effects. The carry-over effects of communal breeding may impact individuals in a sex-specific way. Here, the authors conduct an interesting study to tease apart how experience of communal breeding may impact future breeding attempts.

I think it would strengthen the manuscript to have a clearer unifying framework that provides context for the tests and results and clearly states what a) is known/not known and b) what each measure provides inference into. Why was the experiment set up the way that it was, and why were the factors that you measured selected? E.g., small vs large beetles were used, but it is not explicit in the Introduction what this would be a test of (although it is briefly mentioned on line 200). Why was carcass burial, egg laying period, larval hatching time etc. recorded? I'm sure there are good biological reasons to do so, but what do they tell you specifically? E.g., is a longer egg-laying period "bad" for some reason? Why was parental weight change investigated as a metric of parental effort? What is known versus not known for this system? For example - on line 235 it states that weight change was used to gauge parental effort, but then on line 314 it makes it sound like it was unknown whether weight change was correlated with variation in parental investment. Is this a test of assumption or a result or something unexpected based on previous research, and if so, how does it fit into the overall framework of the study? Overall, it would provide clarity to the manuscript to state in the Introduction what kinds of things you need to measure to answer the question, e.g., measures of parental effort, care behavior and competitive ability, and then in the Methods, you define how each of these was measured: e.g., parental effort was measured as amount of time spent on carcass, length of egg-laying period (because this is a reliable proxy of xx), etc.

The experiment is set up to distinguish between age-effects and experience effects (communal to non-communal and non-communal to non-communal), but changes in reproductive investment over aging are not discussed. There are good life history reasons to expect that investment may change over an individual's lifespan, and it needs to be mentioned (at minimum).

Also, for the experiment, why were the mates switched in the second breeding bout (line 248)? The authors mention that paternity assurance may affect behavior, but then are manipulating paternity assurance. Doesn't this species store sperm to some extent? So that males in the second breeding bout will likely not have paternity assurance? Is there any reason to expect that this would influence behavior or investment in later breeding attempts? This is worth mentioning/discussing somewhere, especially since the authors already state that paternity assurance is expected to impact behavior (e.g., line 477).

While the study design does seem well set-up, the sample sizes are both low and uneven. E.g., for the test of carry-over effects, there ends up being comparisons of treatments with an $n=3$ vs $n=12$ (Figure 5, larval hatching times), but these discrepancies are not described. Who was excluded and why? Why don't the sample sizes match up for the same treatments across the figures - e.g., on Figure 4a/b there are 21 communal and 13 non-communal females, and only 20 and 14 males, respectively. The individuals were paired, so where does this disparity come from?

How often do you expect for individuals of this species to have multiple breeding attempts in the wild, given how limited/ephemeral the breeding resource is? If it is reasonably rare – realistically, how important may carry-over effects be for generating variation for selection to act on (e.g., line 392, 420)? How big of deal do we expect first breeding experience to be, if it's almost always the only breeding experience? Is this species appropriate for a test of carry-over effects across breeding bouts? Also, it is perhaps too broad of a stroke to say that breeding communally decreases fitness. If they have no other opportunity to breed besides communally, it's still a better option than not breeding at all.

I think that the results should be re-written with the means \pm SE written in the body of the text. I understand that the statistical analyses are in Table 1, but this table is reasonably cumbersome to navigate because there are so many footnotes, and the directionality of these results would be more straightforward to grasp in the body of the text.

The discussion does not put the results into context. Why do you think you saw the results that you did? What mechanisms could describe your results, what considerations must be considered, how does this improve or change our understanding of carry-over effects? For example, it is concluded that there is a carry-over effect on mortality but no discussion of this mortality. Why do you think that so many of the individuals from the communal treatment die – where they killed by the other beetles, starved, or just dead? What have other studies found for mortality in such group breeding, if any? Why does experience matter – because it changes physiological state, or the beetles are learning, or something else? Why does the experience of the partner in the second brood matter? E.g., why do they spend more time on the carcass if the partner was from a communal environment? Do you think they can tell what their partner's previous experience was, or are they responding to changes in partners behavior or pheromones or something else? There are a lot of interesting results and as a reader, I want to hear the authors take on what they mean.

I am so curious what was going on during the communal breeding events – did the authors observe competition, actual cooperation between them, did the beetles mostly ignore the other pair or were there constant interactions, etc.? I think understanding what happens socially on the communal carcass will be important for understanding what carry-over effects we may expect to see. At minimum, may be an interesting discussion point or future direction.

Aside from the content-based edits to the Introduction and Discussion, this paper needs substantial editing for clarity and copy-editing throughout. Some examples are included in the line-by-line comments below, but I omitted quite a few given how many there were.

Line specific comments:

Line 49: It is not clear why subjecting individuals of varying sizes to communal or non-communal breeding would provide insight into carry over effects – I imagine there are quite good reasons to expect that social experience would differ between large/small beetles, so perhaps a short plug here that puts this into context for non-beetle people – e.g., “the impact of social environment may be expected to affect individuals of varying competitive environment differently . . . we took advantage of the known size-based competitive ability of *N. vespilloides* to test . . .”

Line 62-63: How often do you expect for individuals of this species to have multiple breeding attempts in the wild, given how limited/ephemeral the breeding resource is? If it is reasonably rare – realistically, how important may carry-over effects be for generating variation for selection to act on?

Line 76-77: “to find food”? Sentence reads a bit awkward.

Line 75-92: repetitive use of “cooperative and communal breeding” – may reduce the wordiness to state once and call “group breeding” or just “communal breeding” after first use.

Line 94: It would be helpful to have the definition of carry-over effects at first mention (i.e., previous paragraph). Also, by this definition, it is hard to pinpoint what is important about carryover effects – by the examples provided below, everything could have a carryover effect, and if everything matters, then none of it does. Would we expect for carry-over effects to impact some traits more than others, or have a disproportionate effect on fitness?

Line 94-95: By your definition, isn't a carry-over effect how an individual's experience impacts things down the road? For this reason, it makes the definition circular to say that experience can lead to carry over effects on individual conditions based on experience.

Line 98: "within and across seasons or between reproductive events"

Line 99 – 100: sentence awkward, revise. E.g., instead of "different levels of physical states in terms of energy reserves" – "variation in condition".

Line 114 – 116: Are the authors just stating that life-history tradeoffs are just classic examples of carry-over effects? If so, simplify, it is not clear what is meant by "manifested" here and the sentence reads a bit circular.

Line 118: communal/cooperative breeders compared to "non-group breeding species" here. It would really streamline the introduction to refer to communal/cooperative ones as "group breeding". I recommend saying specifically what you are talking about at first – cooperative/communal – and then say "hereafter group breeding" – until the manuscript focuses down onto communal breeding (otherwise quite repetitive to read "communal and cooperative breeding" every sentence).

Line 141: remove "the" before social dominance.

Line 147: typo after "group"

Line 160: what is the direction in which they adjust parental investment? Will provide context for why you are predicting a directional relationship on line 161.

Line 178-181: How are you extricating the potential effect of differing social environment that occurred pre-breeding from the social environment after breeding initiated? I.e., says that individuals were kept in varying groups sizes prior to the experiment (perhaps just requires a quick rewording).

Line 189-191: revise this sentence to be in the active voice.

Line 191: "and body size as measured".

Line 202: be consistent with how each treatment is named – e.g., communal treatment throughout rather than "double pair breeding system".

Line 204 -205: This is a test of assumption, so it is a result – should go in results.

Line 217-218: mouse characteristics described twice in this section of methods.

Line 227-228: Why do we care about carcass burial? The introduction should provide a clear roadmap of what general factors are needed to answer the question, and then the methods tie each measure directly back to the framework – e.g., "need to measure parental care behavior" – so measured these things that are important to burying beetle care behavior.

Line 248-251: why were the mates switched? This seems to complicate the matter, since individuals were size-matched for the first treatment and then not-size matched for the second – this seems like it may confound any age-specific effects versus

Line 258: "providing care" rather than "spending care"

Line 259: edit for clarity.

Line 281: I am unclear how parental behavior was measured – it sounded like broods were checked briefly twice a day and their activities at that moment in time were noted, and then the proportion of those times they were providing care were used as a proxy for care provided throughout. Where does the "total amount of time" come from, and how does it differ from "parenting investment time", line 290?

Line 299 – 313: I see that the stats are included in Table 1a, but where are the means for these statements? E.g., what was burial time for a communal vs non-communal group? This comment applies to all the methods.

Line 309: Editing for active voice and clarity throughout, simplify the language. E.g., this sentence could read: "Individuals in communal groups spent more time on the carcass than non-communal groups".

Line 314: If it was unknown whether weight change was correlated with variation in parental investment, why was weight change described as a metric to track parental investment earlier in the manuscript? Is this a test of assumption or a result, and if so, how does it fit into the overall framework of the manuscript?

Line 318: What was the source of mortality (if possible to infer)?

Table 1: There are quite a lot of results, but I did not find the table format to be particularly accessible (especially given the number of footnotes needed to describe what the variables mean relative to the statistics). May be more straightforward to include these statistics in the written methods, and then have a summary table of them all somewhere – or at least make a table that makes the comparison/directionality of the results more straightforward.

Line 331: How was a rate of carcass burial determined? And what is the significance of the rate of carcass burial? Need to add to methods.

Line 339: As currently written, there are contradictory statements in this sentence.

Line 343: this sentence should start the paragraph and then go into details.

Line 362: typo - start of sentence "with the regardless"

Line 364: "outcomes" instead of "outcome"

Line 384: Alternatively, you don't know what their options are – e.g., if that one big carcass is literally the only one they can breed on, there is not a cost of breeding on it since they would have nothing if they didn't.

Line 389: Are lighter larvae bad? I assume so, based on context, but to an outside reader, there are tradeoffs in development that could be compensated for later, and there is no reason provided for why we would expect for lighter offspring to be bad off the bat.

Line 420: Do individuals need to gain more breeding experience? Does this species usually breed more than once in nature? Or do you expect that larger individuals are more likely to breed a second time?

Line 433: Didn't the beetles have similar weights after the first breeding attempt? So there wasn't a test of the effect of weight gains on future reproduction? Please clarify.

Line 436 (and discussion more broadly): How are you envisioning that experience is impacting this? It would be useful to hear a discussion of the mechanisms that would produce these results.

Review form: Reviewer 2

Is the manuscript scientifically sound in its present form?

Yes

Are the interpretations and conclusions justified by the results?

Yes

Is the language acceptable?

Yes

Do you have any ethical concerns with this paper?

No

Have you any concerns about statistical analyses in this paper?

No

Recommendation?

Accept with minor revision (please list in comments)

Comments to the Author(s)

Overall this was a good manuscript. I liked the experimental design and the results were intriguing. Some clarifications need to be made in the Materials and Methods section. See the attached (Appendix A) for the specific comments. Do you think that you would find similar results in other invertebrate models?

Decision letter (RSOS-211179.R0)

Dear Dr Ma

The Editors assigned to your paper RSOS-211179 "Sex-specific influence of communal breeding experience on parenting performance and fitness in a burying beetle" have now received comments from reviewers and would like you to revise the paper in accordance with the reviewer comments and any comments from the Editors. Please note this decision does not guarantee eventual acceptance.

Please submit your revised manuscript and required files (see below) no later than 21 days from today's (ie 02-Sep-2021) date. Note: the ScholarOne system will 'lock' if submission of the revision is attempted 21 or more days after the deadline. If you do not think you will be able to meet this deadline please contact the editorial office immediately.

Kind regards,
Royal Society Open Science Editorial Office
Royal Society Open Science

on behalf of Dr Polly Campbell (Associate Editor) and Kevin Padian (Subject Editor)
 openscience@royalsociety.org

Associate Editor Comments to Author (Dr Polly Campbell):

Associate Editor: 1

Comments to the Author:

Both reviewers find value in this study's topic and experimental design and agree that the results are interesting. Reviewer 1 provides a detailed and constructive critique that identifies multiple aspects of the framing and presentation of the study that require revision. I encourage the authors to not only address each of this reviewer's points, but to recognize that many of the issues raised reflect a lack of clarity in how the study is framed, in how some of the methods and results are presented, and in aspects of the discussion of findings.

Additional comments:

L79-86: Please use more standard terminology in this summary of the potential fitness benefits of cooperative/communal breeding. Specifically, kin-directed = indirect fitness benefits; add "reproduce" after "helping their relatives"; the term "non-kin benefits" doesn't really capture the idea that cooperating with non-kin can, in some cases, increase an individual's direct (not immediate) fitness. When discussing fitness gained in a single reproductive event suggest using "short term" rather than "immediate".

I don't think the authors measured reproductive success/short term fitness in the communal breeding groups because paternity wasn't assigned to larvae and I didn't see evidence that they could identify which larvae belonged to which female. If I'm wrong about this please clarify. If not, please remove references to reproductive success and fitness in the context of the first breeding experiment. This includes L150-156, L322-323 and any others.

Associate Editor: 2

Comments to the Author:

(There are no comments.)

Reviewer comments to Author:

Reviewer: 1

Comments to the Author(s)

Letter to Authors:

The social environment that an individual experiences may have profound effects on individual reproductive success within and across breeding attempts. Here, the authors argue that early breeding experience has the potential to impact lifetime fitness due to carry-over effects. The carry-over effects of communal breeding may impact individuals in a sex-specific way. Here, the authors conduct an interesting study to tease apart how experience of communal breeding may impact future breeding attempts.

I think it would strengthen the manuscript to have a clearer unifying framework that provides context for the tests and results and clearly states what a) is known/not know and b) what each measure provides inference into. Why was the experiment set up the way that it was, and why were the factors that you measured selected? E.g., small vs large beetles were used, but it is not explicit in the Introduction what this would be a test of (although it is briefly mentioned on line 200). Why was carcass burial, egg laying period, larval hatching time etc. recorded? I'm sure there are good biological reasons to do so, but what do they tell you specifically? E.g., is a longer egg-

laying period “bad” for some reason? Why was parental weight change investigated as a metric of parental effort? What is known versus not known for this system? For example – on line 235 it states that weight change was used to gauge parental effort, but then on line 314 it makes it sound like it was unknown whether weight change was correlated with variation in parental investment. Is this a test of assumption or a result or something unexpected based on previous research, and if so, how does it fit into the overall framework of the study? Overall, it would provide clarity to the manuscript to state in the Introduction what kinds of things you need to measure to answer the question, e.g., measures of parental effort, care behavior and competitive ability, and then in the Methods, you define how each of these was measured: e.g., parental effort was measured as amount of time spent on carcass, length of egg-laying period (because this is a reliable proxy of xx), etc.

The experiment is set up to distinguish between age-effects and experience effects (communal to non-communal and non-communal to non-communal), but changes in reproductive investment over aging are not discussed. There are good life history reasons to expect that investment may change over an individual’s lifespan, and it needs to be mentioned (at minimum).

Also, for the experiment, why were the mates switched in the second breeding bout (line 248)? The authors mention that paternity assurance may affect behavior, but then are manipulating paternity assurance. Doesn’t this species store sperm to some extent? So that males in the second breeding bout will likely not have paternity assurance? Is there any reason to expect that this would influence behavior or investment in later breeding attempts? This is worth mentioning/discussing somewhere, especially since the authors already state that paternity assurance is expected to impact behavior (e.g., line 477).

While the study design does seem well set-up, the sample sizes are both low and uneven. E.g., for the test of carry-over effects, there ends up being comparisons of treatments with an $n=3$ vs $n=12$ (Figure 5, larval hatching times), but these discrepancies are not described. Who was excluded and why? Why don’t the sample sizes match up for the same treatments across the figures – e.g., on Figure 4a/b there are 21 communal and 13 non-communal females, and only 20 and 14 males, respectively. The individuals were paired, so where does this disparity come from?

How often do you expect for individuals of this species to have multiple breeding attempts in the wild, given how limited/ephemeral the breeding resource is? If it is reasonably rare – realistically, how important may carry-over effects be for generating variation for selection to act on (e.g., line 392, 420)? How big of deal do we expect first breeding experience to be, if it’s almost always the only breeding experience? Is this species appropriate for a test of carry-over effects across breeding bouts? Also, it is perhaps too broad of a stroke to say that breeding communally decreases fitness. If they have no other opportunity to breed besides communally, it’s still a better option than not breeding at all.

I think that the results should be re-written with the means \pm SE written in the body of the text. I understand that the statistical analyses are in Table 1, but this table is reasonably cumbersome to navigate because there are so many footnotes, and the directionality of these results would be more straightforward to grasp in the body of the text.

The discussion does not put the results into context. Why do you think you saw the results that you did? What mechanisms could describe your results, what considerations must be considered, how does this improve or change our understanding of carry-over effects? For example, it is concluded that there is a carry-over effect on mortality but no discussion of this mortality. Why do you think that so many of the individuals from the communal treatment die – where they killed by the other beetles, starved, or just dead? What have other studies found for mortality in such group breeding, if any? Why does experience matter – because it changes physiological

state, or the beetles are learning, or something else? Why does the experience of the partner in the second brood matter? E.g., why do they spend more time on the carcass if the partner was from a communal environment? Do you think they can tell what their partner's previous experience was, or are they responding to changes in partners behavior or pheromones or something else? There are a lot of interesting results and as a reader, I want to hear the authors take on what they mean.

I am so curious what was going on during the communal breeding events – did the authors observe competition, actual cooperation between them, did the beetles mostly ignore the other pair or were there constant interactions, etc.? I think understanding what happens socially on the communal carcass will be important for understanding what carry-over effects we may expect to see. At minimum, may be an interesting discussion point or future direction.

Aside from the content-based edits to the Introduction and Discussion, this paper needs substantial editing for clarity and copy-editing throughout. Some examples are included in the line-by-line comments below, but I omitted quite a few given how many there were.

Line specific comments:

Line 49: It is not clear why subjecting individuals of varying sizes to communal or non-communal breeding would provide insight into carry over effects – I imagine there are quite good reasons to expect that social experience would differ between large/small beetles, so perhaps a short plug here that puts this into context for non-beetle people – e.g., “the impact of social environment may be expected to affect individuals of varying competitive environment differently . . . we took advantage of the known size-based competitive ability of *N. vespilloides* to test . . .”

Line 62-63: How often do you expect for individuals of this species to have multiple breeding attempts in the wild, given how limited/ephemeral the breeding resource is? If it is reasonably rare – realistically, how important may carry-over effects be for generating variation for selection to act on?

Line 76-77: “to find food”? Sentence reads a bit awkward.

Line 75-92: repetitive use of “cooperative and communal breeding” – may reduce the wordiness to state once and call “group breeding” or just “communal breeding” after first use.

Line 94: It would be helpful to have the definition of carry-over effects at first mention (i.e., previous paragraph). Also, by this definition, it is hard to pinpoint what is important about carryover effects – by the examples provided below, everything could have a carryover effect, and if everything matters, then none of it does. Would we expect for carry-over effects to impact some traits more than others, or have a disproportionate effect on fitness?

Line 94-95: By your definition, isn't a carry-over effect how an individual's experience impacts things down the road? For this reason, it makes the definition circular to say that experience can lead to carry over effects on individual conditions based on experience.

Line 98: “within and across seasons or between reproductive events”

Line 99 – 100: sentence awkward, revise. E.g., instead of “different levels of physical states in terms of energy reserves” – “variation in condition”.

Line 114 – 116: Are the authors just stating that life-history tradeoffs are just classic examples of carry-over effects? If so, simplify, it is not clear what is meant by “manifested” here and the sentence reads a bit circular.

Line 118: communal/cooperative breeders compared to “non-group breeding species” here. It would really streamline the introduction to refer to communal/cooperative ones as “group breeding”. I recommend saying specifically what you are talking about at first – cooperative/communal – and then say “hereafter group breeding” – until the manuscript focuses down onto communal breeding (otherwise quite repetitive to read “communal and cooperative breeding” every sentence).

Line 141: remove “the” before social dominance.

Line 147: typo after “group”

Line 160: what is the direction in which they adjust parental investment? Will provide context for why you are predicting a directional relationship on line 161.

Line 178-181: How are you extricating the potential effect of differing social environment that occurred pre-breeding from the social environment after breeding initiated? I.e., says that individuals were kept in varying groups sizes prior to the experiment (perhaps just requires a quick rewording).

Line 189-191: revise this sentence to be in the active voice.

Line 191: "and body size as measured".

Line 202: be consistent with how each treatment is named – e.g., communal treatment throughout rather than "double pair breeding system".

Line 204 -205: This is a test of assumption, so it is a result – should go in results.

Line 217-218: mouse characteristics described twice in this section of methods.

Line 227-228: Why do we care about carcass burial? The introduction should provide a clear roadmap of what general factors are needed to answer the question, and then the methods tie each measure directly back to the framework – e.g., "need to measure parental care behavior" – so measured these things that are important to burying beetle care behavior.

Line 248-251: why were the mates switched? This seems to complicate the matter, since individuals were size-matched for the first treatment and then not-size matched for the second – this seems like it may confound any age-specific effects versus

Line 258: "providing care" rather than "spending care"

Line 259: edit for clarity.

Line 281: I am unclear how parental behavior was measured – it sounded like broods were checked briefly twice a day and their activities at that moment in time were noted, and then the proportion of those times they were providing care were used as a proxy for care provided throughout. Where does the "total amount of time" come from, and how does it differ from "parenting investment time", line 290?

Line 299 – 313: I see that the stats are included in Table 1a, but where are the means for these statements? E.g., what was burial time for a communal vs non-communal group? This comment applies to all the methods.

Line 309: Editing for active voice and clarity throughout, simplify the language. E.g., this sentence could read: "Individuals in communal groups spent more time on the carcass than non-communal groups".

Line 314: If it was unknown whether weight change was correlated with variation in parental investment, why was weight change described as a metric to track parental investment earlier in the manuscript? Is this a test of assumption or a result, and if so, how does it fit into the overall framework of the manuscript?

Line 318: What was the source of mortality (if possible to infer)?

Table 1: There are quite a lot of results, but I did not find the table format to be particularly accessible (especially given the number of footnotes needed to describe what the variables mean relative to the statistics). May be more straightforward to include these statistics in the written methods, and then have a summary table of them all somewhere – or at least make a table that makes the comparison/directionality of the results more straightforward.

Line 331: How was a rate of carcass burial determined? And what is the significance of the rate of carcass burial? Need to add to methods.

Line 339: As currently written, there are contradictory statements in this sentence.

Line 343: this sentence should start the paragraph and then go into details.

Line 362: typo - start of sentence "with the regardless"

Line 364: "outcomes" instead of "outcome"

Line 384: Alternatively, you don't know what their options are – e.g., if that one big carcass is literally the only one they can breed on, there is not a cost of breeding on it since they would have nothing if they didn't.

Line 389: Are lighter larvae bad? I assume so, based on context, but to an outside reader, there are tradeoffs in development that could be compensated for later, and there is no reason provided for why we would expect for lighter offspring to be bad off the bat.

Line 420: Do individuals need to gain more breeding experience? Does this species usually breed more than once in nature? Or do you expect that larger individuals are more likely to breed a second time?

Line 433: Didn't the beetles have similar weights after the first breeding attempt? So there wasn't a test of the effect of weight gains on future reproduction? Please clarify.

Line 436 (and discussion more broadly): How are you envisioning that experience is impacting this? It would be useful to hear a discussion of the mechanisms that would produce these results.

Reviewer: 2

Comments to the Author(s)

Overall this was a good manuscript. I liked the experimental design and the results were intriguing. Some clarifications need to be made in the Materials and Methods section. See the attached for the specific comments. Do you think that you would find similar results in other invertebrate models?

===PREPARING YOUR MANUSCRIPT===

===PREPARING YOUR REVISION IN SCHOLARONE===

To revise your manuscript, log into <https://mc.manuscriptcentral.com/rsos> and enter your Author Centre - this may be accessed by clicking on "Author" in the dark toolbar at the top of the

page (just below the journal name). You will find your manuscript listed under "Manuscripts with Decisions". Under "Actions", click on "Create a Revision".

Author's Response to Decision Letter for (RSOS-211179.R0)

See Appendix B.

RSOS-211179.R1 (Revision)

Review form: Reviewer 1

Is the manuscript scientifically sound in its present form?

Yes

Are the interpretations and conclusions justified by the results?

No

Is the language acceptable?

No

Do you have any ethical concerns with this paper?

No

Have you any concerns about statistical analyses in this paper?

Yes

Recommendation?

Major revision is needed (please make suggestions in comments)

Comments to the Author(s)

The authors have completed a very thorough revision of the manuscript based on reviewer feedback, and many items are much clearer.

My main comment on the manuscript at this stage is that extensive editing for sentence clarity, manuscript organization, grammar, and other general issues still needs to be done. Please see the line-by-line comments for examples – this is not exhaustive; I am mainly indicating a number of specific locations that need work or are confusing for the reader.

Terminology is inconsistent and confusing throughout the paper. I strongly encourage the authors to pick a specific term and to stick with it. E.g., “reproductive performance”, “reproductive outcome”, “parental performance”, “parental care”, “parental investment” – these all invoke slightly different messages and I can’t quite figure out where they do and do not cross over (sometimes used as synonyms, sometimes as different measures, etc.). For example, “higher reproductive outcome” is used, but technically a reproductive outcome could be complete failure, so unclear how that could be higher. Also, some locations where word choice should be updated to the common term; e.g., “reproductive restraint” is more commonly described as “reproductive constraint”, etc.

I am still concerned about the sample sizes and the number of fixed effects and interactions used in the statistical analyses. For carry-over effects in particular – the sample sizes are quite small and the models have a lot of variables in there, I think the risk of overparameterizing the model is quite high – small vs large, communal versus non-communal, males versus females etc., etc.

Along the same lines, on Figure 5a., why does the sample size of $n=3$ not have an error bar? And why does it look like that time = 0? Same question for 5b. These look more like artifacts of small sample size rather than biologically-relevant outcomes for communal breeding.

The authors need to tone down their interpretations of the results. Two main examples. First, individual reproductive success in communal set-ups was not measured – so comparing reproductive measures for individuals that were in a communal vs non-communal setting does not give us insight into the short-term or carry-over effects for individuals (line 448-452). For this reason, need to be careful discussing results. Second, you are conducting this experiment in the most benign conditions possible, and yet there is incredibly high mortality in your beetles (over 50%; based on the numbers provided on line 423). Beetles have high mortality by second breeding attempt in benign conditions, carcasses are rare to some extent (e.g., line 472), so seems unlikely that that many beetles actually end up breeding more than once. For this reason, it seems unlikely that selection has, for example, honed specific pheromone cocktail that indicates that a partner bred communally in a previous attempt (line 566). So many factors changed between the first and second attempt (size, age, breeding experience, feeding prior to experimentation, paternity assurance, etc) that, when paired with the low sample size, indicates that there are likely multiple explanations for the patterns the authors observed, and not all of them are based on adaptation. This can be fixed by more cautious wording.

This is a paper about the impacts of communal behavior and yet no data about the communal behavior are presented. How often were more than two beetles on the carcass? What proportion of the time? What kind of things were they seen doing – i.e., how often were they fighting? Working together? This would be very useful information to inform your discussion of sources of mortality in particular. The mechanism of mortality is still presented as a hypothesis (line 486), where you are stating what it might be due to and citing references. What did you observe? Were they ripped apart? Did they have injuries? Were they whole but just dead?

It would be helpful for the reader to have the samples sizes included in the results section, with their respective statistics (e.g., after means and standard errors).

Line-specific comments:

Line 99-101: Edit sentence for grammar.

Line 104-106: “Special” carry-over effect- if carry-over effects are just the effect of previous experience, then why specify a “special” carry-over effect?

Line 108-109: talking from viewpoint of dominant? Revise: “allocation of resources by breeding individuals can be adjusted by their relative position in a dominance hierarchy” or some such.

Line 113: This sentence needs to be edited. Says that “impacts . . . reproductive success . . . through difference in . . . reproduction”. Suggestion: “as such, breeding in groups may impact reproductive success not only through short-term effects, but also through carry-over effects on future reproductive allocation.”

Line 131: “large individuals become dominants can” – edit

Line 139: “individual’s breeding opportunity” – carcass is plural

Line 141: remove comma after “reproduction”

Line 140-144: shorten sentence. E.g. “While breeding communally may decrease reproductive

Line 146-149: edit sentence “breeding communally among individuals”. Also – this is hypothesis, state as the hypothesis.

Line 149-153: how do short-term benefits on fitness act on selection? Fitness is a measure of selection; it does not “act on” selection. Re-word.

Line 161: use consistent terms throughout – “reproductive outcome”? Reproductive success is a common term for this.

Line 165: “each pairs” edit “each pair”. But didn’t you switch individuals between pairs? So the pair was not a consistent unit – this is not quite accurate of a description here then.

Line 173: “parental efforts” - “parental effort”

Line 166-175: You state here that “individuals have been shown to reduce their efforts [in a communal setting]” but then on line 154-155 you are saying one of the goals of your project is to see whether they adjust parental care behavior in a communal saying. This is contradictory – edit accordingly.

Line 174: what are “parental resources”? Use consistent terminology.

Line 187 – 190: need to edit. “All beetles were reared in small groups, and thus the social environment in early life was similar for all beetles” or something like that

Line 195: also throughout – is “parental performance” the same as “parental care”? I’d say that parental care is part of reproduction as well. Solution - could say parental care and reproductive success and it would be immediately obvious to the reader.

Line 196-198: why two names for the treatments? Just call “communal” (define as two males and two females) and “non-communal” (one male, one female).

Line 206-207: Need to edit this sentence, also provide directionality: e.g., “In burying beetles, adult body size determines competitive ability, such that larger individuals are more likely to monopolize a carcass and have higher reproductive success.”

Line 210: “is” instead of “was” – or rephrase

Line 211-214: How does this sentence differ from the previous – i.e., small pair + large pair = stable dominance hierarchy. Remove one. Also “was accordance”

Line 214-216: Provide directionality. I.e., Larger individuals invest more in brood? Less?

Line 216-218: Selected one large pair and one small pair for non-communal? Makes it sound like you only have two replicates of non-communal.

Line 218-219: Sentence awkward, make it into active voice. E.g., Individuals were weighed immediately prior to onset of experiment.

Line 220: “communal breeding is more likely to take place on carcasses larger than 25 g”

Line 224: 9 out of 10 females produced? Are you saying that 9 out of 10 small females produced offspring in a communal attempt, and only 5 out of 10 males? This makes it seem a bit odd to measure group outcome, especially for males.

Line 229: Need to edit sentence. E.g., “At the onset of each treatment, beetles were placed in a breeding box with a thawed mouse.”

Line 238: should this “and” be “and/or”? Larva provisioning occurs inside of the carcass rather than on surface?

Line 244: “joint parental performance” – what does this mean? As a proxy for cooperation perhaps? State why this is a good proxy.

Line 246: Start new paragraph. It may benefit the methods section overall to have subheadings that describe what measures are. E.g., “Communal versus non-communal breeding”, then “Measurement of parental care”, “Reproductive success” – or whichever you think best. Right now, the methods seem to fluctuate between discussion (e.g., line 248 “resulted in selection . . .”) and the actual instructions for carrying out experiment.

Line 250 – 254: Edit sentence – three variations of “reproduction” in same sentence – reproduction, reproductive strategies, and reproductive performance – are they all different? Are they all the same?

Line 261-263: Why was weight change used as a proxy for parental effort instead of the parental behavior that you monitored? Were both used in conjunction?

Line 265: “were not fed prior to subsequent experimentation”.

Line 280-288: This reads more like introduction or discussion. Also, you are probably okay with paternity assurance - individuals were switched around (i.e., all individuals experienced same treatment) so paternity assurance will be equally low across treatments. This could be a discussion point but not appropriate here.

Line 294 and line 317: says “reproductive and developmental timing” on line 294, and then the developmental timing is lumped into reproductive timing for the analyses on line 316.

Line 325-328: Carcass burial was analyzed with time, previous experience, size, sex, and interactions as fixed factors? Given the sample size, seems like too many variables.

Line 329-335: this is not appropriate in the "Statistical analyses" section. Also- you are testing whether there are carry-over effects, rephrase so that it is not implying that you already know there are carry-over effects.

Line 333-336: Sentence confusing, rephrase.

Line 351: "the access of carcass"

Line 365-368: how can total investment in care be different but parental behavior not be different? Care implies parental behavior. This sentence is also an interpretation – save interpretations for the discussion.

Line 381: does egg-laying period = the amount of time to finish laying eggs? Onset of egg-laying? How is this different from the egg laying time (reported in previous sentence)? Or time it took for the eggs to hatch?

Line 407: Need to remove "specifically" – this makes it sound like you are providing specific results to a general one in the previous sentence, when in fact you are providing different results.

Line 407-409: This sentence is confusing – rephrase for clarity.

Line 423: which one is males and which is females. Did these all die during/right after communal breeding, or just at some point in experiment? What are sources of mortality?

Line 448-452: There was no individual level – you measured reproductive success for the whole group, so you don't know whether there were short-term effects on parental investment and reproductive success.

Line 467: "reproduce a majority of offspring"? Rephrase.

Line 469-474: seems to contradict conclusions of lines 455-458.

Line 487-489: you are contrasting reproduction and reproduction – first half and second half of sentence seem to be re-writes of each other. Rephrase.

Line 492: "deteriorate some opportunistic infections"? Implies the infection gets better because of microbe-rich soil. Rephrase.

Line 494 and 495 – these two sentences are repeating each other.

Line 497-500: unclear and no literature cited for immunity?

Line 509: "save more resources" – did they? You have measurements of weight change, state that here if you found this.

Line 523: This sentence contradicts previous paragraph.

Line 537-539: wait so we already know about carry-over effects in communal breeders? Describe in intro.

Line 589: edit

Line 594: within carcasses?

Line 605: This sentence is confusing

Line 614: I thought that they didn't "save resources", according to your results?

Line 630-634: Both of the future directions seem to be the same project...?

Decision letter (RSOS-211179.R1)

Dear Dr Ma

The Editors assigned to your paper RSOS-211179.R1 "Sex-specific influence of communal breeding experience on parenting performance and fitness in a burying beetle" have now received comments from reviewers and would like you to revise the paper in accordance with the

reviewer comments and any comments from the Editors. Please note this decision does not guarantee eventual acceptance.

Please submit your revised manuscript and required files (see below) no later than 21 days from today's (ie 22-Nov-2021) date. Note: the ScholarOne system will 'lock' if submission of the revision is attempted 21 or more days after the deadline. If you do not think you will be able to meet this deadline please contact the editorial office immediately.

on behalf of Dr Polly Campbell (Associate Editor) and Kevin Padian (Subject Editor)
openscience@royalsociety.org

Associate Editor Comments to Author (Dr Polly Campbell):

Associate Editor: 1

Comments to the Author:

The authors have done a great deal to improve the clarity of this manuscript and their scholarly approach in responses to reviewer critiques is very much appreciated. However, there are still many places where sentence structure and/or the flow of ideas between sentences is very hard to follow. The reviewer provides detailed examples of these issues and suggestions on how to fix them, and I provide additional non-overlapping comments below. Could the authors please check that they addressed Reviewer 1's line-by-line comments from the first submission? If not, please address those along with the current comments. The authors should also consider having a fluent English speaker check the manuscript for clarity before resubmission.

The reviewer's remaining concerns about the statistical analyses and interpretation of results are also important; please give these careful consideration.

Specific comments

Fitness: there are still multiple references to measuring fitness in the Introduction (L150-175). Referring to short-term fitness implications (L156) doesn't address the issue that individual reproductive output couldn't be measured in the communal breeding treatment, it just makes things a bit fuzzy. The measures of "parental performance" are indices of parental investment. This is fine but these measures are not a proxy for short-term fitness. Please go through the manuscript using the find function in MS Word and for every use of "fitness" in reference to the experiments consider whether another more specific and accurate term could substitute.

L221-227 I appreciate that this sentence was added in response to a comment but it is confusingly written and is out of place in Methods. Is the point to provide confirmation that double-pair treatments induce communal breeding or to provide evidence that small individuals' reproduction suffers in communal breeding? Either way, the numbers in parentheses are not necessary and the authors might consider integrating the result into the Discussion.

L233-236 "Each check lasted for 30 seconds so as to make sure that beetles were providing care or not when they were present or absent on the carcass, which could also avoid the counts of any accidental events, for example, beetles were wandering on the carcass without providing care." This sentence is very hard to follow. Please revise.

L407-414 This sentence is very hard to follow and seems to contradict the previous sentence (404-407).

L513 not clear what "increase individual conditions" means

L516-519 not clear what the last clause of this sentence means ("which may be due to reproductive restraints [constraints?] in females.") and how it relates to the first part.

L553-554 Not clear what "the presence of males does not seem to buffer this negative effect..." refers to. Males are present in all treatments...

Reviewer comments to Author:

Reviewer: 1

Comments to the Author(s)

The authors have completed a very thorough revision of the manuscript based on reviewer feedback, and many items are much clearer.

My main comment on the manuscript at this stage is that extensive editing for sentence clarity, manuscript organization, grammar, and other general issues still needs to be done. Please see the line-by-line comments for examples – this is not exhaustive; I am mainly indicating a number of specific locations that need work or are confusing for the reader.

Terminology is inconsistent and confusing throughout the paper. I strongly encourage the authors to pick a specific term and to stick with it. E.g., "reproductive performance", "reproductive outcome", "parental performance", "parental care", "parental investment" – these all invoke slightly different messages and I can't quite figure out where they do and do not cross over (sometimes used as synonyms, sometimes as different measures, etc.). For example, "higher reproductive outcome" is used, but technically a reproductive outcome could be complete failure, so unclear how that could be higher. Also, some locations where word choice should be updated to the common term; e.g., "reproductive restraint" is more commonly described as "reproductive constraint", etc.

I am still concerned about the sample sizes and the number of fixed effects and interactions used in the statistical analyses. For carry-over effects in particular – the sample sizes are quite small and the models have a lot of variables in there, I think the risk of overparameterizing the model is quite high – small vs large, communal versus non-communal, males versus females etc., etc. Along the same lines, on Figure 5a., why does the sample size of $n=3$ not have an error bar? And why does it look like that time = 0? Same question for 5b. These look more like artifacts of small sample size rather than biologically-relevant outcomes for communal breeding.

The authors need to tone down their interpretations of the results. Two main examples. First, individual reproductive success in communal set-ups was not measured – so comparing reproductive measures for individuals that were in a communal vs non-communal setting does not give us insight into the short-term or carry-over effects for individuals (line 448-452). For this reason, need to be careful discussing results. Second, you are conducting this experiment in the most benign conditions possible, and yet there is incredibly high mortality in your beetles (over 50%; based on the numbers provided on line 423). Beetles have high mortality by second breeding attempt in benign conditions, carcasses are rare to some extent (e.g., line 472), so seems unlikely that that many beetles actually end up breeding more than once. For this reason, it seems unlikely that selection has, for example, honed specific pheromone cocktail that indicates that a partner bred communally in a previous attempt (line 566). So many factors changed between the first and second attempt (size, age, breeding experience, feeding prior to experimentation, paternity assurance, etc) that, when paired with the low sample size, indicates that there are likely multiple explanations for the patterns the authors observed, and not all of them are based on adaptation. This can be fixed by more cautious wording.

This is a paper about the impacts of communal behavior and yet no data about the communal behavior are presented. How often were more than two beetles on the carcass? What proportion of the time? What kind of things were they seen doing – i.e., how often were they fighting? Working together? This would be very useful information to inform your discussion of sources of mortality in particular. The mechanism of mortality is still presented as a hypothesis (line 486), where you are stating what it might be due to and citing references. What did you observe? Were they ripped apart? Did they have injuries? Were they whole but just dead?

It would be helpful for the reader to have the samples sizes included in the results section, with their respective statistics (e.g., after means and standard errors).

Line-specific comments:

Line 99-101: Edit sentence for grammar.

Line 104-106: “Special” carry-over effect- if carry-over effects are just the effect of previous experience, then why specify a “special” carry-over effect?

Line 108-109: talking from viewpoint of dominant? Revise: “allocation of resources by breeding individuals can be adjusted by their relative position in a dominance hierarchy” or some such.

Line 113: This sentence needs to be edited. Says that “impacts . . . reproductive success . . . through difference in . . . reproduction”. Suggestion: “as such, breeding in groups may impact reproductive success not only through short-term effects, but also through carry-over effects on future reproductive allocation.”

Line 131: “large individuals become dominants can” – edit

Line 139: “individual’s breeding opportunity” – carcass is plural

Line 141: remove comma after “reproduction”

Line 140-144: shorten sentence. E.g. “While breeding communally may decrease reproductive

Line 146-149: edit sentence “breeding communally among individuals”. Also – this is hypothesis, state as the hypothesis.

Line 149-153: how do short-term benefits on fitness act on selection? Fitness is a measure of selection; it does not “act on” selection. Re-word.

Line 161: use consistent terms throughout – “reproductive outcome”? Reproductive success is a common term for this.

Line 165: “each pairs” edit “each pair”. But didn’t you switch individuals between pairs? So the pair was not a consistent unit – this is not quite accurate of a description here then.

Line 173: “parental efforts” - “parental effort”

Line 166-175: You state here that “individuals have been shown to reduce their efforts [in a communal setting]” but then on line 154-155 you are saying one of the goals of your project is to see whether they adjust parental care behavior in a communal saying. This is contradictory – edit accordingly.

Line 174: what are “parental resources”? Use consistent terminology.

Line 187 – 190: need to edit. “All beetles were reared in small groups, and thus the social environment in early life was similar for all beetles” or something like that

Line 195: also throughout – is “parental performance” the same as “parental care”? I’d say that parental care is part of reproduction as well. Solution - could say parental care and reproductive success and it would be immediately obvious to the reader.

Line 196-198: why two names for the treatments? Just call “communal” (define as two males and two females) and “non-communal” (one male, one female).

Line 206-207: Need to edit this sentence, also provide directionality: e.g., “In burying beetles, adult body size determines competitive ability, such that larger individuals are more likely to monopolize a carcass and have higher reproductive success.”

Line 210: “is” instead of “was” – or rephrase

Line 211-214: How does this sentence differ from the previous – i.e., small pair + large pair = stable dominance hierarchy. Remove one. Also “was accordance”

Line 214-216: Provide directionality. I.e., Larger individuals invest more in brood? Less?

Line 216-218: Selected one large pair and one small pair for non-communal? Makes it sound like you only have two replicates of non-communal.

Line 218-219: Sentence awkward, make it into active voice. E.g., Individuals were weighed immediately prior to onset of experiment.

Line 220: “communal breeding is more likely to take place on carcasses larger than 25 g”

Line 224: 9 out of 10 females produced? Are you saying that 9 out of 10 small females produced offspring in a communal attempt, and only 5 out of 10 males? This makes it seem a bit odd to measure group outcome, especially for males.

Line 229: Need to edit sentence. E.g., “At the onset of each treatment, beetles were placed in a breeding box with a thawed mouse.”

Line 238: should this “and” be “and/or”? Larva provisioning occurs inside of the carcass rather than on surface?

Line 244: “joint parental performance” – what does this mean? As a proxy for cooperation perhaps? State why this is a good proxy.

Line 246: Start new paragraph. It may benefit the methods section overall to have subheadings that describe what measures are. E.g., “Communal versus non-communal breeding”, then “Measurement of parental care”, “Reproductive success” – or whichever you think best. Right now, the methods seem to fluctuate between discussion (e.g., line 248 “resulted in selection . . .”) and the actual instructions for carrying out experiment.

Line 250 – 254: Edit sentence – three variations of “reproduction” in same sentence – reproduction, reproductive strategies, and reproductive performance – are they all different? Are they all the same?

Line 261-263: Why was weight change used as a proxy for parental effort instead of the parental behavior that you monitored? Were both used in conjunction?

Line 265: “were not fed prior to subsequent experimentation”.

Line 280-288: This reads more like introduction or discussion. Also, you are probably okay with paternity assurance - individuals were switched around (i.e., all individuals experienced same treatment) so paternity assurance will be equally low across treatments. This could be a discussion point but not appropriate here.

Line 294 and line 317: says “reproductive and developmental timing” on line 294, and then the developmental timing is lumped into reproductive timing for the analyses on line 316.

Line 325-328: Carcass burial was analyzed with time, previous experience, size, sex, and interactions as fixed factors? Given the sample size, seems like too many variables.

Line 329-335: this is not appropriate in the “Statistical analyses” section. Also- you are testing whether there are carry-over effects, rephrase so that it is not implying that you already know there are carry-over effects.

Line 333-336: Sentence confusing, rephrase.

Line 351: “the access of carcass”

Line 365-368: how can total investment in care be different but parental behavior not be different? Care implies parental behavior. This sentence is also an interpretation – save interpretations for the discussion.

Line 381: does egg-laying period = the amount of time to finish laying eggs? Onset of egg-laying? How is this different from the egg laying time (reported in previous sentence)? Or time it took for the eggs to hatch?

Line 407: Need to remove “specifically” – this makes it sound like you are providing specific results to a general one in the previous sentence, when in fact you are providing different results.

Line 407-409: This sentence is confusing – rephrase for clarity.

Line 423: which one is males and which is females. Did these all die during/right after communal breeding, or just at some point in experiment? What are sources of mortality?

Line 448-452: There was no individual level – you measured reproductive success for the whole group, so you don’t know whether there were short-term effects on parental investment and reproductive success.

Line 467: “reproduce a majority of offspring”? Rephrase.

Line 469-474: seems to contradict conclusions of lines 455-458.

Line 487-489: you are contrasting reproduction and reproduction – first half and second half of sentence seem to be re-writes of each other. Rephrase.

Line 492: “deteriorate some opportunistic infections”? Implies the infection gets better because of microbe-rich soil. Rephrase.

Line 494 and 495 – these two sentences are repeating each other.

Line 497-500: unclear and no literature cited for immunity?

Line 509: “save more resources” – did they? You have measurements of weight change, state that here if you found this.

Line 523: This sentence contradicts previous paragraph.

Line 537-539: wait so we already know about carry-over effects in communal breeders? Describe in intro.

Line 589: edit

Line 594: within carcasses?

Line 605: This sentence is confusing

Line 614: I thought that they didn’t “save resources”, according to your results?

Line 630-634: Both of the future directions seem to be the same project...?

===PREPARING YOUR MANUSCRIPT===

Please ensure that you include an acknowledgements' section before your reference list/bibliography. This should acknowledge anyone who assisted with your work, but does not

qualify as an author per the guidelines at <https://royalsociety.org/journals/ethics-policies/openness/>.

If you have been asked to revise the written English in your submission as a condition of publication, you must do so, and you are expected to provide evidence that you have received language editing support. The journal would prefer that you use a professional language editing service and provide a certificate of editing, but a signed letter from a colleague who is a fluent speaker of English is acceptable. Note the journal has arranged a number of discounts for authors using professional language editing services (<https://royalsociety.org/journals/authors/benefits/language-editing/>).

===PREPARING YOUR REVISION IN SCHOLARONE===

- Ensure that your data access statement meets the requirements at <https://royalsociety.org/journals/authors/author-guidelines/#data>. You should ensure that you cite the dataset in your reference list. If you have deposited data etc in the Dryad repository, please include both the 'For publication' link and 'For review' link at this stage.
- If you are requesting an article processing charge waiver, you must select the relevant waiver option (if requesting a discretionary waiver, the form should have been uploaded at Step 3 'File upload' above).
- If you have uploaded ESM files, please ensure you follow the guidance at <https://royalsociety.org/journals/authors/author-guidelines/#supplementary-material> to include a suitable title and informative caption. An example of appropriate titling and captioning may be found at https://figshare.com/articles/Table_S2_from_Is_there_a_trade-off_between_peak_performance_and_performance_breadth_across_temperatures_for_aerobic_scops_in_teleost_fishes_/3843624.

Author's Response to Decision Letter for (RSOS-211179.R1)

See Appendix C.

Decision letter (RSOS-211179.R2)

Dear Dr Ma

On behalf of the Editors, we are pleased to inform you that your Manuscript RSOS-211179.R2 "Sex-specific influence of communal breeding experience on parenting performance and fitness in a burying beetle" has been accepted for publication in Royal Society Open Science subject to minor revision in accordance with the referees' reports. Please find the referees' comments along with any feedback from the Editors below my signature.

Please also note that we require an active email address from all authors that is able to receive messages from the journal. At present 'm.hammers@rug.nl' is unable to receive correspondence - please can you ensure your colleague's email address is amended or communicated to the editorial office with your revised manuscript?

Please submit your revised manuscript and required files (see below) no later than 7 days from today's (ie 11-Jan-2022) date. Note: the ScholarOne system will 'lock' if submission of the revision is attempted 7 or more days after the deadline. If you do not think you will be able to meet this deadline please contact the editorial office immediately.

on behalf of Dr Polly Campbell (Associate Editor) and Kevin Padian (Subject Editor)
openscience@royalsociety.org

Associate Editor Comments to Author (Dr Polly Campbell):

Associate Editor

Comments to the Author:

The authors have, again, done a commendably thorough and thoughtful job on this second revision. Both readability and accurate use of terminology are greatly improved. Please resolve the following minor issues:

L39 Please change that to and

L98-100 Please include common name of animal in cited study: "For example, in _____, experienced individuals" etc.

L148-150 "It is also important to examine whether and how carry-over effects, concurrent with short-term effects, act on the selection for group breeding and its adaptation to different environments." Please revise for clarity. As written, it sounds like these effects act on selection that favors group breeding. This doesn't make any sense and it's unclear what adaptation to different environments refers to.

L159-160 "We predict that communal groups achieve higher reproductive success compared to non-communal groups." I know the authors have done a lot to modify their terminology in relation to fitness and that reproductive success was a reviewer suggestion. But reproductive success IS fitness and it doesn't make sense to talk about it as a single metric for a group of unrelated individuals. It would be fine to predict that brood size will be larger in communal relative to non-communal groups.

L250-255 The use of performance is a bit confusing. Suggest shortening to: "Therefore, we examined the timing of egg-laying and larval-hatching in communal and non-communal groups. Egg-laying and larval hatching times were recorded as the time from the start of the experiment until the onset of laying and hatching, respectively."

L259 Please change success to output.

L285-288 This is a confusing sentence. I think the intended meaning is that all males in the second experiment experience some paternity uncertainty because females mated in the first experiment but this shouldn't affect inferences about carry-over effects from different prior breeding

experiences because 1) mating occurred in both pair and communal breeding groups, and 2) males dilute competitors' sperm by copulating a lot. Please try to get this across more clearly or remove from Methods as suggested by the reviewer and combine with the added text in Discussion (L516). Note that ALL females will have mated in the first experiment, not just the ones in the communal treatment as suggested on L519.

L332-334 "To further test whether the effects of previous breeding experience (communal versus non-communal breeding) on an individual's parental performance are simply by-effects of the previous parental efforts" I think the authors over-interpreted what the reviewer was asking them to do. Please replace with the text from the prior version ("To further test whether the carry-over effects of communal and non-communal breeding on an individual's parental performance in the future breeding events are manifested through the direct effect of the previous parental efforts") and add potential before carry-over.

L372-373 Please just report that there was no effect of breeding treatment on brood size (and any other variable that reproductive success refers to).

L448-451 For the reasons mentioned above, it doesn't make sense to talk about fitness/reproductive success at the group level and here, without knowing individual reproductive output for communal breeders in the first experiment it doesn't make sense to talk about that either. Please modify as follows: "We found that communal breeding appeared to have no short-term effects of parental investment. However, we found..." etc. "Our results indicated that communally breeding groups had improved carcass burial performance and increased total time spent on parental behavior, whereas..." etc.

L488-489 Grammar. Suggest changing incur high to incurring higher.

Reviewer comments to Author:

===PREPARING YOUR MANUSCRIPT===

one version should clearly identify all the changes that have been made (for instance, in coloured highlight, in bold text, or tracked changes);

If you have been asked to revise the written English in your submission as a condition of publication, you must do so, and you are expected to provide evidence that you have received language editing support. The journal would prefer that you use a professional language editing

service and provide a certificate of editing, but a signed letter from a colleague who is a proficient user of English is acceptable. Note the journal has arranged a number of discounts for authors using professional language editing services (<https://royalsociety.org/journals/authors/benefits/language-editing/>).

===PREPARING YOUR REVISION IN SCHOLARONE===

-- If you are requesting an article processing charge waiver, you must select the relevant waiver option (if requesting a discretionary waiver, the form should have been uploaded, see 'File upload' above).

-- If you have uploaded any electronic supplementary (ESM) files, please ensure you follow the guidance at <https://royalsociety.org/journals/authors/author-guidelines/#supplementary-material> to include a suitable title and informative caption. An example of appropriate titling and captioning may be found at https://figshare.com/articles/Table_S2_from_Is_there_a_trade-off_between_peak_performance_and_performance_breadth_across_temperatures_for_aerobic_scope_in_teleost_fishes_/3843624.

Author's Response to Decision Letter for (RSOS-211179.R2)

See Appendix D.

Decision letter (RSOS-211179.R3)

Dear Dr Ma,

I am pleased to inform you that your manuscript entitled "Sex-specific influence of communal breeding experience on parenting performance and fitness in a burying beetle" is now accepted for publication in Royal Society Open Science.

Please see the Royal Society Publishing guidance on how you may share your accepted author manuscript at <https://royalsociety.org/journals/ethics-policies/media-embargo/>. After publication, some additional ways to effectively promote your article can also be found here

<https://royalsociety.org/blog/2020/07/promoting-your-latest-paper-and-tracking-your-results/>.

on behalf of Dr Polly Campbell (Associate Editor) and Kevin Padian (Subject Editor)
openscience@royalsociety.org

Appendix A

Page 7 lines 191-196: I was a little confused, were the beetles bred first before being put in experiment 1? So, they were paired or grouped up after insemination. Why?

Page 8 top paragraph: There was a breeding period that differed from the brooding period and the beetles bred with different partners that were not their brooding partners? You need to clarify this part a bit. Also, you need to justify why this was done.

Page 8 line 221: For how long did you make observations? How many days or weeks?

Page 8 lines 228-235: Did you measure offspring mortality? If you did not, why not?

Page 9, Exp. 2: What was the time delay between experiments? What was the beetle husbandry between experiments? In the results, you mention they all pretty much lost weight. Did you try to increase their weight between the experiments?

In the results: Be consistent with reporting the statistical results verses just listing the table where the results can be found.

Page 15 line 440: Be consistent with citations. Need to change this citation to numbers.

On the graphs: Use a darker grey for the non-communal lines and info. When I printed out the sheets, the lines did not show up. They were also very faint on the electronic version.

Figure S3: Add brood size, body weight, and ave. larval weight as titles to the 3 graphs.

Appendix B

October 7, 2021

Dear Editors, Dr. Polly Campbell and Dr. Kevin Padian,

We would like to thank you and the reviewers for the constructive reviews of our manuscript entitled "*Sex-specific influence of communal breeding experience on parenting performance and fitness in a burying beetle*", and for the opportunity to resubmit a revised version. We have carefully read all comments, replied to each of them below and revised our manuscript accordingly. We very much hope that our amendments, in the light of the reviewers' comments, have improved the manuscript. We sincerely hope that you will find the current version acceptable for publication in *Royal Society Open Science*. Thank you very much for reconsidering our manuscript.

Yours sincerely,
Long Ma, on behalf of all authors

Dear Dr Ma

The Editors assigned to your paper RSOS-211179 "Sex-specific influence of communal breeding experience on parenting performance and fitness in a burying beetle" have now received comments from reviewers and would like you to revise the paper in accordance with the reviewer comments and any comments from the Editors. Please note this decision does not guarantee eventual acceptance.

Please submit your revised manuscript and required files (see below) no later than 21 days from today's (ie 02-Sep-2021) date. Note: the ScholarOne system will 'lock' if submission of the revision is attempted 21 or more days after the deadline. If you do not think you will be able to meet this deadline please contact the editorial office immediately.

on behalf of Dr Polly Campbell (Associate Editor) and Kevin Padian (Subject Editor)

Associate Editor Comments to Author (Dr Polly Campbell):

Associate Editor: 1

Comments to the Author:

Both reviewers find value in this study's topic and experimental design and agree that the results are interesting. Reviewer 1 provides a detailed and constructive critique that identifies multiple aspects of the framing and presentation of the study that require revision. I encourage the authors to not only address each of this reviewer's points, but to recognize that many of the issues raised reflect a lack of clarity in how the study is framed, in how some of the methods and results are presented, and in aspects of the discussion of findings.

Thank you very much for these comments. We agree that multiple aspects of the framing and presentation of our study required revision. Below (see our response to reviewer 1), we explain in detail how we addressed the constructive critique of reviewer 1 in our revised manuscript.

Additional comments:

L79-86: Please use more standard terminology in this summary of the potential fitness benefits of cooperative/communal breeding. Specifically, kin-directed = indirect fitness benefits; add "reproduce" after "helping their relatives"; the term "non-kin benefits" doesn't really capture the idea that cooperating with non-kin can, in some cases, increase an individual's direct (not immediate) fitness. When discussing fitness gained in a single reproductive event suggest using "short term" rather than "immediate".

Thank you very much for the comment, and we fully agree. We have now revised the terminology in the revised main manuscript (Line 65-75):

"More advanced forms of social groups may occur among conspecifics, for example in cooperatively and communally breeding systems (hereafter group breeding), where multiple individuals live and reproduce together to rear a single brood or litter [3–5]. In such breeding groups, group members may gain kin-selected benefits (i.e. the indirect fitness benefits individuals gain through helping their relatives reproduce) and various aspects of direct fitness benefits (e.g. non-kin reciprocity and mutualism) from group living [1,6–8], although they may or may not benefit in terms of direct reproductive success [9–12]. A large number of studies have focused on the short-term fitness benefits of kin-selected and direct benefits of group breeding for each individual involved, such as the joint defence of territories and resources against intruders [8,13–15], reduced workloads during parental care [16–20], higher reproductive success [2,21,22], and higher survival and longevity [23]."

I don't think the authors measured reproductive success/short term fitness in the communal breeding groups because paternity wasn't assigned to larvae and I didn't see evidence that they could identify which larvae belonged to which female. If I'm wrong about this please clarify. If not, please remove references to reproductive success and fitness in the context of the first breeding experiment. This includes L150-156, L322-323 and any others.

Thank you very much for your comments. We agree with you that we did not measure reproductive success/short-term fitness for individuals in communal breeding because parentage was not assigned to each larvae. Instead, we measured the reproductive outcome at the level of group (communal vs non-communal groups; group productivity)(Clutton-Brock, 2009; Eggert et al., 2008; Komdeur et al., 2013; Shen et al., 2017; Richardson and Smiseth, 2020).

We have now clarified this measure in our revised manuscript (Line 157-163).

"To investigate the short-term fitness implications of communal breeding, we measured parental performance for each individual (i.e. time spent providing care on the carcass and weight change)

and of groups (i.e. carcass burial and the total amount of parental investment by groups), as well as reproductive outcome of groups (i.e. brood size and larval weight), in communal and non-communal breeding. We predict that communal groups achieve higher reproductive outcome compared to non-communal groups.”

We also agree with you that it is important to test each individual’s reproductive fitness benefits in communal groups. So, in another study, we analysed the parentage of offspring produced on large carcasses (25 g) in such communally breeding events, and found that large individuals produced a large proportion of offspring (approx. 80% of offspring), whereas small individuals produced a relatively small proportion of offspring in a shared brood (small females produced approx. 20% of offspring)(these results were mentioned in the methodology; Line 222-228). In communal groups of burying beetles, the allocation of reproduction between dominants and subordinates is largely determined by carcass size. Specifically, on small carcasses, there often occurs an extremely skewed allocation in reproduction for individuals (i.e. the dominant pair produces most of the offspring), while the reproductive allocation is often more equal and relatively constant between dominant and subordinate individuals on large carcasses (e.g. 80% of offspring produced by dominants) (Trumbo and Fiore, 1994; Eggert and Müller, 1992; Müller et al., 2007).

“In another study, we tested the parentage of offspring produced in double-pair breeding groups using 5 microsatellite loci [80], and found that small pairs of individuals produced a relatively small proportion of offspring in a shared brood (the percentage of offspring produced by small females: mean \pm SE = 20.87 \pm 6.16 %, 9 out of 10 small females produced; and by small males: mean \pm SE = 7.18 \pm 4.13 %, 5 out of 10 small males produced; Nbrood = 10, Noffspring = 204)[76], indicating that communal breeding events (i.e. one shared brood) were indeed induced in such double-pair treatment.”

References:

- Clutton-Brock T.M. (2009) Cooperation between non-kin in animal societies. *Nature*, 462: 51-57.
- Eggert A.K., Müller J.K. (1992) Joint breeding in female burying beetles. *Behavioral Ecology and Sociobiology*, 31: 237-242.
- Eggert A.-K., Otte T., Müller J.K. (2008) Starving the competition: a proximate cause of reproductive skew in burying beetles (*Nicrophorus vespilloides*). *Proc. R. Soc. B*, 275: 2521-2525.
- Komdeur J., Schrama M.J.J., Meijer K., Moore A.J., Beukeboom L.W. (2013) Cobreeding in the burying beetle, *Nicrophorus vespilloides*: Tolerance rather than cooperation. *Ethology*, 119(12): 1138-1148.
- Müller J.K., Braunisch V., Hwang W., Eggert A.K. (2007) Alternative tactics and individual reproductive success in natural associations of the burying beetle, *Nicrophorus vespilloides*. *Behavioral Ecology*, 18(1): 196-203.
- Richardson J., Smiseth P.T. (2020) Maternity uncertainty in cobreeding beetles: females lay more and larger eggs and provide less care. *31(3)*: 641-650.
- Shen S.F., Emlen S.T., Koenig W.D., Rubenstein D.R. (2017) The ecology of cooperative breeding behaviour. *Ecology Letters*, 20(6): 708-720.
- Trumbo S.T., Fiore A.J. (1994) Interspecific competition and the evolution of communal breeding in burying beetles. *Am. Midl. Nat.* 131: 169-174.

Reviewer comments to Author:

Reviewer: 1

Comments to the Author(s)

Letter to Authors:

1.1 The social environment that an individual experiences may have profound effects on individual reproductive success within and across breeding attempts. Here, the authors argue that early breeding experience has the potential to impact lifetime fitness due to carry-over effects. The carry-over effects of communal breeding may impact individuals in a sex-specific way. Here, the authors

conduct an interesting study to tease apart how experience of communal breeding may impact future breeding attempts.

Thank you very much for the positive comment. We have now revised the whole manuscript according to your constructive and helpful comments as outlined below.

1.2. I think it would strengthen the manuscript to have a clearer unifying framework that provides context for the tests and results and clearly states what a) is known/not know and b) what each measure provides inference into. Why was the experiment set up the way that it was, and why were the factors that you measured selected? E.g., small vs large beetles were used, but it is not explicit in the Introduction what this would be a test of (although it is briefly mentioned on line 200). Why was carcass burial, egg laying period, larval hatching time etc. recorded? I'm sure there are good biological reasons to do so, but what do they tell you specifically? E.g., is a longer egg-laying period "bad" for some reason? Why was parental weight change investigated as a metric of parental effort? What is known versus not known for this system? For example – on line 235 it states that weight change was used to gauge parental effort, but then on line 314 it makes it sound like it was unknown whether weight change was correlated with variation in parental investment. Is this a test of assumption or a result or something unexpected based on previous research, and if so, how does it fit into the overall framework of the study? Overall, it would provide clarity to the manuscript to state in the Introduction what kinds of things you need to measure to answer the question, e.g., measures of parental effort, care behavior and competitive ability, and then in the Methods, you define how each of these was measured: e.g., parental effort was measured as amount of time spent on carcass, length of egg-laying period (because this is a reliable proxy of xx), etc.

We thank the reviewer very much for these constructive comments. We have now clarified the set-up of our study, the selected measures and their importance in answering our research questions in the introduction and method sections of our revised manuscript.

(Responses to the comments in yellow above)

First, in burying beetles, adult body size affects an individual's competitive ability, which then is closely linked with carcass monopolization and reproductive success. Importantly, in communal breeding, a dominance hierarchy in carcass monopolization and reproduction is largely established according to size difference between individuals. That is, large individuals become dominants and monopolize the carcass, while small individuals become subordinates and have limited access to the carcass and reproductive shares. Thus, in our study, we created communal breeding events by introducing two pairs of individuals that differed in body size. We have now clarified this point in the abstract and introduction of our revised manuscript (Line 40-43 and 129-133).

"In such communal groups, large individuals became dominant that had a monopoly in carcass and reproduction, while small individuals (i.e. subordinates) were restricted to access the carcass."

"In such associations, social dominance is largely established through several rounds of fights, with larger individuals being more likely to win fights and become dominant compared to smaller individuals [35,64 - 66]. Specifically, large individuals become dominants can largely monopolize the carcass, while subordinates are smaller individuals and have limited access to the carcass [35,65,66]."

We have also now clarified the experimental design in the revised method section (Line 207-220):

"In burying beetles, adult body size affected an individual's competitive ability, while this was closely linked with carcass monopolization and reproductive success [65,74,75]. To create the communally breeding treatment, one large pair (dominants) and one small pair (subordinates) of beetles ($n = 19$) were selected, and we ensured that the two pairs of beetles were unrelated and differed approximately 10 % in body size, because the likelihood of stable dominance hierarchy establishment in carcass use was enhanced when size difference between opponents was larger [64,72]. Other studies of ours indicated that the double-pair breeding treatment could create a communal breeding event with a size-dependent dominance hierarchy in carcass monopolization

[76], which was accordance with previous studies about communal breeding in burying beetles [63,65,77]. For these beetles, an individual's body size has also been found to influence its parental investment towards current brood [78]. Thus, we randomly selected one pair of beetles from small-pair ($n = 10$) or large-pair ($n = 10$) groups to create the non-communal breeding treatment, which could control the effect of body size on individual behaviour (i.e. parental investment) [78,79]. Just prior to creation of communal and non-communal breeding, each individual was weighed (accuracy: 0.0001 g).”

(Responses to the comments in pink above)

Second, we thank the reviewer very much for the comment about the justification for measures such as carcass burial, egg-laying period, larval hatching time. The degree of carcass burial is a good measure for assessing the joint performance of individuals of breeding groups or pairs, because burying beetles cooperate and invest in carcass burial together. In communal breeding events, dominant individuals largely monopolized the carcasses and produce a large proportion of offspring, while the smaller subordinate individuals could also reproduce by laying eggs around the large carcasses. Subsequently, this may lead to intense reproductive competition between the females, resulting in higher selection pressure on females to lay their eggs earlier (which hatch earlier), thereby benefiting their own offspring. Thus, we expect egg laying time and period and larvae hatching time to be earlier and shorter in communal groups than in non-communal groups. Therefore, we used egg-laying time (the time from the start of the experiment until the onset of egg laying) and larvae-hatching time as measures to investigate the extent of reproductive competition and the reproductive strategies in groups. We have now clarified these in our revised manuscript (Line 245-264):

“As a proxy for the joint parental performance of breeding groups or breeding pairs in burying beetles, the degree of carcass burial was estimated according to the fraction of the mouse above the ground and carcass roundness [85]. In communal breeding events, dominant individuals largely monopolized the carcass, while both dominant and subordinate females were able to reproduce offspring by laying eggs surrounding the carcass [65,66,75]. This resulted in selection for females to lay their eggs earlier (which hatch earlier) and synchronously lay eggs with other female cobreeders [66,72,77]. Thus, we examined the competitive interaction in reproduction and reproductive strategies between communal and non-communal groups, by recording the timing of reproductive performance and offspring development. As measures for timing of reproductive performance, we recorded egg-laying time (the time from the start of the experiment until the onset of egg laying) and larvae-hatching time (the time from the start of the experiment until the onset of larvae hatching). We defined 'the onset of egg laying' as the onset observational time when some eggs were found at the bottom of the box, and defined 'the onset of larvae-hatching' as the onset observational time when some of larvae newly hatched and did not enter into 1st instar. As measures for offspring development and reproductive performance, we used larvae-dispersing time (the time from the start of the experiment until the onset of larvae dispersing), brood size (number of larvae) and average larval weight (total weight of larvae/brood size) at the larval dispersal (i.e. larvae dispersed from the carcass). Also, we recorded weight change during breeding ($[\text{final weight} - \text{initial weight}] / \text{initial weight}$) and survival at the end of the experiment of adult individuals, as parameters for parental effort during the entire breeding period and reproductive costs for individuals, respectively [49,57].”

(Responses to the comments in grey above)

Third, we thank very much for your comment about weight change as metric for parental effort. In burying beetles, weight change during the entire breeding period can be investigated as a metric of parental effort. However, weight change of individual burying beetles is a mixed consequence of weight gain, as a consequence of individuals feeding from the carcass, and weight loss as a consequence of providing parental care. It cannot be ignored that weight change should be investigated for assessing parental effort in burying beetles, and there was a significant negative correlation between parental effort and weight change (Wang et al., 2021). Moreover, some studies on burying beetles suggested that individual's weight prior to breeding was positively associated with its future parental effort

(Creighton et al., 2009; Richardson et al., 2020; Wang et al., 2021). Thus, we used prior weight change to assess its effect on future parental effort. We have now clarified this in our revised manuscript (Line 262-268 and 330-334):

“Also, we recorded weight change during breeding ($[\text{final weight} - \text{initial weight}] / \text{initial weight}$) and survival at the end of the experiment of adult individuals, as parameters for parental effort during the entire breeding period and reproductive costs for individuals, respectively [49,57]. After the first reproductive round, surviving beetles were kept individually for five days in rearing boxes (10×5 cm and 8.5 cm high) and not fed by any food (i.e. mealworms) for subsequent experimentation. We did not feed beetles during this period to avoid the potential effect of food consumption and weight change on an individual’s behaviour in the subsequent period. ...

In burying beetles, an individual’s body weight prior to breeding has been shown to be positively associated with its parental effort in the subsequent breeding events [54,89]. In our study, we also examined whether the body weight of individuals at the end of breeding either as communal or non-communal groups was associated with their parental effort when subsequently breeding as pairs.”

References:

Creighton J.C., Heflin N.D., Belk M.C. (2009) Cost of reproduction, resource quality, and terminal investment in a burying beetle. *Am. Nat.* 174(5): 673-684.

Richardson J., Stephens J., Smiseth P.T. (2020) Increased allocation to reproduction reduces future competitive ability in a burying beetle. *J. Anim. Ecol.* 89(8): 1918-1926.

Wang W., Ma L., Versteegh M.A., Wu H., Komdeur J. (2021) Parental care system and brood size drive sex difference in reproductive allocation: an experimental study on burying beetles. *Front. Ecol. Evol.* 9: 739396.

The experiment is set up to distinguish between age-effects and experience effects (communal to non-communal and non-communal to non-communal), but changes in reproductive investment over aging are not discussed. There are good life history reasons to expect that investment may change over an individual’s lifespan, and it needs to be mentioned (at minimum).

Thank you very much for the comment. We agree with you that changes in reproductive investment over lifespan should also be discussed because of the importance of age-effect on parental investment between breeding attempts. We have now discussed this point in our revised manuscript (Line 527-537):

*“Previous studies have demonstrated age-related differences in reproductive effort and reproductive allocation between breeding attempts [89,99 – 101]. In *N. orbicollis*, older individuals have a higher level of investment to current reproduction (e.g. produce larger broods and consume less of the carcass) compared to younger individuals, because older individuals with a low residual reproductive value have limited future reproductive opportunities [89,100]. These previous studies also highlight the impact of individual reproductive restraint and senescence on individual reproductive effort [89,99]. Moreover, more recent work revealed an interplay between age and previous breeding experience on reproductive investment in burying beetles [62,101]. For example, female *N. vespilloides* that have invested more in reproduction during earlier reproductive attempts show a decline in reproductive investment, but this negative effect of previous reproductive investment is observed only in older females [62].”*

References:

Cotter S.C., Ward R.J.S., Kilner R.M. (2010) Age-specific reproductive investment in female burying beetles: independent effects of state and risk of death. *Functional Ecology*, 25(3): 652-660.

Also, for the experiment, why were the mates switched in the second breeding bout (line 248)? The authors mention that paternity assurance may affect behavior, but then are manipulating paternity assurance. Doesn’t this species store sperm to some extent? So that males in the second breeding bout will likely not have paternity assurance? Is there any reason to expect that this would influence behavior or investment in later breeding attempts? This is worth mentioning/discussing somewhere,

especially since the authors already state that paternity assurance is expected to impact behavior (e.g., line 477).

We thank the referee very much for the constructive comment. In our study, the mates were switched in the second breeding bout, because we aimed to create new breeding pairs where females and males had either same or different prior breeding experiences (communal and non-communal breeding experience) but were not familiar with each other. When multiple males compete for a single carcass, uncertain and reduced paternity may negatively affect a male's parental behaviour. For example, when multiple males compete for one carcass the level reproductive competition is high, and consequently males extend their parental care duration (Hopwood et al., 2015). However, such effect on parental behaviour is not the case for breeding pairs (without other conspecifics), because in pairs males are always the sole father and can assure their paternity by repeatedly copulating with their mates (House et al., 2007; Pettinger et al., 2011; Engel et al., 2014; Luzar et al., 2017). While female burying beetles can store sperm of multiple males, a high copulation rate likely functions to dilute stored sperm of the female's previous mates or fresh sperm from sneaky males, thereby maximizing a male's paternity in the current brood (Pettinger et al., 2011; Engel et al., 2014). This indicates that our experimental setup of breeding pairs consisting of an unfamiliar male and female is unlikely influence behaviour or investment in later breeding attempts, and males will have paternity assurance in the second breeding bout. We agree with you that it is worth mentioning the impact of paternity assurance on behaviour. We have now mentioned these in our revised manuscript (Line 281-291 and Line 589-605):

"When more than one male beetles were found to compete for one carcass, uncertain and reduced paternity may affect an individual's parental behaviour. For example, males extended parental care duration when reproductive competition is high (Hopwood et al., 2015). However, such effect on parental behaviour was not the case for newly-formed breeding pairs in the second breeding bout (without other conspecifics). Females may still have stored sperm from the first breeding bout, with which she could fertilize eggs produced in the second breeding bout. However, males prevent this through their copulation tactics during the pre-hatching period. For males, a high copulation rate likely functions to dilute stored sperm of the female's previous mates or fresh sperm from sneaky males, thereby maximizing a male's paternity in the current brood [86,87]. This suggested that, in the second experiment, the uncertainty of paternity was unlikely to have an effect on an individual's parental care."

*"However, when more than one males compete for one carcass (e.g. in communal breeding), males may adjust the level of parental care in response to the uncertainty of paternity by reducing investment to the current broods and allocating more resources to future reproduction [49,56]. Previous study shows that male *N. vespilloides* have an extended their parental care duration when other same-sex conspecifics are present and reproductive competition is high. It is likely that males could gain compensatory benefits by extending their duration within carcasses, or consuming more of the carcasses [115,116]. Dominant males are found to produce a large proportion of offspring in communal groups because of their monopoly in carcass access, and they are able to maximize their paternity in the current brood by repeatedly and frequently copulating with dominant females (also with subordinate females) prior to and during the period of egg laying [116,117]. However, there still occurs a limited control in reproduction between males because subordinate males may occasionally sire a proportion of the offspring by sneakily mating with all females [87,117]. While subordinate males usually leave the carcass earlier and do not provide post-hatching care towards offspring when the dominant male is present, the opportunity to access the carcass can motivate male parental behavior [87,116]. These results indicate that access to the carcass is beneficial for males, because they are able to save resources for themselves or compensate for a reproductive cost due to the uncertainty of paternity."*

References:

Hopwood P.E., Moore A.J., Tregenza T., Royle N.J. (2015) Male burying beetles extend, not reduce, parental care duration when reproductive competition is high. *Journal of Evolutionary Biology*, 28(7): 1394-1402.

- House C.M., Hunt J., Moore A.J. (2007) Sperm competition, alternative mating tactics and context-dependent fertilization success in the burying beetle, *Nicrophorus vespilloides*. *Proc. R. Soc. B*: 274: 1309-1315.
- Pettinger A.M., Steiger S., Muller J.K., Sakaluk S.K., Eggert A.-K. (2011) Dominance status and carcass availability affect the outcome of sperm competition in burying beetles. *Behavioral Ecology*, 22(5): 1079-1087.
- Engel K.C., von Hoermann C., Eggert A.-K., Muller J.K., Steiger S. (2014) When males stop having sex: adaptive insect mating tactics during parental care. *Animal Behaviour*, 90: 245-253.
- Luzar A.B., Schweizer R., Sakaluk S.K., Steiger S. (2017) Access to a carcass, but not mating opportunities, influences paternal care in burying beetles. *Behavioral Ecology and Sociobiology*, 71(348).

While the study design does seem well set-up, the sample sizes are both low and uneven. E.g., for the test of carry-over effects, there ends up being comparisons of treatments with an $n=3$ vs $n=12$ (Figure 5, larval hatching times), but these discrepancies are not described. Who was excluded and why? Why don't the sample sizes match up for the same treatments across the figures – e.g., on Figure 4a/b there are 21 communal and 13 non-communal females, and only 20 and 14 males, respectively. The individuals were paired, so where does this disparity come from?

Thank you very much for the comment. These discrepancies in sample size (e.g. larvae-hatching time) are because the first experiment generated different sample sizes (i.e. number of surviving beetles from the first breeding attempts) and we excluded some unreasonable data in the second breeding experiment. In the first experiment, communal or non-communal breeding events both led to death of individuals due to fightings and costs of reproduction and then generated different number of surviving beetles that could be used for the subsequent experiment. In our study, we defined larvae-hatching time as observational time when most of larvae newly hatched but not entered into 1st instar. So, we did not include any observational time if most of larvae have not newly hatched and entered into 1st instar, which could largely avoid any erroneous records about larvae-hatching time. We have now clarified the definition in our revised manuscript (Line 253-258). In Figure 4a/b, while the individuals were paired, for each pair females and males are similar or different with respect to prior breeding experience. Thus, this leads to the disparity in sample size.

“As measures for timing of reproductive performance, we recorded egg-laying time (the time from the start of the experiment until the onset of egg laying) and larvae-hatching time (the time from the start of the experiment until the onset of larvae hatching). We defined ‘the onset of egg laying’ as the onset observational time when some eggs were found at the bottom of the box, and defined ‘the onset of larvae-hatching’ as the onset observational time when some of larvae newly hatched and did not enter into 1st instar.”

How often do you expect for individuals of this species to have multiple breeding attempts in the wild, given how limited/ephemeral the breeding resource is? If it is reasonably rare – realistically, how important may carry-over effects be for generating variation for selection to act on (e.g., line 392, 420)? How big of deal do we expect first breeding experience to be, if it's almost always the only breeding experience? Is this species appropriate for a test of carry-over effects across breeding bouts? Also, it is perhaps too broad of a stroke to say that breeding communally decreases fitness. If they have no other opportunity to breed besides communally, it's still a better option than not breeding at all.

Thank you very much for this valuable comment. Burying beetles are multiple breeders with most individuals having more than one breeding attempts in the wild, if breeding resources are abundantly available (Scott, 1998; Ward et al., 2009). Although for burying beetles, the breeding resource (i.e. carcasses) is unpredictable and limited which may cause communal breeding, however – albeit unpredictable and limited - breeding resources are readily available across the entire breeding season enabling individuals breeding multiple times

during a breeding event. Individuals could have an improved group performance in carcass preparation and defence against intruders (mutualistic benefits; Sun et al., 2014; Liu et al., 2020). However, individuals that reproduce in communal groups may suffer reproductive costs compared to individuals that breed as pairs (mutualistic benefits; Eggert and Muller, 1992; Komdeur et al., 2013; Liu et al., 2020), although on the other hand, breeding communally is a better option than not breeding at all (i.e. 'a-best-of-the-bad-job' strategy) if they have no other breeding opportunity. It is likely that communal breeding is an adaptive strategy that could improve an individual's overall fitness benefits across breeding bouts and over and individual's lifetime. Therefore, it is worth to study the carry-over effects of communal breeding in the burying beetle in particular, and it is important to examine whether and how carry-over effects act on the selection for group breeding in animals in general. As such, the communally breeding system in burying beetles would be good model for studying the evolution of group breeding. We have now added this in our revised manuscript (Line 139-154):

"For burying beetles, the breeding resource (i.e. carcasses) is ephemeral and limited, which largely determines an individual's breeding opportunity during a given time frame, as well as its reproductive strategy [66,70]. While individuals that breed in communal groups may, at least in the short term, achieve a reduced or no fitness benefit in reproduction, compared to individuals that breed alone or as pairs, breeding communally is still a better option for burying beetles than not breeding at all (i.e. 'the-best-of-a-bad-job' strategy), if they have no another breeding opportunity because of the constraints of breeding resources [59,64,70]. These beetles are opportunistic breeders, and may have multiple breeding attempts over their lifetime in a benign environment (e.g. a high resource availability and a low interspecific competition pressure) [70,71]. It is likely that breeding communally among individuals evolves as an adaptive strategy, which may largely reduce the reproductive costs in the adverse environmental conditions but potentially improve the overall fitness benefits across breeding bouts and even over lifetime. Thus, it is worth studying the carry-over effects of communal breeding on parental behaviour and fitness across breeding attempts in burying beetles, and it is also important to examine whether and how such effects, concurrent with the short-term benefits on fitness, act on the selection for group breeding and its adaptation to ecological environments."

References:

- Scott, M.P. (1998) The ecology and behavior of burying beetles. *Ann. Review Entomol.* 43: 595-618.
- Ward R.J.S., Cotter S.C., Kilner R.M. (2009) Current brood size and residual reproductive value predict offspring desertion in the burying beetle *Nicrophorus vespilloides*. *Behavioral Ecology*, 20(6): 1274-1281.
- Sun S.J., Rubenstein D.R., Chen B.F., Chan S.F., Liu J.N., Liu M., Hwang W., Yang P.S., Shen S.F. (2014) Climate-mediated cooperation promotes niche expansion in burying beetles. *eLife*, 3: e02440.
- Liu M., Chen B-F., Rubenstein D.R., Shen S.F. (2020) Social rank modulates how environmental quality influences cooperation and conflict within animal societies. *Proc. R. Soc. B*, 287: 20201720.
- Eggert A.K., Müller J.K. (1992) Joint breeding in female burying beetles. *Behavioral Ecology and Sociobiology*, 31: 237-242.
- Komdeur J., Schrama M.J.J., Meijer K., Moore A.J., Beukeboom L.W. (2013) Cobreeding in the burying beetle, *Nicrophorus vespilloides*: Tolerance rather than cooperation. *Ethology*, 119(12): 1138-1148.

I think that the results should be re-written with the means \pm SE written in the body of the text. I understand that the statistical analyses are in Table 1, but this table is reasonably cumbersome to navigate because there are so many footnotes, and the directionality of these results would be more straightforward to grasp in the body of the text.

We thank the reviewer very much for the comment and we have now re-written the result sections (with means \pm SE in the body of the text) in our revised manuscript (Line 344-443).

“(a) Experiment 1: Immediate implications of communal versus non-communal breeding on individual parental performance and fitness

Our results showed that communal groups had a better performance in carcass burial at the initial stage of the experiment, as communal groups were significantly faster in burying carcasses than non-communally breeding groups at 36, 44 and 60 hours (36-h: $t = 3.16$, $p = 0.02$; 44-h: $t = 3.25$, $p = 0.01$ and 60-h: $t = 7.28$, $p < 0.001$; figure 2; supplementary materials, table S1). After ca. 68 hours, there was no difference in the degree of carcass burial between communal and non-communal groups (figure 2; table S1). In the communal groups, we found that the larger pairs of individuals became dominant and significantly monopolized the access of carcass, whereas the smaller individuals were subordinates that spent less time on the carcass compared with larger individuals during the entire breeding period (mean \pm SE of large vs. small individuals: 58.96 ± 2.95 vs. $7.22 \pm 1.33\%$; $\chi^2 = 315.65$, $p < 0.001$; Table 1). At the individual level and within size classes, large females spent a similar amount of time providing care on the carcass in communal and non-communal groups (mean \pm SE in communal vs. non-communal groups: 64.79 ± 4.64 vs. $74.89 \pm 4.07\%$; hereafter mean \pm SE shown as communal versus non-communal groups), while large males of communal groups spent significantly more time on the carcass than large males of non-communal groups (mean \pm SE: 53.43 ± 3.23 vs. $34.24 \pm 3.79\%$; figure 3a; table 1A and S2). Both large females and large males spent a similar amount of time on the carcass in communal and non-communal groups, while small females and small males spent significantly less time on the carcass in communal groups than in non-communal groups (mean \pm SE of small females: 10.34 ± 2.22 vs. $68.37 \pm 6.80\%$; of small males: 4.09 ± 1.13 vs. $41.11 \pm 6.08\%$; figure 3a; table 1A and S2). Moreover, at the group level, communal groups had a significantly higher total amount of time spent by all individuals than non-communal groups (mean \pm SE: 33.09 ± 1.78 vs. $27.33 \pm 1.57\%$; table 1A). These results indicated that breeding in communal groups showed a better group performance in carcass burial and total investment in care compared with breeding in non-communal groups, whereas parental behaviour was similar for individuals in communal and non-communal groups.

During breeding, on average, individuals lost weight (mean \pm SE = $-24.11 \pm 2.74\%$, $n = 78$; table 1A), indicating that parental investment during breeding was costly. No difference in weight change was observed for large females, small females, and large males in communal and non-communal groups, while small males had significantly less weight loss in communal groups than in non-communal groups (figure 3b; table 1A and S2). At larval dispersal, there was no difference in mortality of both large and small females, as well as for large males, in communal and non-communal groups, while small males experienced higher mortality in communal groups than in non-communal groups (see supplementary materials, figure S1).

There was no evidence that communal breeding resulted in higher reproductive outcome of groups compared to non-communal breeding (table S1). Communal groups laid their eggs significantly earlier and produced lighter larvae compared to non-communal groups (mean \pm SE of egg-laying times: 77.00 ± 5.16 vs. 93.33 ± 5.34 h, $\chi^2 = 4.75$, $p = 0.03$; of averaged larval weight: 0.1160 ± 0.0090 vs. 0.1457 ± 0.0070 g, $F = 6.39$, $p = 0.02$; table S1). Both groups had a similar egg-laying period and larvae-hatching time (mean \pm SE of egg-laying period: 94.67 ± 4.78 vs. 83.14 ± 6.28 h, $\chi^2 = 2.88$, $p = 0.09$; of larvae-hatching time: 143.59 ± 7.58 vs. 149.14 ± 7.30 h, $\chi^2 = 1.18$, $p = 0.28$), and at larval dispersal, brood size of offspring was also similar (mean \pm SE: 15.85 ± 1.65 vs. 14.73 ± 1.61 ; $F = 0.23$, $p = 0.64$; table S1).

“(b) Experiment 2: Carry-over effects of communal versus non-communal breeding on parental performance and fitness

Prior breeding experience of individuals had a significant effect on parental performance in carcass burial (table 2 and S3). Specifically, females that originated from communal groups had lower rates of carcass burial than females from non-communal groups (experience: $\chi^2 = 26.41$, $p < 0.001$; experience \times time: $\chi^2 = 7.67$, $p = 0.01$; figure 4a; table 2 and S3), whereas males that originated from communal and non-communal groups had similar rates of carcass burial (experience: $\chi^2 =$

9.09, $p = 0.005$; experience \times time: $\chi^2 = 2.34$, $p = 0.13$; figure 4b; table 2 and S3). Moreover, the interaction of prior breeding experience of females and males had a significant influence on the rates of carcass burial, as a higher rate of carcass burial was observed in pairs where females originated from non-communal groups and males originated from communal groups, compared with other pairs (female experience \times male experience \times time: $\chi^2 = 4.84$, $p = 0.03$; figure S2; table S3).

The prior breeding experience of focal individuals and that of their partners jointly influenced their parental investment time (table 1B). At the individual level, females and males that originated from communal or non-communal groups spent a similar parental investment time on the carcass (mean \pm SE of females: 61.15 ± 5.93 vs. 78.92 ± 6.84 %; of males: 38.56 ± 7.50 vs. 42.45 ± 5.91 %; figure 4c; table 1B and S2) and had similar weight changes during breeding (table 1B and S2). Regardless of their own prior breeding experience, both females and males spent significantly more time on the carcass when their partners originated from communal groups than from non-communal groups (mean \pm SE of females' partner: 77.19 ± 4.15 vs. 54.75 ± 8.74 %; of males' partner: 46.99 ± 6.44 vs. 29.12 ± 7.10 %; figure 4c, d; table 1B and S2). Specifically, both females and males that originated from communal groups spent significantly less time on the carcass than individuals that originated from communal groups when their partners originated from non-communal groups (mean \pm SE of females: 44.88 ± 9.45 vs. 72.50 ± 15.85 %; of males: 20.41 ± 8.64 vs. 43.06 ± 10.24 %; $z = -3.01$, $p = 0.005$; table 1B and S2), whereas parenting investment time was similar for individuals that originated from communal or non-communal groups when their partners originated from communal groups (mean \pm SE of females: 73.36 ± 5.62 vs. 82.94 ± 5.90 %; of males: 50.65 ± 11.38 vs. 42.11 ± 7.68 %; $z = -0.43$, $p = 0.89$; table 1B and S2). At the pair level, the interaction of female and male breeding experience had a significant effect on the total parenting investment time by pairs (table 1B). When females originated from communal groups, there was a higher total parental investment time in the presence of males from communal groups than from non-communal groups (mean \pm SE: 62.00 ± 5.82 vs. 43.50 ± 5.50 %; $z = 3.04$, $p = 0.005$; table 1B and S2), whereas in groups where females originated from non-communal groups, the total investment time by pairs was similar when male originated from communal or non-communal groups (mean \pm SE: 51.67 ± 3.94 vs. 57.78 ± 9.39 %; $z = -0.82$, $p = 0.65$; table 1B and S2). For both females and males, we found that all individuals that originated from communal groups suffered higher mortality during breeding than from non-communal groups (26 out of 42 vs. 18 out of 28; table 1B). Additionally, an individual's parental performance (i.e. parenting investment time and weight change of each individual) was not associated with its weight change during the previous communal breeding events (table S4).

Females with communal breeding experience produced larvae that hatched and dispersed later from the carcass compared to females that originated from non-communal groups (figure 5a,b; table 2 and S3), while no difference was observed for males that originated from communal and non-communal groups in the timing of offspring development (table 2 and S3). Regardless of male's prior breeding experience, females that originated from communal and non-communal groups had similar reproductive outcomes (mean \pm SE of brood size: 22.61 ± 2.15 vs. 25.08 ± 1.28 ; of average larval weight: 0.1389 ± 0.0067 vs. 0.1333 ± 0.0053 g; table 2 and S3). However, females that originated from communal groups produced smaller broods of offspring when males originated from non-communal groups, compared to females from non-communal groups (mean \pm SE: 17.57 ± 3.93 vs. 25.40 ± 2.56 ; $z = -2.36$, $p = 0.04$; table S3). Males that originated from communal groups produced larger broods than males that originated from non-communal groups (mean \pm SE: 25.42 ± 1.33 vs. 20.83 ± 2.70), while the average larval weight was similar for males that originated from communal and non-communal groups (mean \pm SE: 0.1389 ± 0.0006 vs. 0.1328 ± 0.0056 g; table S2). Also, the size of males that originated from communal and non-communal groups significantly influenced the reproductive outcomes (table S3). In particular, large males that originated from communal groups produced heavier larvae than large males from non-communal groups (figure 5d; table 2 and S3), whereas the reproductive outcome was similar for small males that originated from communal and non-communal groups (figure 5c, d and S3; table 2 and S3)."

The discussion does not put the results into context. Why do you think you saw the results that you did? What mechanisms could describe your results, what considerations must be considered, how does this improve or change our understanding of carry-over effects? For example, it is concluded that there is a carry-over effect on mortality but no discussion of this mortality. Why do you think that so many of the individuals from the communal treatment die – where they killed by the other beetles, starved, or just dead? What have other studies found for mortality in such group breeding, if any? Why does experience matter – because it changes physiological state, or the beetles are learning, or something else? Why does the experience of the partner in the second brood matter? E.g., why do they spend more time on the carcass if the partner was from a communal environment? Do you think they can tell what their partner's previous experience was, or are they responding to changes in partners behavior or pheromones or something else? There are a lot of interesting results and as a reader, I want to hear the authors take on what they mean.

Thank you very much for the comment and also for the notion that there are a lot of interesting results. We have now discussed the results further in our revised manuscript. First, we have now discussed the underlying mechanism about the carry-over effect of communal breeding on mortality. We suggest that this effect may be due to that each individual incurs a physical cost from communally breeding (e.g. more fighting and high injuries compared to when breeding in pairs; Komdeur et al., 2013), or that enhanced allocation to reproduction comes at the costs of high mortality for individuals that communally bred. (Line 487-501)

*“This may be due to that each individual incurs a high cost in reproduction and survival from communally breeding, or that an enhanced allocation to reproduction comes to at the cost of high mortality for individuals that communally bred [57,62,90]. In communal groups, the establishment of a dominance hierarchy is suggested to reduce the extent of aggressive interaction, while injuries are often observed [64,70]. It is likely that these injuries affect an individual's future reproduction and increase the risk of death. In this scenario, a microbe-rich soil environment might deteriorate some opportunistic infections arising from fight wounds, thereby increasing the rate of mortality [62]. For each individual, dominance status is closely associated with fecundity and nutritional state [35,63]. For example, in *N. vespilloides*, a difference in fecundity between females is due to their dominance status, because limited access to the carcass leads to low fecundity by causing nutritional deficiencies [63]. Such effects in fecundity and nutritional state are also accompanied with some negative changes in physiological state and immunity, which may have long-lasting influences on individual fitness benefits in the future, e.g. high mortality and short lifespan for individuals that originate from communal groups. To which extent the high mortality for males and females that originated from communal groups may offset their reproductive benefits remains to be investigated [49,54,96].”*

Second, the carry-over effect of communal breeding may function through the gain of prior breeding experience as communal breeders, and this is associated with age- and state-dependent effects (including physiological changes and gain of skills). We have now discussed this in our revised manuscript (Line 524-544):

*“This might mean that the carry-over effects of communal breeding on individual parenting behaviour and fitness is not simply determined by the parental investment to previous reproduction, but instead is associated with the individual's prior experience [57,96]. Previous studies have demonstrated age-related differences in reproductive effort and reproductive allocation between breeding attempts [90,100 - 102]. In *N. orbicollis*, older individuals have a higher level of investment to current reproduction (e.g. produce larger broods and consume less of the carcass) compared to younger individuals, because older individuals with a low residual reproductive value have limited future reproductive opportunities [90,101]. These previous studies also highlight the impact of individual reproductive restraint and senescence on individual reproductive effort [90,100]. Moreover, more recent work revealed an interplay between age and previous breeding experience on reproductive investment in burying beetles [62,102]. For example, female *N. vespilloides* that have invested more in reproduction during earlier reproductive attempts show a decline in reproductive investment, but this negative effect of previous reproductive investment is observed only in older*

females [62]. Also, having prior breeding experience as communal breeders is associated with changes in physiological states, as well as the gaining of social skills [35,63,103]. These state-associated changes that may influence individual reproductive residual value should be considered, as they could determine an individual's reproductive effort for future reproduction, probably due to physiological constraints and the lack of experience. Thus, we suggest that the effects of communal breeding on fitness are associated not only with short-term costs and benefits, but also with carry-over effects on future fitness [32,59,67,104], albeit such effects are more pronounced for dominant males than for dominant females [49,82,97,98].”

Third, we further discuss the effect of each individual's partner's previous experience on its parental investment and have now clarified this in our revised manuscript (Line 561-576):

“In support of this, in the presence of females that originated from communal groups, there occurred an increased level of care provided by pairs when males had experience as communal breeders compared with males that had no experience as communal breeders. It has been well known that hormonal levels (e.g. juvenile hormones) rapidly change during breeding in burying beetles [52,110]. These changes intrinsically mediate the emission of pheromones involved, which may have a lasting influence on individual reproductive states and parental behaviour in the subsequent period. During breeding, two parents are able to effectively communicate with each other via the emission of pheromones, which could help them to simply recognize their partner's reproductive states and benefits for coordinating parental and mating effort [111]. We suggest, for each individual, such adjustment in parental investment depending on its partner's prior breeding experience may be via the pheromone-dependent recognition. That is, each individual is likely to recognize its partner's history of prior breeding via the emission of pheromones, and then strategically adjust its own parental investment to current brood [52,111]. We suggest that future work should consider the flexible adjustment of females and males in parental investment based on the prior breeding experience of their partners and the recognition system involved, and its subsequent influence on common benefits for pairs.”

References:

- Komdeur J., Schrama M.J.J., Meijer K., Moore A.J., Beukeboom L.W. (2013) Cobreeding in the burying beetle, *Nicrophorus vespilloides*: Tolerance rather than cooperation. *Ethology*, 119(12): 1138-1148.**
- Scott M.P. (2006) The role of juvenile hormone in competition and cooperation by burying beetles. *Journal of Insect Physiology*, 52(10): 1005-1011.**

I am so curious what was going on during the communal breeding events – did the authors observe competition, actual cooperation between them, did the beetles mostly ignore the other pair or were there constant interactions, etc.? I think understanding what happens socially on the communal carcass will be important for understanding what carry-over effects we may expect to see. At minimum, may be an interesting discussion point or future direction.

Thank you very much for your constructive comment. We have now clarified the biology of communal breeding in our revised manuscript. In burying beetles, we observed a consistent competition in carcass access and reproduction between individuals, especially before larvae-hatching, and observed that more than two individuals jointly engaged in carcass burial and dominant pairs sometimes ignore the other pair (i.e. multiple individuals stayed on the carcass at the same time). These are also consistent with previous studies (Eggert and Sakaluk, 2000; Eggert and Muller, 1992; 2011; Muller et al., 2007). Our study also supports that communal groups have a better joint performance in carcass burial compared to pair-breeding groups. This could be considered a mutualistic benefit of communal breeding. In communal groups, a dominance hierarchy could be established after several rounds of fights, with larger individuals being more likely to become dominant. Even though dominant individuals could have a monopoly in carcass occupation, they are not able to completely reject smaller individuals (i.e. subordinates) from the carcass. Thus, the dominant pair may ignore other individuals and show tolerance behaviour towards each other, as rejecting others from the carcass is costly for dominants, especially on large carcasses (Eggert and

Muller, 1992; Trumbo and Fiore, 1994; Komdeur et al., 2013). Such social interactions are temporarily stable, especially after larvae hatch, as subordinates often stay in the vicinity of the breeding pair or leave the carcass. Such social interaction is temporarily stable but not constant, because subordinates often leave the carcass. Subordinates not usually participate in offspring provisioning, at least if the dominant pair is present on the carcass, and the dominant pair may ignore the other pair to some extent. In contrast to the stable social groups in some eusocial insects (Nowak et al., 2010; Meunier, 2015), such groups will break up when reproduction ends and even before. We agree with you that social interaction in communal groups and its evolutionary role is important for understanding the carry-over effects and this should be mentioned for future direction (Lines 622-634).

“Our study on burying beetles offers a novel perspective to our understanding of communal breeding and shows no experimental evidence for short term effects of communal breeding on parental investment but a significant effect on future fitness, and a sex difference in these carry-over effects. Studies based on the short term effects are insufficient in understanding the evolution of group breeding. Living and breeding in social groups may generate positive or negative effects for each individual involved in the short term. Some of these effects (e.g. physiological changes and prior breeding experience) arising from group breeding can mask some direct, short-term fitness benefits. But, they may have a long-lasting effect on an individual's future fitness and its adaptation to complex environmental conditions. Future research on the evolution of social behaviour in animals should investigate the importance of the carry-over effects of group breeding, and sex-differences in these carry-over effects. Further study should investigate whether such carry-over effects on individual fitness differ between individuals of the same sex, and between sexes, thereby selecting for variation in behaviour.”

References:

- Eggert A.K., Müller J.K. (1992) Joint breeding in female burying beetles. Behavioral Ecology and Sociobiology, 31: 237-242.**
- Eggert A.K., Muller J.K. (2011) Timing of oviposition enables dominant female burying beetles to destroy brood-parasitic young. Animal Behaviour, 82(6): 1227-1233.**
- Eggert A.K., Sakaluk S.K. (2000) Benefits of communal breeding in burying beetles: a field experiment. Ecol. Entomol. 25(3): 262-266.**
- Komdeur J., Schrama M.J.J., Meijer K., Moore A.J., Beukeboom L.W. (2013) Cobreeding in the burying beetle, *Nicrophorus vespilloides*: Tolerance rather than cooperation. Ethology, 119(12): 1138-1148.**
- Meunier J. (2015) Social immunity and the evolution of group living in insects. Phil. Trans. R. Soc. B 370: 20140102.**
- Müller J.K., Braunisch V., Hwang W., Eggert A.K. (2007) Alternative tactics and individual reproductive success in natural associations of the burying beetle, *Nicrophorus vespilloides*. Behavioral Ecology, 18(1): 196-203.**
- Nowak M.A., Tarnita C.E., Wilson E.O. (2010) The evolution of eusociality. Nature, 466: 1057-1062.**
- Trumbo S.T., Fiore A.J. (1994) Interspecific competition and the evolution of communal breeding in burying beetles. Am. Midl. Nat. 131: 169-174.**

Aside from the content-based edits to the Introduction and Discussion, this paper needs substantial editing for clarity and copy-editing throughout. Some examples are included in the line-by-line comments below, but I omitted quite a few given how many there were.

Thank you very much the comment. We have now responded to all specific comments and revised the manuscript throughout according to the comments raised by reviewer 1.

Line specific comments:

Line 49: It is not clear why subjecting individuals of varying sizes to communal or non-communal breeding would provide insight into carry over effects – I imagine there are quite good reasons to expect that social experience would differ between large/small beetles, so perhaps a short plug here that puts this into context for non-beetle people – e.g., “the impact of social environment may be

expected to affect individuals of varying competitive environment differently . . . we took advantage of the known size-based competitive ability of *N. vespilloides* to test . . .”

Thank you very much for the comments. We subjected individuals of varying sizes to communal breeding in order to create a size-dependent dominance hierarchy in such breeding groups. In burying beetles, an individual’s competitive ability is closely linked with carcass monopolization and reproductive success, while adult body size is a determinant factor influencing competitive success (Otronen, 1988; Muller et al., 1990; Safryn and Scott, 2000). In communal groups of burying beetles, larger individuals are found to have advantages in carcass access and become dominant in reproduction, while smaller individuals are subordinates that have restricted access to the carcass and may gain a small proportion of reproduction (Eggert et al., 2008; Muller et al., 1990; Eggert and Muller, 2011). Thus, in our study, we could create a communally breeding event with a size-dependent dominance, by selecting two pairs (large versus small) of individuals that differed in body size (Komdeur et al., 2013; Safryn and Scott, 2000). Because burying beetle’s body size has been found to influence its parental effort (Pilakouta et al., 2015), we also subjected individuals of varying sizes (large and small individuals) to non-communal breeding (i.e. pair breeding) so as to control the effect of size on parental investment. We have now revised the study aims in our revised manuscript (Line 35-39):

“We subjected individuals to communal or non-communal breeding (i.e. pair-breeding) during their first breeding event, and to non-communal breeding during their second breeding event, and measured parental effort and reproductive outcome. In such communal groups, large individuals became dominant that had a monopoly in carcass and reproduction, while small individuals (i.e. subordinates) were restricted to access the carcass.”

We also now clarified the creation of communal and non-communal breeding groups, as well as controlling for effects of body size, in the methodology of our revised manuscript (Line 207-220):

“In burying beetles, adult body size affected an individual’s competitive ability, while this was closely linked with carcass monopolization and reproductive success [65,74,75]. To create the communally breeding treatment, one large pair (dominants) and one small pair (subordinates) of beetles (n = 19) were selected, and we ensured that the two pairs of beetles were unrelated and differed approximately 10 % in body size, because the likelihood of stable dominance hierarchy establishment in carcass use was enhanced when size difference between opponents was larger [64,72]. Other studies of ours indicated that the double-pair breeding treatment could create a communal breeding event with a size-dependent dominance hierarchy in carcass monopolization [76], which was accordance with previous studies about communal breeding in burying beetles [63,65,77]. For these beetles, an individual’s body size has also been found to influence its parental investment towards current brood [78]. Thus, we randomly selected one pair of beetles from small-pair (n = 10) or large-pair (n = 10) groups to create the non-communal breeding treatment, which could control the effect of body size on individual behaviour (i.e. parental investment) [78,79]. Just prior to creation of communal and non-communal breeding, each individual was weighed (accuracy: 0.0001 g).”

References:

- Eggert A.-K., Otte T., Muller J.K. (2008) Starving the competition: a proximate cause of reproductive skew in burying beetles (*Nicrophorus vespilloides*). Proc. R. Soc. B, 275: 2521-2525.**
- Eggert A.K., Muller J.K. (2011) Timing of oviposition enables dominant female burying beetles to destroy brood-parasitic young. Animal Behaviour, 82(6): 1227-1233.**
- Muller J.K., Eggert A.K., Dressel J. (1990) Intraspecific brood parasitism in the burying beetle, *Nicrophorus vespilloides* (Coleoptera: Silphidae). Animal Behaviour, 40(3): 491-499.**
- Otronen M. (1988) The effect of body size on the outcome of fights in burying beetles (*Nicrophorus*). Ann. Zool. Fennici. 25: 191-201.**
- Pilakouta N., Richardson J., Smiseth P.T. (2015) State-dependent cooperation in burying beetles: parents adjust their contribution towards care based on both their own and their partner’s size. J. Evol. Biol. 28(11): 1965-1974.**

Safryn S.A., Scott M.P. (2000) Sizing up the competition: Do burying beetles weigh or measure their opponents? J. Insect Behaviour, 13(2):225.

Line 62-63: How often do you expect for individuals of this species to have multiple breeding attempts in the wild, given how limited/ephemeral the breeding resource is? If it is reasonably rare – realistically, how important may carry-over effects be for generating variation for selection to act on?

Thank you very much for the comment. In this species, individuals may have multiple breeding attempts in the wild, and we have now clarified the importance of studying carry-over effects of communal breeding in our revised manuscript (Line 139-154, also see response to the reviewer above):

“For burying beetles, the breeding resource (i.e. carcasses) is ephemeral and limited, which largely determines an individual's breeding opportunity during a given time frame, as well as its reproductive strategy [66,70]. While individuals that breed in communal groups may, at least in the short term, achieve a reduced or no fitness benefit in reproduction, compared to individuals that breed alone or as pairs, breeding communally is still a better option for burying beetles than not breeding at all (i.e. ‘the-best-of-a-bad-job’ strategy), if they have no another breeding opportunity because of the constraints of breeding resources [59,64,70]. These beetles are opportunistic breeders, and may have multiple breeding attempts over their lifetime in a benign environment (e.g. a high resource availability and a low interspecific competition pressure) [70,71]. It is likely that breeding communally among individuals evolves as an adaptive strategy, which may largely reduce the reproductive costs in the adverse environmental conditions but potentially improve the overall fitness benefits across breeding bouts and even over lifetime. Thus, it is worth studying the carry-over effects of communal breeding on parental behaviour and fitness across breeding attempts in burying beetles, and it is also important to examine whether and how such effects, concurrent with the short-term benefits on fitness, act on the selection for group breeding and its adaptation to ecological environments.”

Line 76-77: “to find food”? Sentence reads a bit awkward.

We have now revised this sentence in our revised manuscript (Line 63-65):

“Social groups of conspecifics occur in most animal species, for example because individuals form temporary associations due to aggregated resources, or because individuals could forage as groups (i.e. group hunting) [1,2].”

Line 75-92: repetitive use of “cooperative and communal breeding” – may reduce the wordiness to state once and call “group breeding” or just “communal breeding” after first use.

Thank you for this comment. We have now revised this to reduce the wordiness in our revised manuscript (Line 67-75):

“In such breeding groups, group members may gain kin-selected benefits (i.e. the indirect fitness benefits individuals gain through helping their relatives reproduce) and various aspects of direct fitness benefits (e.g. non-kin reciprocity and mutualism) from group living [1,6 – 8], although they may or may not benefit in terms of direct reproductive success [9 – 12]. A large number of studies have focused on the short-term fitness benefits of kin-selected and direct benefits of group breeding for each individual involved, such as the joint defence of territories and resources against intruders [8,13 – 15], reduced workloads during parental care [16 – 20], higher reproductive success [2,21,22], and higher survival and longevity [23].”

Line 94: It would be helpful to have the definition of carry-over effects at first mention (i.e., previous paragraph). Also, by this definition, it is hard to pinpoint what is important about carryover effects – by the examples provided below, everything could have a carryover effect, and if everything matters, then none of it does. Would we expect for carry-over effects to impact some traits more than others, or have a disproportionate effect on fitness?

Thank you very much for your comments. We have now clarified the definition of carry-over effects at first mention and its importance for understanding the evolution of social behaviour in our revised manuscript (Line 82-89 and 93-102). It also could be expected that the carry-over effects have disproportionate impacts on some traits under some circumstances. For example, an individual's dominance hierarchy in some groups may have a significant carry-over influence on its future reproduction and parenting behaviour. For example, dominants exhibit higher immunity and good nutritional state, while subordinates are characterized by a lower weight gain and reduced immune response. Such differences in immunity and nutritional state may influence individual parental effort and reproductive capacity (Pettinger et al., 2011; Steiger et al., 2012).

“Carry-over effects refer to a potential consequence wherein an individual's prior conditions (e.g. physical state of body and prior experience) could exert an impact on its subsequent performance, such as parenting behaviour and reproductive success. Such effects can be due to differences in access to resources, or variation in the resource allocation at one stage in life to another stage, e.g. within and across breeding seasons or between reproductive events [30 – 33]. Under these circumstances, individuals may trade off their efforts in self-maintenance and reproduction between stages of the life cycle, due to variation in physiological condition (e.g. body weight and immunocompetence) [27,34,35]. ...

In some breeding groups, dominance hierarchy may generate difference in body condition between individuals [42,43]. For example, high-ranked, dominant individuals often monopolize high-quality resources or the access to resources in social groups. This monopolization of resources by dominants may result in lower-ranked individuals (i.e. subordinates) utilizing lower-quality resources or having restricted access to resources. This differential access to resources may therefore result in carry-over effects for dominants and subordinates, e.g. subordinates suffering low survival rates and low reproductive success in the future [42 – 44]. More, a gain of breeding experience may have carry-over, disproportionate effects on fitness and reproductive performance [45,46]. For example, experienced individuals with prior breeding have increased parental care towards the current brood compared to inexperienced individuals [45].”

References:

- Pettinger A.M., Steiger S., Muller J.K., Sakaluk S.K., Eggert A.-K. (2011) Dominance status and carcass availability affect the outcome of sperm competition in burying beetles. *Behavioral Ecology*, 22(5): 1079-1087.**
- Steiger S., Gershman S.N., Pettinger A.M., Eggert A.-K., Sakaluk S.K. (2012) Dominance status and sex influence nutritional state and immunity in burying beetles *Nicrophorus orbicollis*. *Behavioral Ecology*, 23(5): 1126-1132.**

Line 94-95: By your definition, isn't a carry-over effect how an individual's experience impacts things down the road? For this reason, it makes the definition circular to say that experience can lead to carry over effects on individual conditions based on experience.

Thank you very much for this comment. We have now revised the definition of carry-over effects to avoid the circular mention in our revised manuscript (Line 82-86):

“Carry-over effects refer to a potential consequence wherein an individual's prior conditions (e.g. physical state of body and prior experience) could exert an impact on its subsequent performance, such as parenting behaviour and reproductive success. Such effects can be due to differences in access to resources, or variation in the resource allocation at one stage in life to another stage, e.g. within and across breeding seasons or between reproductive events [30 – 33]. ”

Line 98: “within and across seasons or between reproductive events”

Corrected.

Line 99 – 100: sentence awkward, revise. E.g., instead of “different levels of physical states in terms of energy reserves” – “variation in condition”.

Thank you very much for the comment. We have now corrected this sentence in our revised manuscript (Line 86-89):

“Under these circumstances, individuals may trade off their efforts in self-maintenance and reproduction between stages of the life cycle, due to variation in physiological condition (e.g. body weight and immunocompetence) [27,34,35].”

Line 114 – 116: Are the authors just stating that life-history tradeoffs are just classic examples of carry-over effects? If so, simplify, it is not clear what is meant by “manifested” here and the sentence reads a bit circular.

We thank very much for this comment and have now rephrased this sentence in our revised manuscript (Line 105-107):

“The underlying impact of allocation of parental effort to current reproduction on subsequent reproductive performance and survival can also be viewed as a special type of carry-over effect [32,44,48].”

Line 118: communal/cooperative breeders compared to “non-group breeding species” here. It would really streamline the introduction to refer to communal/cooperative ones as “group breeding”. I recommend saying specifically what you are talking about at first – cooperative/communal – and then say “hereafter group breeding” – until the manuscript focuses down onto communal breeding (otherwise quite repetitive to read “communal and cooperative breeding” every sentence).

Thank you very much for the comment and your recommendation. We have now rephrased this in our revised manuscript to avoid repetition (for example, Line 65-67 and 107-108).

“More advanced forms of social groups may occur among conspecifics, for example in cooperatively and communally breeding systems (hereafter group breeding), where multiple individuals live and reproduce together to rear a single brood or litter [3 – 5]. ... Compared to non-group breeding species, group breeding species are particularly interesting to study trade-offs between current and future reproduction...”

Line 141: remove “the” before social dominance.

Corrected.

Line 147: typo after “group”

Corrected.

Line 160: what is the direction in which they adjust parental investment? Will provide context for why you are predicting a directional relationship on line 161.

Thank you very much for this comment. We have now clarified the direction in our revised manuscript (Line 167-169):

“Given that individuals have been shown to reduce their efforts in parental investment (e.g. shared investment in carcass preparation and provide less post-hatching care towards offspring) during communal breeding [49,55], ...”

Line 178-181: How are you extricating the potential effect of differing social environment that occurred pre-breeding from the social environment after breeding initiated? I.e., says that individuals were kept in varying groups sizes prior to the experiment (perhaps just requires a quick rewording).

Thank you very much for the comment. In our setup used for beetle rearing, four to six adult beetles were kept with a small and similar size of group and were reared in the absence of a breeding resource, which enabled to reduce the potential effect of social environment occurring at pre-breeding stage on individual behaviour after breeding initiated. We have now clarified this in our revised manuscript (Line 185-191):

“During the entire rearing and experimental period, four to six adult beetles of the same sex were kept in plastic boxes (23 × 19 cm and 12.5 cm high), at 20 ° C with a 16:8 h light to dark photoperiod, and fed with mealworms twice a week. In the setup used for beetle rearing, beetles were kept with a small and similar size of group and were reared in the absence of a breeding resource (e.g. mouse carcass), which enabled to reduce the potential effect of different social environment occurring at pre-breeding stage on individual behaviour after breeding initiated.”

Line 189-191: revise this sentence to be in the active voice.

We thank very much for this comment and have now rephrased this sentence in the active voice (Line 199-201):

“We selected sexually mature adult beetles, aged between 10 – 14 days at post-eclosion, for our experiments. Each individual was sexed according to morphological traits, and body size was measured (pronotum width; accuracy: 0.01 mm) [73].”

Line 191: “and body size as measured”.

We have now corrected this in our revised manuscript.

Line 202: be consistent with how each treatment is named – e.g., communal treatment throughout rather than “double pair breeding system”.

Thank you very much for the comment. We have now revised the naming of this term throughout the revised manuscript (Line 207-215).

“In burying beetles, adult body size affected an individual's competitive ability, while this was closely linked with carcass monopolization and reproductive success [65,74,75]. To create the communally breeding treatment, one large pair (dominants) and one small pair (subordinates) of beetles (n = 19) were selected, and we ensured that the two pairs of beetles were unrelated and differed approximately 10 % in body size, because the likelihood of stable dominance hierarchy establishment in carcass use was enhanced when size difference between opponents was larger [64,72]. Other studies of ours indicated that the double-pair breeding treatment could create a communal breeding event with a size-dependent dominance hierarchy in carcass monopolization [76], which was accordance with previous studies about communal breeding in burying beetles [63,65,77].”

Line 204 -205: This is a test of assumption, so it is a result – should go in results.

Thank you very much for the comment. We agree with you that this is a test of assumption and the result further indicated that the double-pair breeding treatment could create a communal breeding event with a size-dependent dominance hierarchy in carcass monopolization (e.g. large pairs of individuals spent significantly more time on the carcass compared to small individuals). We have now moved this result in the section of results (Line 351-355):

“In the communal groups, we found that the larger pairs of individuals became dominant and significantly monopolized the access of carcass, whereas the smaller individuals were subordinates that spent less time on the carcass compared with larger individuals during the entire breeding period (mean ± SE of large vs. small individuals: 58.96 ± 2.95 vs. 7.22 ± 1.33%; $\chi^2 = 315.65$, p

< 0.001; Table 1)."

Line 217-218: mouse characteristics described twice in this section of methods.

Thank you very much for the comment. We have now deleted this repetitive description about mouse characteristics in the section of methods of our revised manuscript (Line 230-231):

"All beetles were placed in breeding boxes, and a mouse carcass, thawed before the experiment, was introduced as breeding resource.."

Line 227-228: Why do we care about carcass burial? The introduction should provide a clear roadmap of what general factors are needed to answer the question, and then the methods tie each measure directly back to the framework – e.g., "need to measure parental care behavior" – so measured these things that are important to burying beetle care behavior.

Thank you very much for the comment. For burying beetles, we measured degree of carcass burial over observational time, which could be used for assessing the parental performance of groups. It is suggested that there may occur an improved performance in carcass preparation in communal breeding, compared to non-communal breeding (i.e. pair breeding), because multiple individuals in communal groups may have increased combined efforts in carcass burial, with faster rates and higher degree of carcass burial, compared to pairs. We have now clarified the factors that are investigated to answer the specific questions in our revised manuscript (Line 157-169).

"To investigate the short-term fitness implications of communal breeding, we measured parental performance for each individual (i.e. time spent providing care on the carcass and weight change) and of groups (i.e. carcass burial and the total amount of parental investment by groups), as well as reproductive outcome of groups (i.e. brood size and larval weight), in communal and non-communal breeding. ... To examine the carry-over effects of communal breeding experimentally, we investigated the impact of prior breeding experience, i.e. communal versus non-communal breeding, on future parental performance (e.g. carcass burial for each pair, an individual's time spent providing care and weight change) and reproductive outcome for each individual and for each pairs in a subsequent pair-breeding event."

Also, we have now clarified the factors that were measured and their importance in assessing parental behaviour in the methodology of the revised manuscript (Line 245-268):

"As a proxy for the joint parental performance of breeding groups or breeding pairs in burying beetles, the degree of carcass burial was estimated according to the fraction of the mouse above the ground and carcass roundness [85]. In communal breeding events, dominant individuals largely monopolized the carcass, while both dominant and subordinate females were able to reproduce offspring by laying eggs surrounding the carcass [65,66,75]. This resulted in selection for females to lay their eggs earlier (which hatch earlier) and synchronously lay eggs with other female cobreeders [66,72,77]. Thus, we examined the competitive interaction in reproduction and reproductive strategies between communal and non-communal groups, by recording the timing of reproductive performance and offspring development. As measures for timing of reproductive performance, we recorded egg-laying time (the time from the start of the experiment until the onset of egg laying) and larvae-hatching time (the time from the start of the experiment until the onset of larvae hatching). We defined 'the onset of egg laying' as the onset observational time when some eggs were found at the bottom of the box, and defined 'the onset of larvae-hatching' as the onset observational time when some of larvae newly hatched and did not enter into 1st instar. As measures for offspring development and reproductive performance, we used larvae-dispersing time (the time from the start of the experiment until the onset of larvae dispersing), brood size (number of larvae) and average larval weight (total weight of larvae/brood size) at the larval dispersal (i.e. larvae dispersed from the carcass). Also, we recorded weight change during breeding $([final\ weight - initial\ weight]/initial\ weight)$ and survival at the end of the experiment of adult individuals, as parameters for parental effort during the entire breeding period and reproductive costs for individuals, respectively [49,57].

After the first reproductive round, surviving beetles were kept individually for five days in rearing boxes (10×5 cm and 8.5 cm high) and not fed by any food (i.e. mealworms) for subsequent experimentation. We did not feed beetles during this period to avoid the potential effect of food consumption and weight change on an individual's behaviour in the subsequent period."

Line 248-251: why were the mates switched? This seems to complicate the matter, since individuals were size-matched for the first treatment and then not-size matched for the second – this seems like it may confound any age-specific effects versus.

Thank you very much for the comment. Yes, females and males were not size matched as pair for the second experiment to avoid prior familiarity. In burying beetles, while some studies suggested that individuals could adjust their parental investment according to the body size of their own and their partners within pairs, we have included the body size of each individuals as an explanatory factor in our analyses in order to reduce the size effect. Also, in our study, age is not a significant effect that should be considered to determine individual behaviour and parental effort. We have now discussed this in our revised manucrypt (Line 527-544):

*“Previous studies have demonstrated age-related differences in reproductive effort and reproductive allocation between breeding attempts [90,100 – 102]. In *N. orbicollis*, older individuals have a higher level of investment to current reproduction (e.g. produce larger broods and consume less of the carcass) compared to younger individuals, because older individuals with a low residual reproductive value have limited future reproductive opportunities [90,101]. These previous studies also highlight the impact of individual reproductive restraint and senescence on individual reproductive effort [90,100]. Moreover, more recent work revealed an interplay between age and previous breeding experience on reproductive investment in burying beetles [62,102]. For example, female *N. vespilloides* that have invested more in reproduction during earlier reproductive attempts show a decline in reproductive investment, but this negative effect of previous reproductive investment is observed only in older females [62]. Also, having prior breeding experience as communal breeders is associated with changes in physiological states, as well as the gaining of social skills [35,63,103]. These state-associated changes that may influence individual reproductive residual value should be considered, as they could determine an individual's reproductive effort for future reproduction, probably due to physiological constraints and the lack of experience. Thus, we suggest that the effects of communal breeding on fitness are associated not only with short-term costs and benefits, but also with carry-over effects on future fitness [32,59,67,104], albeit such effects are more pronounced for dominant males than for dominant females [49,82,97,98].”*

Line 258: “providing care” rather than “spending care”

We thank very much for the comment and have now corrected this.

Line 259: edit for clarity.

Thank you very much for the comment. We have now clarified this in our revised manuscript (Line 258-264):

“As measures for offspring development and reproductive performance, we used larvae-dispersing time (the time from the start of the experiment until the onset of larvae dispersing), brood size (number of larvae) and average larval weight (total weight of larvae/brood size) at the larval dispersal (i.e. larvae dispersed from the carcass). Also, we recorded weight change during breeding ($[\text{final weight} - \text{initial weight}] / \text{initial weight}$) and survival at the end of the experiment of adult individuals, as parameters for parental effort during the entire breeding period and reproductive costs for individuals, respectively [49,57].”

Line 281: I am unclear how parental behavior was measured – it sounded like broods were checked briefly twice a day and their activities at that moment in time were noted, and then the proportion of

those times they were providing care were used as a proxy for care provided throughout. Where does the “total amount of time” come from, and how does it differ from “parenting investment time”, line 290?

Thank you very much for the comment. We checked beetle's activity on and around the carcass two times per day, while its activity was checked at that moment. In our study, each check lasted for 30 seconds so as to ascertain that beetles were providing care when they were present on the carcass or they were not providing care when they were absent on the carcass. This also reduced the counts of any accidental events. For example, beetles can simply wander on the carcass but without providing care. In our study, we defined individual parental investment by calculating the proportion of times each individual was found to provide care on the carcass. Also, at the pair (or group) level, we defined the total amount of time spent providing care on the carcass by calculating the porportion of total times that dominants or subordinates (or all individuals) were found to provide care on the carcass. 'Time spent providing care on the carcass for each individual' and 'total amount of time spent by pairs or groups' could be used to investigate the parental investment time at the individual level, as well as at the pair (or group) level. These two different measures are both necessary for assessing the parental efforts in groups. We have now revised these in our revised manuscript (Line 232-245):

“During the entire reproductive period (from the onset of experimentation until larvae dispersal), beetle activity on and around the carcass was checked twice daily by visual inspection at 9:00 and 17:00. Each check lasted for 30 seconds so as to make sure that beetles were providing care or not when they were present or absent on the carcass, which could also avoid the counts of any accidental events, for example, beetles were wandering on the carcass without providing care [70]. We recorded parental care behaviour as when an individual was providing indirect care in carcass guarding and maintenance on the surface of the carcass, and was providing direct care (i.e. larvae provisioning inside the carcass)[63,81,82]. For each individual, we defined individual parental investment time by calculating the proportion of times that each individual was found to provide care on the carcass (i.e. carcass preparation and offspring provisioning) during the entire observation period [49,83,84]. We also defined the total amount of time spent on parental care by breeding groups (including communal and non-communal groups) by calculating the proportion of total times that all individuals in groups were found to provide care on the carcass.”

Line 299 – 313: I see that the stats are included in Table 1a, but where are the means for these statements? E.g., what was burial time for a communal vs non-communal group? This comment applies to all the methods.

Thank you very much for the comment. We have now showed statistics in our revised manuscript and showed significance in figures (Line 346-351 and 390-395). In our study, we did not measured burial time for communal versus non-communal groups, while we just measured and compared burial degree at different observational times.

“Our results showed that communal groups had a better performance in carcass burial at the initial stage of the experiment, as communal groups were significantly faster in burying carcasses than non-communally breeding groups at 36, 44 and 60 hours (36-h: $t = 3.16$, $p = 0.02$; 44-h: $t = 3.25$, $p = 0.01$ and 60-h: $t = 7.28$, $p < 0.001$; figure 2; supplementary materials, table S1). After ca. 68 hours, there was no difference in the degree of carcass burial between communal and non-communal groups (figure 2; table S1).

Prior breeding experience of individuals had a significant effect on parental performance in carcass burial (table 2 and S3). Specifically, females that originated from communal groups had lower rates of carcass burial than females from non-communal groups (experience: $\chi^2 = 26.41$, $p < 0.001$; experience \times time: $\chi^2 = 7.67$, $p = 0.01$; figure 4a; table 2 and S3), whereas males that originated from communal and non-communal groups had similar rates of carcass burial (experience: $\chi^2 = 9.09$, $p = 0.005$; experience \times time: $\chi^2 = 2.34$, $p = 0.13$; figure 4b; table 2 and S3). Moreover, the interaction of prior breeding experience of females and males had a significant

influence on the rates of carcass burial, as a higher rate of carcass burial was observed in pairs where females originated from non-communal groups and males originated from communal groups, compared with other pairs (female experience × male experience × time: $\chi^2 = 4.84$, $p = 0.03$; figure S2; table S3)."

Line 309: Editing for active voice and clarity throughout, simplify the language. E.g., this sentence could read: "Individuals in communal groups spent more time on the carcass than non-communal groups".

We thank very much for the comment and have now rephrased this sentence in our revised manuscript (Line 364-366):

"Moreover, at the group level, communal groups had a significantly higher total amount of time spent by all individuals than non-communal groups (mean \pm SE: 33.09 ± 1.78 vs. $27.33 \pm 1.57\%$; table 1A)."

Line 314: If it was unknown whether weight change was correlated with variation in parental investment, why was weight change described as a metric to track parental investment earlier in the manuscript? Is this a test of assumption or a result, and if so, how does it fit into the overall framework of the manuscript?

Thank you very much for the comment. In some studies on burying beetles, an individual's weight change during the entire breeding period was closely linked with its parental investment to current brood, while weight change was a mixed consequence of reproduction because beetles could gain weights by feeding from the carcass (Creighton et al., 2009; Trumbo and Xihani, 2015; Wang et al., 2021). However, weight changes could also be a proxy for parental efforts in reproduction for each individual. We have now clarified this in our revised manuscript (Line 262-264):

"Also, we recorded weight change during breeding ($[\text{final weight} - \text{initial weight}] / \text{initial weight}$) and survival at the end of the experiment of adult individuals, as parameters for parental effort during the entire breeding period and reproductive costs for individuals, respectively [49,57]."

References:

- Creighton J.C., Heflin N.D., Belk M.C. (2009) Cost of reproduction, resource quality, and terminal investment in a burying beetle. Am. Nat. 174(5): 673-684.**
- Trumbo S.T., Xihani E. (2015) Influences of parental care and food deprivation on regulation of body mass in a burying beetle. Ethology, 121(10): 985-993.**
- Wang W., Ma L., Versteegh M.A., Wu H., Komdeur J. (2021) Parental care system and brood size drive sex difference in reproductive allocation: an experimental study on burying beetles. Front. Ecol. Evol. 9: 739396.**

Line 318: What was the source of mortality (if possible to infer)?

Thank you very much for the comment. In communally breeding burying beetles, an individual's mortality was jointly influenced by an intense competition between individuals (e.g. severe flights) and its efforts in parental care (i.e. the costs of reproduction). In the communally breeding events where individuals varied in size and were introduced with a large size of carcass, there was a low probability of showing aggressive behaviour towards each other. However, in some groups, dominants could often show more aggressive towards subordinate individuals, and even killed them, especially for subordinate males. We have now stated this in our revised manuscript (Line 485-500):

"Both males and females that originated from communal breeding experienced higher mortality compared with individuals from non-communal breeding. This may be due to that each individual incurs a high cost in reproduction and survival from communally breeding, or that an enhanced allocation to reproduction comes to at the cost of high mortality for individuals that communally bred [57,62,90]. In communal groups, the establishment of a dominance hierarchy is suggested to

reduce the extent of aggressive interaction, while injuries are often observed [64,70]. It is likely that these injuries affect an individual's future reproduction and increase the risk of death. In this scenario, a microbe-rich soil environment might deteriorate some opportunistic infections arising from fight wounds, thereby increasing the rate of mortality [62]. For each individual, dominance status is closely associated with fecundity and nutritional state [35,63]. For example, in *N. vespilloides*, a difference in fecundity between females is due to their dominance status, because limited access to the carcass leads to low fecundity by causing nutritional deficiencies [63]. Such effects in fecundity and nutritional state are also accompanied with some negative changes in physiological state and immunity, which may have long-lasting influences on individual fitness benefits in the future, e.g. high mortality and short lifespan for individuals that originate from communal groups.”

Table 1: There are quite a lot of results, but I did not find the table format to be particularly accessible (especially given the number of footnotes needed to describe what the variables mean relative to the statistics). May be more straightforward to include these statistics in the written methods, and then have a summary table of them all somewhere – or at least make a table that makes the comparison/directionality of the results more straightforward.

We thank very much for the comment and have now re-organized the table format in our revised manuscript and add a new summary table showing the comparison of the results (Table S2) in the supplementary materials.

Line 331: How was a rate of carcass burial determined? And what is the significance of the rate of carcass burial? Need to add to methods.

Thank you very much for the comment. Here, we did not calculate the rate of carcass burial, but the degree of carcass burial over time. The degree of carcass burial over observational time were used for assessing the parental performance of groups. We have now revised this measure in our revised manuscript (Line 245-247):

“As a proxy for the joint parental performance of breeding groups or breeding pairs in burying beetles, the degree of carcass burial was estimated according to the fraction of the mouse above the ground and carcass roundness [85].”

Line 339: As currently written, there are contradictory statements in this sentence.

We thank very much for the comment and have now rephrased this sentence in our revised manuscript (Line 401-408):

“The prior breeding experience of focal individuals and that of their partners jointly influenced their parental investment time (table 1B). At the individual level, females and males that originated from communal or non-communal groups spent a similar parental investment time on the carcass (mean \pm SE of females: 61.15 ± 5.93 vs. 78.92 ± 6.84 %; of males: 38.56 ± 7.50 vs. 42.45 ± 5.91 %; figure 4c; table 1B and S2) and had similar weight changes during breeding (table 1B and S2). Regardless of their own prior breeding experience, both females and males spent significantly more time on the carcass when their partners originated from communal groups than from non-communal groups (mean \pm SE of females’ partner: 77.19 ± 4.15 vs. 54.75 ± 8.74 %; of males’ partner: 46.99 ± 6.44 vs. 29.12 ± 7.10 %; figure 4c, d; table 1B and S2).”

Line 343: this sentence should start the paragraph and then go into details.

We thank very much for the comment and have now moved this sentence as the start of the paragraph in our revised manuscript (Line 400-408):

“The prior breeding experience of individuals and their partners jointly influenced their parental investment time (table 1B). At the individual level, females and males that originated from communal or non-communal groups spent a similar parental investment time on the carcass (mean \pm SE of

females: 61.15 ± 5.93 vs. 78.92 ± 6.84 %; of males: 38.56 ± 7.50 vs. 42.45 ± 5.91 %; figure 4c; table 1B and S2) and had similar weight changes during breeding (table 1B and S2). Regardless of their own prior breeding experience, both females and males spent significantly more time on the carcass when their partners originated from communal groups than from non-communal groups (mean \pm SE of females' partner: 77.19 ± 4.15 vs. 54.75 ± 8.74 %; of males' partner: 46.99 ± 6.44 vs. 29.12 ± 7.10 %; figure 4c, d; table 1B and S2)."

Line 362: typo - start of sentence "with the regardless"

We thank very much for the comment and have now corrected this sentence in our revised manuscript (Line 429-432):

"Regardless of male's prior breeding experience, females that originated from communal and non-communal groups had similar reproductive outcomes (mean \pm SE of brood size: 22.61 ± 2.15 vs. 25.08 ± 1.28 ; of average larval weight: 0.1389 ± 0.0067 vs. 0.1333 ± 0.0053 g; table 2 and S3)."

Line 364: "outcomes" instead of "outcome"

Corrected.

Line 384: Alternatively, you don't know what their options are – e.g., if that one big carcass is literally the only one they can breed on, there is not a cost of breeding on it since they would have nothing if they didn't.

Thank you very much for the comment. We agree with you that, in communal breeding, individuals would have nothing in reproduction if they did not communally breed on one carcass because of limited and ephemeral carcasses, which has been suggested that communally breeding is a 'best-of-the-bad' job strategy for burying beetles (Trumbo and Fiore, 1994; Scott, 1998; Komdeur et al., 2013). Some previous studies (including field and lab studies) found that individuals that bred in communal groups had no significant higher fitness benefits than individuals that bred as pairs (i.e. less *per capita* offspring was observed in communal breeding, compared with in non-communal groups), which could be considered as the cost of communal breeding in reproduction (Komdeur et al., 2013; Eggert and Muller, 1992; Muller et al., 2007). But if they do not breed communally and did not breed at all they have lower fitness in reproduction.

References:

- Eggert A.K., Müller J.K. (1992) Joint breeding in female burying beetles. *Behavioral Ecology and Sociobiology*, 31: 237-242.**
- Komdeur J., Schrama M.J.J., Meijer K., Moore A.J., Beukeboom L.W. (2013) Cobreeding in the burying beetle, *Nicrophorus vespilloides*: Tolerance rather than cooperation. *Ethology*, 119(12): 1138-1148.**
- Müller J.K., Braunisch V., Hwang W., Eggert A.K. (2007) Alternative tactics and individual reproductive success in natural associations of the burying beetle, *Nicrophorus vespilloides*. *Behavioral Ecology*, 18(1): 196-203.**
- Scott, M.P. (1998) The ecology and behavior of burying beetles. *Ann. Review Entomol.* 43: 595-618.**
- Trumbo S.T., Fiore A.J. (1994) Interspecific competition and the evolution of communal breeding in burying beetles. *Am. Midl. Nat.* 131: 169-174.**

Line 389: Are lighter larvae bad? I assume so, based on context, but to an outside reader, there are tradeoffs in development that could be compensated for later, and there is no reason provided for why we would expect for lighter offspring to be bad off the bat.

Thank you very much for the comment. Yes, lighter larvae at the larval dispersal are bad, because lighter larvae will become smaller individuals when they emerge as adults (Eggert et

al., 1998; Scott, 1998). Subsequently, smaller adults in body size may have no advantages in competitive ability (i.e. fighting ability and competing carcass with others), compared with larger adults (Otronen, 1988; Muller et al., 1990; Safryn and Scott, 2000). Thus, we could expect for lighter offspring future fitness to be bad. Also, we have now clarified the size effect on offspring fitness in the future in our revised manuscript (Line 458-464):

“Our results further suggest that such costs in reproduction have a negative influence on offspring fitness, i.e. lighter larvae produced in communal breeding, which may be due to a high level of group conflict over reproduction between individuals [66,68,72]. In burying beetles, it has been found that the weight of larvae that dispersed from the carcass was positively associated with their body size when they emerged as adults, while this subsequently determines an individual's competitive ability (i.e. fighting ability) and its advantages in resource monopolization and reproduction [65,74,75].”

References:

Eggert A.-K., Reinking M., Muller J.K. (1998) Parental care improves offspring survival and growth in burying beetles. *Animal Behaviour*, 55(1): 97-107.

Line 420: Do individuals need to gain more breeding experience? Does this species usually breed more than once in nature? Or do you expect that larger individuals are more likely to breed a second time?

Thank you very much for the comment. For burying beetles, having prior breeding experience is important for their future parental performance and fitness, for example, an improved parental performance and an enhanced sexual attractiveness for males (Creighton et al., 2009; Billman et al., 2014; Chemnitz et al., 2017; Richardson and Smiseth, 2020). This could benefit for each individual through direct energy saves for future reproduction, or the gaining of experience. While individuals in this species could breed more than once and suffer a low cost of reproduction under the laboratory conditions, they have to pay more costs in searching for a breeding resource and defending the resource against intruders (Ward et al., 2009; Cotter et al., 2010). Although for burying beetles, the breeding resource (i.e. carcasses) is unpredictable and limited which may cause communal breeding, however – albeit unpredictable and limited - breeding resources are readily available across the entire breeding season enabling individuals breeding multiple times during a breeding event. Individuals could have an improved group performance in carcass preparation and defence against intruders (mutualistic benefits; Sun et al., 2014; Liu et al., 2020). However, individuals that reproduce in communal groups may suffer reproductive costs compared to individuals that breed as pairs (mutualistic benefits; Eggery and Muller, 1992; Komdeur et al., 2013; Liu et al., 2020), although on the other hand, breeding communally is a better option than not breeding at all (i.e. ‘a-best-of-the-bad-job’ strategy) if they have no other breeding opportunity. Also, we expect that larger individuals have a high probability of breed multiple times, because they have a higher competitive ability compared with other smaller individuals if a breeding resource was found to utilize. This should be investigated in the future. Please also see the response to the reviewer 1 above.

References:

Ward R.J.S., Cotter S.C., Kilner R.M. (2009) Current brood size and residual reproductive value predict offspring desertion in the burying beetle *Nicrophorus vespilloides*. *Behavioral Ecology*, 20(6): 1274-1281.

Cotter S.C., Ward R.J.S., Kilner R.M. (2010) Age-specific reproductive investment in female burying beetles: independent effects of state and risk of death. *Functional Ecology*, 25(3): 652-660.

Line 433: Didn't the beetles have similar weights after the first breeding attempt? So there wasn't a test of the effect of weight gains on future reproduction? Please clarify.

Thank you very much for the comment. Yes, you are correct that beetles that bred in communal and non-communal groups have similar weight changes after the first breeding attempt. This means that it is likely that the carry-over effects of communal breeding on individual parental performance and fitness is not through the previous weight change. We have now clarified this suggestion in our revised manuscript (Line 519-527):

“As such, we suggest that individuals that reproduce in communal groups may have enhanced physical conditions and allocate more resources and parental efforts for future reproduction, because there occurs a limited resource availability for each individual and a high pressure of intraspecific competition. As argued above, we lack evidence for the effect of the energetic saves (i.e. weight gains) during previous reproductive events on an individual’s subsequent behaviour and future reproduction. This might mean that the carry-over effects of communal breeding on individual parenting behaviour and fitness is not simply determined by the parental investment to previous reproduction, but instead is associated with the individual’s prior experience [57,96].”

Line 436 (and discussion more broadly): How are you envisioning that experience is impacting this? It would be useful to hear a discussion of the mechanisms that would produce these results.

Thank very much for the helpful comment. We have now revised the impact of prior experience in the new version of our manuscript (Line 507-514 and 515-544; also see my response to the reviewer above):

*“These results may suggest that the improved future benefit for dominant males is due to dominant males being able to save more resources and gain more experience than males that breed as pairs [49,52,82,97,98]. Burying beetles could gain fitness benefits from access to carcasses during breeding, because access to resources (e.g. spend more time on the carcass, consume more of the carcass) while breeding may offset the energetic costs of reproduction, and individuals may save more resources and increase individual conditions (e.g. immunity) for future reproduction, such as weight gains and improved sexual attractiveness for males [57,59,63,90,99]. ... For burying beetles, an individual’s investment in the current reproduction is influenced by resource availability, as well as its prior experience [57,90]. For example, in *N. orbicollis*, females that reproduce on low-quality carcasses invest less in current reproduction and allocate more to future reproduction compared with females given high-quality carcasses, which may be due to the reproductive restraints by females [57,90]. As such, we suggest that individuals that reproduce in communal groups may have enhanced physical conditions and allocate more resources and parental efforts for future reproduction, because there occurs a limited resource availability for each individual and a high pressure of intraspecific competition. As argued above, we lack evidence for the effect of the energetic saves (i.e. weight gains) during previous reproductive events on an individual’s subsequent behaviour and future reproduction. This might mean that the carry-over effects of communal breeding on individual parenting behaviour and fitness is not simply determined by the parental investment to previous reproduction, but instead is associated with the individual’s prior experience [57,96]. Previous studies have demonstrated age-related differences in reproductive effort and reproductive allocation between breeding attempts [90,100 – 102]. In *N. orbicollis*, older individuals have a higher level of investment to current reproduction (e.g. produce larger broods and consume less of the carcass) compared to younger individuals, because older individuals with a low residual reproductive value have limited future reproductive opportunities [90,101]. These previous studies also highlight the impact of individual reproductive restraint and senescence on individual reproductive effort [90,100]. Moreover, more recent work revealed an interplay between age and previous breeding experience on reproductive investment in burying beetles [62,102]. For example, female *N. vespilloides* that have invested more in reproduction during earlier reproductive attempts show a decline in reproductive investment, but this negative effect of previous reproductive investment is observed only in older females [62]. Also, having prior breeding experience as communal breeders is associated with changes in physiological states, as well as the gaining of social skills [35,63,103]. These state-associated changes that may influence individual reproductive residual value should be considered, as they could determine an individual’s reproductive effort for future reproduction, probably due to physiological constraints and the lack of experience. Thus, we suggest that the effects of communal breeding on fitness are associated not*

only with short-term costs and benefits, but also with carry-over effects on future fitness [32,59,67,104], albeit such effects are more pronounced for dominant males than for dominant females [49,82,97,98].”

Reviewer: 2

Comments to the Author(s)

Overall this was a good manuscript. I liked the experimental design and the results were intriguing. Some clarifications need to be made in the Materials and Methods section. See the attached for the specific comments. Do you think that you would find similar results in other invertebrate models?

Thank you very much for the positive comment. We are confident that similar results may have been found in other invertebrate models and vertebrate models. There does occur communal breeding in many invertebrate species, like spiders, paper wasps and some beetles, as well as in vertebrate species (such as fish, mammals and birds). Our experimental findings show that social experience (including communal breeding experience) is an important determinant factor influencing individual's reproductive allocation between current and future reproduction. Such social experience is involved in many aspects, including differential access to the resource in groups, dominance status and sex. Broadly, it could be suggested that social experience could be a driver influencing a trade-off in parental investment, not only for invertebrate but also for vertebrate species. We have now clarified these points in the methodology section of our revised manuscript, and have now added the further discussion in our revised manuscript (Line XX, also see our response to reviewer 1).

Page 7 lines 191-196: I was a little confused, were the beetles bred first before being put in experiment 1? So, they were paired or grouped up after insemination. Why?

Thank you very much for the comment. In our study, the beetles were paired first to ensure female insemination and partner recognition before the start of experimentation 1, but they cannot breed in the absence of breeding resources. After insemination, they were introduced to a breeding carcass. In our study, they were kept as pairs, while these pairs were categorized into two groups according to their size (large- and small-pairs groups). In burying beetles, there is no direct evidence for pair-bonded relationships, but individuals are more likely to copulate with similarly sized mates and to become breeding partners. Also, this could create a relatively stable dyadic interaction between large and small beetles in communal groups, although each individual is able to mate with other individuals (e.g. male sneakers).

Page 8 top paragraph: There was a breeding period that differed from the brooding period and the beetles bred with different partners that were not their brooding partners? You need to clarify this part a bit. Also, you need to justify why this was done.

We thank very much for the comment and have now clarified this in our revised manuscript (Line 207-220, also see the response to the reviewer 1):

“In burying beetles, adult body size affected an individual's competitive ability, while this was closely linked with carcass monopolization and reproductive success [65,74,75]. To create the communally breeding treatment, one large pair (dominants) and one small pair (subordinates) of beetles (n = 19) were selected, and we ensured that the two pairs of beetles were unrelated and differed approximately 10 % in body size, because the likelihood of stable dominance hierarchy establishment in carcass use was enhanced when size difference between opponents was larger [64,72]. Other studies of ours indicated that the double-pair breeding treatment could create a communal breeding event with a size-dependent dominance hierarchy in carcass monopolization [76], which was accordance with previous studies about communal breeding in burying beetles

[63,65,77]. For these beetles, an individual's body size has also been found to influence its parental investment towards current brood [78]. Thus, we randomly selected one pair of beetles from small-pair ($n = 10$) or large-pair ($n = 10$) groups to create the non-communal breeding treatment, which could control the effect of body size on individual behaviour (i.e. parental investment) [78,79]. Just prior to creation of communal and non-communal breeding, each individual was weighed (accuracy: 0.0001 g)."

Page 8 line 221: For how long did you make observations? How many days or weeks?

Thank you very much for the comment. We made observations during the entire reproductive period, from the onset of experimentation until the dispersal of larvae from the carcass. The observational period is flexible (around 10-14 days), which is depending on parental investment time and the offspring developmental period. We have now added the mean and variation of observational period in our revised manuscript (Line XXX).

Page 8 lines 228-235: Did you measure offspring mortality? If you did not, why not?

Thank you very much for the comment. In this study, we did not measure offspring mortality when they emerged as new adults. This is because we measure the immediate benefits of communal breeding in terms of reproductive outcomes, and offspring number and mean size would be best measures for assessing reproductive outcome for parents, as well as offspring fitness. We have now clarified this in our revised manuscript (Line 258-264):
“As measures for offspring development and reproductive performance, we used larvae-dispersing time (the time from the start of the experiment until the onset of larvae dispersing), brood size (number of larvae) and average larval weight (total weight of larvae/brood size) at the larval dispersal (i.e. larvae dispersed from the carcass). Also, we recorded weight change during breeding ($[(final\ weight - initial\ weight)/ initial\ weight]$ and survival at the end of the experiment of adult individuals, as parameters for parental effort during the entire breeding period and reproductive costs for individuals, respectively [49,57].”

Page 9, Exp. 2: What was the time delay between experiments? What was the beetle husbandry between experiments? In the results, you mention they all pretty much lost weight. Did you try to increase their weight between the experiments?

Thank you very much for the comment, The time delay between experiments was five days, and all beetles were kept individually and were not fed. Then, all beetles were used again for subsequent experimentation. In the study, we expect that beetles that bred previously in communal or non-communal groups may gain or lose weight after breeding. To investigate whether previous weight change influenced an individual's parenting behaviour and performance in the subsequent period, we did not feed beetles in the short time between the experiments. We have now clarified this in our revised manuscript (Line 264-268):
“After the first reproductive round, surviving beetles were kept individually for five days in rearing boxes (10×5 cm and 8.5 cm high) and not fed by any food (i.e. mealworms) for subsequent experimentation. We did not feed beetles during this period to avoid the potential effect of food consumption and weight change on an individual's behaviour in the subsequent period.”

In the results: Be consistent with reporting the statistical results versus just listing the table where the results can be found.

We thank very much for the comment and have now revised this listing in our revised manuscript. In our revised manuscript, we reported the statistical results in the main context that was consistent with the listing results in the table. For example, we found that large pairs spent more time on the carcass compared with small individuals, and we reported statistical results (mean \pm SE of large vs. small individuals: 58.96 ± 2.95 vs. $7.22 \pm$

1.33%; Table 1), which was consistent with the listing results in the table ($\chi^2 = 315.65$, $p < 0.001$ shown in Table 1) (Line 351-355):

"In the communal groups, we found that the larger pairs of individuals became dominant and significantly monopolized the access of carcass, whereas the smaller individuals were subordinates that spent less time on the carcass compared with larger individuals during the entire breeding period (mean \pm SE of large vs. small individuals: 58.96 ± 2.95 vs. 7.22 ± 1.33 ; $\chi^2 = 315.65$, $p < 0.001$; Table 1)."

Page 15 line 440: Be consistent with citations. Need to change this citation to numbers.

Thank you very much. We have now corrected these citations in our revised manuscript.

On the graphs: Use a darker grey for the non-communal lines and info. When I printed out the sheets, the lines did not show up. They were also very faint on the electronic version.

Thank you very much for the helpful suggestion. We have now changed the color of the graphs in our revised manuscript.

Figure S3: Add brood size, body weight, and ave. larval weight as titles to the 3 graphs.

We thank very much for the comment and have now added these titles to the Figure S3 in our revised manuscript.

===PREPARING YOUR MANUSCRIPT===

- one version identifying all the changes that have been made (for instance, in coloured highlight, in bold text, or tracked changes);
- a 'clean' version of the new manuscript that incorporates the changes made, but does not highlight them. This version will be used for typesetting if your manuscript is accepted.

If you have been asked to revise the written English in your submission as a condition of

publication, you must do so, and you are expected to provide evidence that you have received language editing support. The journal would prefer that you use a professional language editing service and provide a certificate of editing, but a signed letter from a colleague who is a native speaker of English is acceptable. Note the journal has arranged a number of discounts for authors using professional language editing services (<https://royalsociety.org/journals/authors/benefits/language-editing/>).

===PREPARING YOUR REVISION IN SCHOLARONE===

-- Ensure that your data access statement meets the requirements

at <https://royalsociety.org/journals/authors/author-guidelines/#data>. You should ensure that you cite the dataset in your reference list. If you have deposited data etc in the Dryad

repository, please include both the 'For publication' link and 'For review' link at this stage.

- If you are requesting an article processing charge waiver, you must select the relevant waiver option (if requesting a discretionary waiver, the form should have been uploaded at Step 3 'File upload' above).
- If you have uploaded ESM files, please ensure you follow the guidance at <https://royalsociety.org/journals/authors/author-guidelines/#supplementary-material> to include a suitable title and informative caption. An example of appropriate titling and captioning may be found at https://figshare.com/articles/Table_S2_from_Is_there_a_trade-off_between_peak_performance_and_performance_breadth_across_temperatures_for_aerobic_scope_in_teleost_fishes_/3843624.

Appendix C

December 20, 2021

Dear Editors, Dr. Polly Campbell and Dr. Kevin Padian,

We would like to thank you and the reviewer for the constructive reviews of our manuscript entitled "*Sex-specific influence of communal breeding experience on parental performance and fitness in a burying beetle*", and for the opportunity to resubmit a revised version. We have carefully read all comments, replied to each of them below and revised our manuscript accordingly. We very much hope that our amendments, in the light of your and the reviewer's comments, have improved the manuscript. We sincerely hope that you will find the current version acceptable for publication in *Royal Society Open Science*. Thank you very much for reconsidering our manuscript.

Yours sincerely,
Long Ma, on behalf of all authors

Dear Dr Ma

The Editors assigned to your paper RSOS-211179.R1 "Sex-specific influence of communal breeding experience on parenting performance and fitness in a burying beetle" have now received comments from reviewers and would like you to revise the paper in accordance with the reviewer comments and any comments from the Editors. Please note this decision does not guarantee eventual acceptance.

Please submit your revised manuscript and required files (see below) no later than 21 days from today's (ie 22-Nov-2021) date. Note: the ScholarOne system will 'lock' if submission of the revision is attempted 21 or more days after the deadline. If you do not think you will be able to meet this deadline please contact the editorial office immediately.

on behalf of Dr Polly Campbell (Associate Editor) and Kevin Padian (Subject Editor)
openscience@royalsociety.org

Associate Editor Comments to Author (Dr Polly Campbell):

Associate Editor: 1

Comments to the Author:

The authors have done a great deal to improve the clarity of this manuscript and their scholarly approach in responses to reviewer critiques is very much appreciated. However, there are still many places where sentence structure and/or the flow of ideas between sentences is very hard to follow. The reviewer provides detailed examples of these issues and suggestions on how to fix them, and I provide additional non-overlapping comments below. Could the authors please check that they addressed Reviewer 1's line-by-line comments from the first submission? If not, please address those along with the current comments. The authors should also consider having a fluent English speaker check the manuscript for clarity before resubmission.

Thank you very much for the positive and constructive comments. We have addressed all comments by Associate Editor and Reviewer 1 in our revised manuscript. We have carefully checked the English language of the manuscript for clarity. The entire manuscript has been carefully read and checked by Dr. Sacha Engelhard, who is a native English speaker.

The reviewer's remaining concerns about the statistical analyses and interpretation of results are also important; please give these careful consideration.

Thank you very much for the comment, and we have addressed these issues in our revised manuscript below.

Specific comments

Fitness: there are still multiple references to measuring fitness in the Introduction (L150-175). Referring to short-term fitness implications (L156) doesn't address the issue that individual reproductive output couldn't be measured in the communal breeding treatment, it just makes things a bit fuzzy. The measures of "parental performance" are indices of parental investment. This is fine but these measures are not a proxy for short-term fitness. Please go through the manuscript using the find function in MS Word and for every use of "fitness" in reference to the experiments consider whether another more specific and accurate term could substitute.

Thank you very much for the comment. We agree with you that our measures of 'parental performance' are indices of parental investment, and not a proxy for short-term fitness. We revised the terms of 'fitness' more specific throughout the manuscript. For example, in this study, we changed the short-term fitness implications of communal breeding as the short-term effects of communal breeding

on parental care and reproduction. We have now revised all terms of 'fitness' more specific throughout the revised manuscript (e.g. Line 154-160):

"Second, we examine the short-term fitness implications of communal breeding and its carry-over effects on parental care and reproduction. To investigate the short-term fitness implications of communal breeding on parental care and reproduction, we measured the parental care of each individual (i.e. time spent providing care on the carcass and weight change) and of groups (i.e. burial degree of carcass and the total amount of parental investment by groups), as well as the reproductive success of groups (i.e. brood size and larval weight) in communal and non-communal breeding."

L221-227 I appreciate that this sentence was added in response to a comment but it is confusingly written and is out of place in Methods. Is the point to provide confirmation that double-pair treatments induce communal breeding or to provide evidence that small individuals' reproduction suffers in communal breeding? Either way, the numbers in parentheses are not necessary and the authors might consider integrating the result into the Discussion.

Thank you very much for the comment. Yes, we provided the confirmation that double-pair treatments normally induce communal breeding, and we explained that small individuals reproduce in such groups as well (9 out of 10 small females produced, 5 out of 10 small males produced). We have now rephrased this sentence and deleted some statistics in parentheses in our revised manuscript (Line 220-224):

"In another study, we tested the parentage of offspring produced in double-pair breeding groups using 5 microsatellite loci [80], and found that small pairs of individuals produced a relatively small proportion of offspring in a shared brood (9 out of 10 small females produced, 5 out of 10 small males produced; $N_{brood} = 10$, $N_{offspring} = 204$) [76], indicating that communal breeding events (i.e. one shared brood) were indeed induced in such double-pair treatment."

L233-236 "Each check lasted for 30 seconds so as to make sure that beetles were providing care or not when they were present or absent on the carcass, which could also avoid the counts of any accidental events, for example, beetles were wandering on the carcass without providing care." This sentence is very hard to follow. Please revise.

Thank you for the comment, and we have rephrased this sentence in our revised manuscript (Line 231-237):

"For each check, beetles were observed for 30 seconds per group continuously to ascertain whether or not they were providing parental care on the carcass. Such continuous observations could exclude any cases where beetles were present on the carcass for other reasons (e.g. wandering on the carcass) than to provide parental care [70]. We recorded parental care behaviour when an individual provided indirect parental care (i.e. carcass guarding and maintenance on the surface of the carcass), and/or provided direct parental care (i.e. larvae provisioning inside the carcass) [63,81,82]."

L407-414 This sentence is very hard to follow and seems to contradict the previous sentence (404-407).

Thank you very much for the comment. We have rephrased this sentence in our revised manuscript (Lines 400-411). The previous sentence (Line 400-403) mentions the effect of the partner's communal breeding experience on the focal individuals'

parenting care time, regardless of the focal individual's breeding experience, whereas the current sentence (Line 404-411) highlights the combined effect of previous breeding experience of focal individuals and that of their partners.

The revised sentence is now (Line 400-411):

“Regardless of their own previous breeding experience, female and male individuals spent significantly more time on the carcass when their partners originated from communal groups than from non-communal groups (mean \pm SE of females' partner: $77.19 \pm 4.15\%$, $n = 13$ vs. $54.75 \pm 8.74\%$, $n = 14$; of males' partner: $46.99 \pm 6.44\%$, $n = 21$ vs. $29.12 \pm 7.10\%$, $n = 20$; figure 4c, d; table 1B and S2). In addition, when their partners originated from non-communal groups, both female and male individuals originating from communal groups spent significantly less time on the carcass than those from non-communal groups (mean \pm SE of females: $44.88 \pm 9.45\%$, $n = 9$ vs. $72.50 \pm 15.85\%$, $n = 5$; of males: $20.41 \pm 8.64\%$, $n = 8$ vs. $43.06 \pm 10.24\%$, $n = 5$; $z = -3.01$, $p = 0.005$; table 1B and S2). However, when their partners originated from communal groups, individuals originating from communal or non-communal groups spent similar times providing care on the carcass (mean \pm SE of females: $73.36 \pm 5.62\%$, $n = 12$ vs. $82.94 \pm 5.90\%$, $n = 8$; of males: $50.65 \pm 11.38\%$, $n = 12$ vs. $42.11 \pm 7.68\%$, $n = 9$; $z = -0.43$, $p = 0.89$; table 1B and S2).”

L513 not clear what “increase individual conditions” means

Thank you for the comment. We meant that the physical condition of beetles may improve. We have clarified this in our revised manuscript (Line 511-516):

“Burying beetles could gain fitness benefits from access to carcasses during breeding, because access to resources (e.g. by spending more time on the carcass, and by consuming more of the carcass) while breeding might alleviate the energetic costs of reproduction. As such, individuals with access to additional resources may improve their physical conditions (e.g. body mass and immunity) and sexual attractiveness, which may associate positively with enhanced reproductive success [57,59,63,90,99].”

L516-519 not clear what the last clause of this sentence means (“which may be due to reproductive restraints [constraints?] in females.”) and how it relates to the first part.

Thank you very much for the comment. In the last clause of this sentence, we provide an explanation for the decreased investment in the current reproduction by *N. orbicollis* females that breed on low-quality carcasses compared to females breeding on high-quality carcasses. A change in carcass quality may reflect reproductive restraints (i.e. females self-restrain and decrease the amount of resources they allocate to current reproduction in response to deteriorating physical condition to increase the probability of realizing additional reproductive opportunities), rather than reproductive constraints. We have rephrased this sentence in our revised manuscript (Line 524-528):

“For example, in *N. orbicollis*, females reproducing on low-quality carcasses invest less in current reproduction and allocate more to future reproduction compared with females given high-quality carcasses. As such a change in carcass quality may reflect reproductive restraints by females decreasing the amount of resources in the current reproduction and increasing the probability of future reproductive opportunities [57,90].”

L553-554 Not clear what “the presence of males does not seem to buffer this negative effect...” refers to. Males are present in all treatments...

Thank you very much for the comment. We suggest that the parental effort by males (i.e. the presence of males within pairs) does not seem to buffer this negative effect of communal breeding on female fitness, because males may not provide care and do not share parental duties even when they are present. We have now rephrased this sentence in our revised manuscript (Line 561-566):

“For breeding pairs, the parental investment by males does not seem to buffer this negative effect of communal breeding experience on female fitness, as well as on the joint parental investment of pairs. This may be because males may have partial or no compensation for a reduction in parental investment by females depending on their own and their partners’ history of previous breeding, which is driven by sexual conflict over parental investment between parents [57,81,105,106].”

Reviewer comments to Author:

Reviewer: 1

Comments to the Author(s)

The authors have completed a very thorough revision of the manuscript based on reviewer feedback, and many items are much clearer.

Thank you very much for the positive comment and your previous feedback.

My main comment on the manuscript at this stage is that extensive editing for sentence clarity, manuscript organization, grammar, and other general issues still needs to be done. Please see the line-by-line comments for examples – this is not exhaustive; I am mainly indicating a number of specific locations that need work or are confusing for the reader.

Thank you very much for the constructive comment, and we have carefully revised this manuscript according to your line-by-line comments. The revised manuscript has been read by a native English speaker, Dr. Sacha Engelhard. We believe that the revised manuscript is clearer.

Terminology is inconsistent and confusing throughout the paper. I strongly encourage the authors to pick a specific term and to stick with it. E.g., “reproductive performance”, “reproductive outcome”, “parental performance”, “parental care”, “parental investment” – these all invoke slightly different messages and I can’t quite figure out where they do and do not cross over (sometimes used as synonyms, sometimes as different measures, etc.). For example, “higher reproductive outcome” is used, but technically a reproductive outcome could be complete failure, so unclear how that could be higher. Also, some locations where word choice should be updated to the common term; e.g., “reproductive restraint” is more commonly described as “reproductive constraint”, etc.

Thank you very much for the comment. We agree with you that the terminology is not consistent and makes confusing throughout the paper. We have chosen ‘parental care/investment’, ‘reproductive success’ and ‘breeding events’ as consistent terms in our revised manuscript. For example, we changed ‘parental performance and parental effort’ as ‘parental care/investment’ and ‘reproductive performance and outcome’ as ‘reproductive success’. We have revised the terminology to be consistent throughout the revised manuscript (e.g. Line 33-38 and Line 152-160):

“We experimentally tested the effects of communal breeding on parental care and reproduction in burying beetles (*Nicrophorus vespilloides*), which utilize carcasses as

breeding resources and provide extended care to offspring on buried carcasses. We subjected individuals to communal or non-communal breeding (i.e. pair-breeding) treatments during their first breeding event, and to non-communal breeding during their second breeding event. We measured the parental care of individuals and of groups and the reproductive success of groups during both breeding events.”

“The aims of our study are twofold. First, we investigate whether individuals adjust their parental care behaviour and reduce parental investment in communal groups compared to non-communal groups (pair breeding). Second, we examine the short-term fitness implications of communal breeding and its carry-over effects on parental care and reproduction. To investigate the short-term fitness implications of communal breeding on parental care and reproduction, we measured the parental care of each individual (i.e. time spent providing care on the carcass and weight change) and of groups (i.e. burial degree of carcass and the total amount of parental investment by groups), as well as the reproductive success of groups (i.e. brood size and larval weight) in communal and non-communal breeding.”

I am still concerned about the sample sizes and the number of fixed effects and interactions used in the statistical analyses. For carry-over effects in particular – the sample sizes are quite small and the models have a lot of variables in there, I think the risk of overparameterizing the model is quite high – small vs large, communal versus non-communal, males versus females etc., etc. Along the same lines, on Figure 5a., why does the sample size of $n=3$ not have an error bar? And why does it look like that time = 0? Same question for 5b. These look more like artifacts of small sample size rather than biologically-relevant outcomes for communal breeding.

Thank you very much for the comment. Yes, we agree with your concerns. In particular, the sample sizes of the carry-over effects (i.e. experiment 2) are small, and the models have a lot of variables included. However, we select best models carefully to analyse and show the results involved. In figure 5a and 5b, there are no error bars when the sample sizes are small, because all values are same and the error bar is equal 0. In this study, we checked the beetles and larvae two times per day during the entire breeding period, while this large interval between observational checks results in that we did not collect large sample of data about developing timing from all breeding trials. In Figure 5a,b, larval-hatching time appeared to equal zero, because we had incorrectly set the y-axis scale. We have re-organized these two figures and correct the y-axis scale in our revised manuscript (please see Figures 5a,b). We have added the sample sizes in the result sections (e.g. Line 423-442), figures and supplementary materials of our revised manuscript:

“Females originating from communal groups produced larvae that hatched and dispersed later from the carcass compared to females originating from non-communal groups (mean \pm SE of larvae-hatching time: $88.89 \pm 2.23\text{h}$, $n = 18$ vs. $76.80 \pm 2.44\text{h}$, $n = 10$; of larvae-dispersing time: $172.00 \pm 4.00\text{h}$, $n = 18$ vs. $151.40 \pm 3.20\text{h}$, $n = 13$; figure 5a,b; table 2 and S3), whereas no difference was observed for males originating from communal and non-communal groups in the timing of offspring development (table 2 and S3). Regardless of male’s previous breeding experience, females originating from communal and non-communal groups had similar reproductive success (mean \pm SE of brood size: 22.61 ± 2.15 , $n = 18$ vs. 25.08 ± 1.28 , $n = 13$; of average larval weight: $0.1389 \pm 0.0067\text{g}$, $n = 18$ vs. $0.1333 \pm 0.0053\text{g}$, $n = 13$; table 2 and S3). However, females originating from communal groups produced smaller brood size than females originating from non-communal groups when males originated from non-communal groups (mean \pm SE: 17.57 ± 3.93 , $n = 7$ vs.

25.40 ± 2.56, n = 5; z = -2.36, p = 0.04; table S3). Males originating from communal groups produced larger brood size than males originating from non-communal groups (mean ± SE: 25.42 ± 1.33, n = 19 vs. 20.83 ± 2.70, n = 12), whereas the average larval weight was similar for males originating from communal and non-communal groups (mean ± SE: 0.1389 ± 0.0006g, n = 19 vs. 0.1328 ± 0.0056g, n = 12; table S2). Additionally, the size of males originating from communal and non-communal groups significantly influenced the reproductive success (table S3). In particular, large males originating from communal groups produced heavier larvae than large males from non-communal groups (figure 5d; table 2 and S3), whereas the reproductive success was similar for small males originating from communal and non-communal groups (figure 5c, d and S3; table 2 and S3)."

We discuss the potential effects of small sample size in the dicussion and suggest for future studies larger sample sizes would be better (Line 629-630):

"Future studies with larger sample sizes should investigate the effect of communal breeding on the allocation of parental investment and fitness between sexes."

The authors need to tone down their interpretations of the results. Two main examples. First, individual reproductive success in communal set-ups was not measured – so comparing reproductive measures for individuals that were in a communal vs non-communal setting does not give us insight into the short-term or carry-over effects for individuals (line 448-452). For this reason, need to be careful discussing results.

We thank you for the comment, and we agree with you that individual reproductive success in communal set-up was not measured in this manuscript. We discussed the reproductive success of groups between communal and non-communal breeding at the group level, and individual parental care and parental investment in such groups. Thus, we toned down our conclusions and carefully discussed our findings. For example, we carefully discussed the lack of a biologically significant reproductive benefits at the group level for communal breeding, which is also consistent with previous studies (Eggert and Muller, 1992; Muller and Eggert, 1990; Trumbo, 1992). According to your comment, we have rephrased this point in the discussion of our revised manuscript (e.g. Line 447-455):

"We found that, at the group level, communal breeding appeared to have no short-term effects on parental investment and reproductive success, however we found carry-over effects on parental investment and reproduction. Our results indicated that breeding in communal groups showed enhanced fitness benefits at the group level by improving carcass burial performance and total parental investment time, whereas individual parental investment time was similar in communal and non-communal groups. Our findings on the short-term effects of communal breeding are consistent with previous studies on burying beetles, which indicates that individuals reproducing in communal groups do not have a higher reproductive success than individuals breeding in pairs [59,64,65,67,91,92]."

Eggert A.K., Müller J.K. (1992) Joint breeding in female burying beetles. *Behavioral Ecology and Sociobiology*, 31: 237-242.

Müller J.K., Eggert A.K. (1990) Time-dependent shifts between infanticidal and parental behavior in female burying beetles a mechanism of indirect mother-offspring recognition. *Behavioral Ecology and Sociobiology*, 27: 11-16.

Trumbo S.T. (1992) Monogamy to communal breeding: exploitation of a broad resource base by burying beetles *Nicrophorus*. *Ecological Entomology*, 17: 289-298.

Second, you are conducting this experiment in the most benign conditions possible, and yet there is incredibly high mortality in your beetles (over 50%; based on the numbers provided on line 423). Beetles have high mortality by second breeding attempt in benign conditions, carcasses are rare to some extent (e.g., line 472), so seems unlikely that that many beetles actually end up breeding more than once. For this reason, it seems unlikely that selection has, for example, honed specific pheromone cocktail that indicates that a partner bred communally in a previous attempt (line 566). So many factors changed between the first and second attempt (size, age, breeding experience, feeding prior to experimentation, paternity assurance, etc) that, when paired with the low sample size, indicates that there are likely multiple explanations for the patterns the authors observed, and not all of them are based on adaptation. This can be fixed by more cautious wording.

We thank you for the comment. We agree with you that mortality is high for beetles (over 50%) during the second breeding attempt, and we suggest that a previous experience in the first breeding event incurs significant individual survival costs. In this study, beetles had a high mortality because they breed on large carcasses or under a high level of intraspecific competition. Although breeding resources are ephemeral and limited, it is likely that many beetles could breed more than once, especially when breeding on small or medium-sized carcasses (Ward et al., 2009). We agree with you that such a previous breeding experience has no selection on specific pheromone cocktail. However, it is likely that differences in previous breeding experience (communal vs non-communal breeding, dominance status) may have a subsequent effect on physical and physiological conditions, such as immunity and pheromone profiles. These changes may be associated with subsequent individual parental behaviour. Changes associated with previous breeding experience are discussed, which could help understand the underlying mechanism of previous breeding experience on individual behaviour. To improve our discussion, we have use more cautious wording according to your comments in our revised manuscript (Line 572-577):

“It is well known that hormonal levels (e.g. juvenile hormones) rapidly change during breeding in burying beetles, and these changes may differ with the social environments (e.g. dominance status) [52,110]. Moreover, these physiological changes may intrinsically mediate the emission of pheromones involved and other physiological conditions, which may have influences on individual reproductive states and parental behaviour in the subsequent period.”

References:

Ward R.J.S., Cotter S.C., Kilner R.M. (2009) Current brood size and residual reproductive value predict offspring desertion in the burying beetle *Nicrophorus vespilloides*. Behavioral Ecology, 20(6): 1274-1281.

This is a paper about the impacts of communal behavior and yet no data about the communal behavior are presented. How often were more than two beetles on the carcass? What proportion of the time? What kind of things were they seen doing – i.e., how often were they fighting? Working together? This would be very useful information to inform your discussion of sources of mortality in particular. The mechanism of mortality is still presented as a hypothesis (line 486), where you are stating what it might be due to and citing references. What did you observe? Were they ripped apart? Did they have injuries? Were they whole but just dead?

Thank you very much for the comment. When one carcass is found by more than two burying beetles, they may have several rounds of fights over the carcass. Larger individuals are more likely to become the winner and have preferential access to a carcass due to the establishment of a dominance hierarchy (Eggert et al., 2008; Müller et al., 1990). However, sustained fights between individuals are rare, especially on large-sized carcasses. Thus, we did not measure how often beetles fight in this study. However, we record individual adult survival at the end of the experiment 1 and 2 (see Line 263-266 and Line 301-302). Communal behaviour in communal breeding of burying beetles does not mean that more than two beetles should be present simultaneously on the carcass. Instead, beetles always bury the carcass and provide care to offspring side-by-side or alternatively (Trumbo, 1992; Trumbo and Wilson, 1993). Therefore, in our study (i.e. in closed breeding systems where losing individuals are unable to disperse from the carcass for future reproduction), we measured how often each individual spend providing care on the carcass (Line 235-240):

“We recorded parental care behaviour when an individual provided indirect parental care (i.e. carcass guarding and maintenance on the surface of the carcass), and/or provided direct parental care (i.e. larvae provisioning inside the carcass) [63,81,82]. For each individual, we defined individual parental investment time by calculating the proportion of times that each individual was observed providing parental care on the carcass (i.e. carcass preparation and offspring provisioning) during the entire observation period [49,83,84].”

According to our study, we found that beetles suffered a higher mortality in communal groups than in non-communal groups, which suggests that there is a higher level of conflict when beetles breed with conspecifics (Figure 1S). We also found injuries on beetles in communal groups, but we also found that some beetles died without having significant injuries. Thus, we suggest that individual mortality is due to the fights and the costs of reproduction. We have discussed this in our revised manuscript (e.g. Line 488-490):

“Both males and females originating from communal breeding experienced higher mortality compared with individuals originating from non-communal breeding. This may be due to that individuals originating from communal breeding incur high reproduction and survival costs [57,62,90].”

References:

- Eggert A.K., Otte T., Müller J.K. (2008) Starving the competition: A proximate cause of reproductive skew in burying beetles (*Nicrophorus vespilloides*). *Proc. R. Soc. B.* 275: 2521-2528.
- Müller J.K., Eggert A.K., Dressel J. (1990) Intraspecific brood parasitism in the burying beetle, *Nicrophorus vespilloides* (Coleoptera: Silphidae). *Animal Behaviour*, 40(3): 491-499.
- Trumbo S.T. (1992) Monogamy to communal breeding: exploitation of a broad resource base by burying beetles *Nicrophorus*. *Ecological Entomology*, 17: 289-298.
- Trumbo S.T., Wilson D.S. (1993) Brood discrimination, nest mate discrimination, and determinants of social behavior in facultatively quasisocial beetles (*Nicrophorus* spp.). *Behavioral Ecology*, 4(4): 332-339.

It would be helpful for the reader to have the samples sizes included in the results section, with their respective statistics (e.g., after means and standard errors).

Thank you very much for the comment and we have now added the sample sizes in the results section of our revised manuscript (Line 425-444, also see our above responses to reviewer 1).

Line-specific comments:

Line 99-101: Edit sentence for grammar.

Thank you very much for the comment. We have now revised this sentence in our revised manuscript (Line 98-101):

“Moreover, previous breeding experience and changes in physiological states may have carry-over effects on fitness and reproductive performance for individuals [45,46]. For example, experienced individuals with previous breeding provide more parental care towards the current broods compared to inexperienced individuals.”

Line 104-106: “Special” carry-over effect- if carry-over effects are just the effect of previous experience, then why specify a “special” carry-over effect?

Thank you very much for the comment, and we have now rephrased this sentence in our revised manuscript (Line 104-106):

“The impact of allocation of parental effort to the current reproduction on subsequent reproductive performance and survival can also be viewed as a type of carry-over effect [32,44,48].”

Line 108-109: talking from viewpoint of dominant? Revise: “allocation of resources by breeding individuals can be adjusted by their relative position in a dominance hierarchy” or some such.

Thank you very much for the comment, and we have now revised this (Line 106-109):

“Compared to non-group breeding species, group breeding species are particularly interesting to study trade-offs between current and future reproduction because parenting behaviour and allocation of resources by breeding individuals can be adjusted by their relative position in a dominance hierarchy [19,24,47,49].”

Line 113: This sentence needs to be edited. Says that “impacts . . . reproductive success . . . through difference in . . . reproduction”. Suggestion: “as such, breeding in groups may impact reproductive success not only through short-term effects, but also through carry-over effects on future reproductive allocation.”

Thank you very much for the comment. We have now rephrased this sentence in our revised manuscript (Line 111-113):

“As such, breeding in groups may impact reproductive success not only through short-term effects, but also through carry-over effects on future reproductive allocation [31,48,52,55].”

Line 131: “large individuals become dominants can” – edit

Thank you very much for the comment, and we have now corrected this in our revised manuscript (Line 129-131):

“Specifically, larger individuals become dominants that can largely monopolize the carcass, while subordinates are smaller individuals and have limited access to the carcass [35,65,66].”

Line 139: “individual’s breeding opportunity” – carcass is plural

Thank you for the comment, and we have now corrected this in our revised manuscript (Line 137-139):

“For burying beetles, breeding resources (i.e. carcasses) are ephemeral and limited, which largely determine an individual’s breeding opportunity during a given time frame, as well as its reproductive strategy [66,70].”

Line 141: remove comma after “reproduction”

Corrected.

Line 140-144: shorten sentence. E.g. “While breeding communally may decrease reproductive

Thank you for the comments, and we have now shorten this sentence in our revised manuscript (Line 139-140):

“While breeding communally may decrease fitness benefits in reproduction compared to breeding alone or in pairs, ...”

Line 146-149: edit sentence “breeding communally among individuals”. Also – this is hypothesis, state as the hypothesis.

Thank you very much for the comment, and we have now edited this sentence and have stated this as a hypothesis in our revised manuscript (Line 144-147):

“It can be hypothesized that breeding communally among individuals evolves as an adaptive strategy, which may largely reduce the reproductive costs in adverse environmental conditions and potentially improve the overall fitness benefits across breeding events and over an individual’s lifetime.”

Line 149-153: how do short-term benefits on fitness act on selection? Fitness is a measure of selection; it does not “act on” selection. Re-word.

Thank you very much for the comment. We have now rephrased this sentence in our revised manuscript (Line 147-151):

“Thus, it is worth studying the carry-over effects of communal breeding on parental behaviour and fitness across breeding events in burying beetles. It is also important to examine whether and how carry-over effects, concurrent with short-term effects, act on the selection for group breeding and its adaptation to different environments.”

Line 161: use consistent terms throughout – “reproductive outcome”? Reproductive success is a common term for this.

Thank you very much for the comment, and we have now edited this term in our revised manuscript (Line 160-161). We have also corrected this throughout our revised manuscript:

“We predict that communal groups achieve higher reproductive success compared to non-communal groups.”

Line 165: “each pairs” edit “each pair”. But didn’t you switch individuals between pairs? So the pair was not a consistent unit – this is not quite accurate of a description here then.

Thank you very much for the comment, and we agree with you. Yes, we did switch individuals between pairs to create newly-formed pairs. And we have now clarified this in methodology (Line 281-286):

“For this experiment, beetles that had not formed pairs with each other in the previous experiment, and originated from communal (i.e. double-pair treatment) or non-communal groups (i.e. single-pair treatment), were paired randomly; e.g. large females from communal groups were paired with either large or small males from communal or non-communal groups, and small females from communal groups were paired with either large or small males from communal or non-communal groups (figure 1).”

We have now deleted this in our revised manuscript (Line 161-165):

“To examine the carry-over effects of communal breeding, we experimentally investigated the impact of previous breeding experience, i.e. communal versus non-communal breeding, on future parental care (e.g. burial degree of carcass for each pair, an individual’s time spent providing care and weight change) and reproductive success for each pair in a subsequent pair-breeding event.”

Line 173: “parental efforts” - “parental effort”

Corrected.

Line 166-175: You state here that “individuals have been shown to reduce their efforts [in a communal setting]” but then on line 154-155 you are saying one of the goals of your project is to see whether they adjust parental care behavior in a communal setting. This is contradictory – edit accordingly.

Thank you very much for the comment, and we have now edited this in our revised manuscript (Line 152-154):

“First, we investigate whether individuals adjust their parental care behaviour and reduce parental investment in communal groups compared to non-communal groups (pair breeding).”

Line 174: what are “parental resources”? Use consistent terminology.

Thank you very much for the comment, and we have now rephrased this by using consistent terminology in our revised manuscript (Line 172-174):

“If sex-specific differences in individual resource allocation during communal breeding are more pronounced than during pair breeding, we also expect sex differences in carry-over effects of communal breeding history on future fitness.”

Line 187 – 190: need to edit. “All beetles were reared in small groups, and thus the social environment in early life was similar for all beetles” or something like that

Thank you very much for the comment, and we have now edited this sentence in our revised manuscript (Line 185-187):

“All beetles were reared in small and similarly-sized groups in the absence of a breeding resource (e.g. mouse carcass), and thus the social environment in early life was similar for all beetles.”

Line 195: also throughout – is “parental performance” the same as “parental care”? I’d say that parental care is part of reproduction as well. Solution - could say parental care and reproductive success and it would be immediately obvious to the reader.

Thank you very much for the comment, and we fully agree with you. We have now changed ‘parental performance’ and ‘reproductive outcome’ as ‘parental care’ and ‘reproductive success’, respectively throughout our revised manuscript (e.g. Line 155-156):

“To investigate the short-term fitness implications of communal breeding on parental care and reproduction, ...”

Line 196-198: why two names for the treatments? Just call “communal” (define as two males and two females) and “non-communal” (one male, one female).

Thank you for the comment. In our study, we performed double-pair (i.e. one large pair and one small pair) and single-pair (one female and one male) treatments to create communal and non-communal breeding events, respectively. We used beetle pairs that differed in body size. Thus, it is necessary to use two names and let the methodology more clear. However, in results and other sections, we just stick with one name (communal versus non-communal groups) for the entire manuscript (e.g. Line 194-197).

“To investigate the immediate implications of communal breeding on parental care and reproductive success, we set-up double-pair (consisting of two pairs, i.e., one large pair and one small pair) and single-pair (one male and one female) treatments to create communal and non-communal breeding (i.e. pair breeding) events, respectively [64,72] (figure 1).”

Line 206-207: Need to edit this sentence, also provide directionality: e.g., “In burying beetles, adult body size determines competitive ability, such that larger individuals are more likely to monopolize a carcass and have higher reproductive success.”

Thank you very much for the comment, and we have now edited this sentence in our revised manuscript (Line 212-214):

“For these beetles, an individual’s body size has also been found to influence its parental investment towards the current brood, i.e. larger individuals likely invest more in the current brood than smaller individuals [78].”

Line 210: “is” instead of “was” – or rephrase

Thank you very much for the comment, and we have now rephrased this sentence in our revised manuscript (Line 210-211):

“because a stable dominance hierarchy in carcass use is more likely when the size difference between opponents is larger [64,72].”

Line 211-214: How does this sentence differ from the previous – i.e., small pair + large pair = stable dominance hierarchy. Remove one. Also “was accordance”

Thank you very much for the comment. First, we explained why we used two pairs of individuals that differed in body size - because a stable dominance hierarchy in carcass use is more likely when the size difference between opponents is larger (Line 210-211). Second, we explained that such a double-pair breeding treatment could create a communal breeding event in according with previous studies. We have now rephrased this in our revised manuscript (Line 211-212):

“Previous studies indicated that the double-pair breeding treatment could create a communal breeding event [76,77].”

Line 214-216: Provide directionality. I.e., Larger individuals invest more in brood? Less?

Thank you very much for the comment, and we have now provided more details about the effect of body size on parental investment in our revised manuscript (Line 212-214):

“For these beetles, an individual’s body size has also been found to influence its parental investment towards the current brood, i.e. larger individuals likely invest more in the current brood than smaller individuals [78].”

Line 216-218: Selected one large pair and one small pair for non-communal? Makes it sound like you only have two replicates of non-communal.

Thank you very much for the comment, and we agree with your comment. We have now rephrased this in our revised manuscript (Line 214-216):

“Thus, we randomly selected some pairs of beetles from small-pair (n=10) or large-pair (n=10) groups to create the non-communal breeding treatment, ...”

Line 218-219: Sentence awkward, make it into active voice. E.g., Individuals were weighed immediately prior to onset of experiment.

Thank you very much for the comment, and we have now rephrased this sentence into active voice in our revised manuscript (Line 217-218):

“We weighed individuals (accuracy: 0.0001g) immediately prior to the onset of the experiment.”

Line 220: “communal breeding is more likely to take place on carcasses larger than 25 g”

Thank you for the comment, and we have now rephrased this sentence in our revised manuscript (Line 218-219):

“We chose a large mouse carcass (25.0 ± 2.0 g) as the breeding resource, because communal breeding is more likely to take place on carcasses larger than 25 g in *N. vespilloides* [49,64].”

Line 224: 9 out of 10 females produced? Are you saying that 9 out of 10 small females produced offspring in a communal attempt, and only 5 out of 10 males? This makes it seem a bit odd to measure group outcome, especially for males.

Thank you very much for the comment. We tested for the parentage of offspring produced in double-pair breeding groups to ensure that all females could produce in these groups, and we explained that communal breeding could be induced in such a treatment. To measure the reproductive success of groups, we recorded the total

number of offspring produced in groups. We have now clarified this in our revised manuscript (Line 259-263):

“As measures for offspring development and reproductive success of groups, we used the larvae-dispersing time (the time from the start of the experiment until the onset of larvae dispersing) and the brood size (number of larvae) and the average larval weight (total weight of larvae/brood size) of groups at larval dispersal (i.e. larvae dispersed from the carcass), respectively.”

Line 229: Need to edit sentence. E.g., “At the onset of each treatment, beetles were placed in a breeding box with a thawed mouse.”

Thank you for the comment, and we have now edited this sentence in our revised manuscript (Line 225-227):

“At the onset of each treatment, beetles were placed in a breeding box, where a thawed mouse was introduced as breeding resource.”

Line 238: should this “and” be “and/or”? Larva provisioning occurs inside of the carcass rather than on surface?

Thank you for the comment. Yes, in burying beetles, larvae provisioning occurs inside of the carcass. We have now edited this in our revised manuscript (Line 235-237):

“We recorded parental care behaviour when an individual provided indirect parental care (i.e. carcass guarding and maintenance on the surface of the carcass), and/or provided direct parental care (i.e. larvae provisioning inside the carcass) [63,81,82].”

Line 244: “joint parental performance” – what does this mean? As a proxy for cooperation perhaps? State why this is a good proxy.

Thank you for the comment, and we agree with you that the degree of carcass burial is a proxy for the level of cooperation in burying beetles. In these beetles, the rate of carcass burial is positively associated with resource protection, i.e. a faster rate in carcass burial is likely to reduce the probability of being found or usurped by other intruders (e.g. flies). We have now rephrased this in our revised manuscript (Line 243-247):

“As a proxy for the cooperation of breeding groups or breeding pairs in parental care, the degree of carcass burial was estimated according to the fraction of the mouse above the ground and carcass roundness [85]. In burying beetles, the degree of carcass burial over time may be associated with resource protection, i.e. a faster rate in carcass burial is likely to reduce the probability of being found or usurped by other intruders (e.g. flies) [64,67].”

Line 246: Start new paragraph. It may benefit the methods section overall to have subheadings that describe what measures are. E.g., “Communal versus non-communal breeding”, then “Measurement of parental care”, “Reproductive success” – or whichever you think best. Right now, the methods seem to fluctuate between discussion (e.g., line 248 “resulted in selection . . .”) and the actual instructions for carrying out experiment.

Thank you for the comment, and we have now added the subheadings to highlight the experimental protocols and measures sections in the methodology of our revised manuscript (Line 250-251). For example, in experiment 1, we used ‘communal versus

non-communal breeding’ and ‘measurement of parental care and reproductive success’ as two subheadings. We also added a ‘measurement’ section as a new paragraph and rephrased these sentences in our revised manuscript.

“This results in females laying their eggs earlier (which may hatch earlier) and synchronously laying eggs with other female cobreeders [66,72,77].”

Line 250 – 254: Edit sentence – three variations of “reproduction” in same sentence – reproduction, reproductive strategies, and reproductive performance – are they all different? Are they all the same?

Thank you very much for the comment. These three variations of ‘reproduction’ in the same sentence are all the same, and all refer to individual reproductive performance in oviposition for communal and non-communal groups. We have now rephrased this sentence in our revised manuscript (Line 251-256):

“Thus, we examined such reproductive performance in communal and non-communal groups, by recording the timing of reproductive performance and offspring development. As measures for the timing of reproductive performance, we recorded the egg-laying time (the time from the start of the experiment until the onset of egg laying) and the larvae-hatching time (the time from the start of the experiment until the onset of larvae hatching).”

Line 261-263: Why was weight change used as a proxy for parental effort instead of the parental behavior that you monitored? Were both used in conjunction?

Thank you very much for the comment. For burying beetles, an individual’s weight change during the entire breeding period is a mixture of the costs resulting from providing parental behaviour and the benefits from consuming parts of the carcass. So, weight change is used as proxy for parental effort instead of parental behaviour. We have now rephrased this in our revised manuscript (Line 263-268):

“We recorded weight change during breeding ($[\text{final weight} - \text{initial weight}] / \text{initial weight}$) and survival of adult individuals at the end of the experiment, as parameters for parental investment during the entire breeding period and reproductive costs for individuals, respectively [49,57]. For burying beetles, an individual’s weight change during the entire breeding period is a mixture of the costs of providing parental care and the benefits of consuming parts of the carcass [49].”

Line 265: “were not fed prior to subsequent experimentation”.

Thank you for the comment, and we have now rephrased this sentence in our revised manuscript (Line 268-270):

“After the first breeding event, surviving beetles were kept individually for five days in rearing boxes (10×5 cm and 8.5 cm high) and were not fed with any food (i.e. mealworms) prior to subsequent experimentation.”

Line 280-288: This reads more like introduction or discussion. Also, you are probably okay with paternity assurance - individuals were switched around (i.e., all individuals experienced same treatment) so paternity assurance will be equally low across treatments. This could be a discussion point but not appropriate here.

Thank you very much for the comment, and we have now rephrased this clarification in our revised manuscript (Line 286-291):

“For newly-formed breeding pairs in the second breeding event (without other conspecifics), uncertain paternity is equally low across treatments, because individuals from different groups experienced the same treatment and males could prevent such uncertainty through copulation, e.g. when females may have stored sperm from the first breeding event [86-88]. This indicated that, in the second experiment, the uncertainty of paternity was unlikely to have an effect on an individual’s parental care.”

We have now moved this to discussion section in our revised manuscript (Line 516-522):

“When multiple males compete for one carcass, uncertain and reduced paternity may affect an individual’s parental care. For example, when reproductive competition is high, males spent more time providing parental care [86]. We did not find this effect on parental behaviour during the second breeding event in our study. Females may have stored sperm from the first breeding event (i.e. communal groups), with which she could fertilize eggs produced in the second breeding event (i.e. pair breeding). However, males largely prevent this through a high copulation rate and thus dilute stored sperm from the female’s previous mates or fresh sperm from sneaky males [87,88].”

Line 294 and line 317: says “reproductive and developmental timing” on line 294, and then the developmental timing is lumped into reproductive timing for the analyses on line 316.

Thank you very much for the comment. You are correct, and we have now edited this in our revised manuscript (Line 316-319):

“We used linear models (LMs) or generalized linear models (GLMs) with Poisson error structures to analyse developmental and reproductive timing (including egg-laying, larvae-hatching and larvae-dispersing time), and reproductive success (including brood size and averaged larval weight), using breeding group as a fixed factor.”

Line 325-328: Carcass burial was analyzed with time, previous experience, size, sex, and interactions as fixed factors? Given the sample size, seems like too many variables.

Thank you for the comment. Yes, carcass burial was analyzed with time, previous experience and size of females and males and their interactions (nine explanatory variables in the model, which shouldn’t be an issue given the relatively large sample size of 237). We have now added the sample size (n = 237) in our revised manuscript (Line 327-329):

“We analysed the degree of carcass burial (n = 237) using observational time, previous breeding experience (communal versus non-communal) and size (large versus small) of females and males and their interactions as fixed factors, ... ”

Line 329-335: this is not appropriate in the “Statistical analyses” section. Also- you are testing whether there are carry-over effects, rephrase so that it is not implying that you already know there are carry-over effects.

Thank you very much for the comment, and we agree with you that this is not appropriate in the statistical analysis section. We have now rephrased this in our revised manuscript (Line 331-337):

“For burying beetles, the previous parental investment of individuals (e.g. parental investment time and weight change during previous breeding events) may influence their subsequent parental behaviour [54,90]. To further test whether the effects of previous breeding experience (communal versus non-communal breeding) on an individual’s

parental performance are simply by-effects of the previous parental efforts, we also analysed an individual's parenting investment time and weight change at the second breeding event, using GLMs or LMs fitted with binomial or Gaussian error structures."

Line 333-336: Sentence confusing, rephrase.

Thank you very much for the comment, and we have now rephrased this sentence in our revised manuscript (Line 333-337):

"To further test whether the effects of previous breeding experience (communal versus non-communal breeding) on an individual's parental performance are simply by-effects of the previous parental efforts, we also analysed an individual's parenting investment time and weight change at the second breeding event, using GLMs or LMs fitted with binomial or Gaussian error structures."

Line 351: "the access of carcass"

Thank you for the comment, and we have now corrected this in our revised manuscript (Line 350-353):

"In the communal groups, the larger pairs of individuals became dominant and spent significantly more time on the carcass during the entire breeding period than the smaller pairs of individuals, which were subordinates (mean ± SE of large vs. small individuals: 58.96 ± 2.95 vs. 7.22 ± 1.33%; $\chi^2 = 315.65$, $p < 0.001$; Table 1)."

Line 365-368: how can total investment in care be different but parental behavior not be different? Care implies parental behavior. This sentence is also an interpretation – save interpretations for the discussion.

Thank you for the comment. Here, we suggest that breeding in communal groups showed enhanced fitness benefits at the group level, including a better group performance in carcass burial and total parental investment time by groups, whereas individual parental investment time was similar in communal and non-communal groups. We have rephrased this sentence and saved this interpretation for the discussion section (Line 449-452).

"Our results indicated that breeding in communal groups showed enhanced fitness benefits at the group level by improving carcass burial performance and total parental investment time, whereas individual parental investment time was similar in communal and non-communal groups."

Line 381: does egg-laying period = the amount of time to finish laying eggs? Onset of egg-laying? How is this different from the egg laying time (reported in previous sentence)? Or time it took for the eggs to hatch?

Thank you for the comment. In our study, we defined egg-laying period as the period of time from the onset of egg laying until the onset of larvae hatching, and we have now add this definition in our revised manuscript (Line 258-259):

"We also calculated the egg-laying period (the period of time from the onset of egg laying until the onset of larvae hatching)."

Line 407: Need to remove “specifically” – this makes it sound like you are providing specific results to a general one in the previous sentence, when in fact you are providing different results.

Thank you very much for the comment, and we agree with you. We have now removed this in our revised manuscript (Line 404):

Line 407-409: This sentence is confusing – rephrase for clarity.

Thank you very much for the comment, and we have now rephrased this in our revised manuscript (Line 404-411):

“In addition, when their partners originated from non-communal groups, both female and male individuals originating from communal groups spent significantly less time on the carcass than those from non-communal groups (mean \pm SE of females: $44.88 \pm 9.45\%$, $n = 9$ vs. $72.50 \pm 15.85\%$, $n = 5$; of males: $20.41 \pm 8.64\%$, $n = 8$ vs. $43.06 \pm 10.24\%$, $n = 5$; $z = -3.01$, $p = 0.005$; table 1B and S2). However, when their partners originated from communal groups, individuals originating from communal or non-communal groups spent similar times providing care on the carcass (mean \pm SE of females: $73.36 \pm 5.62\%$, $n = 12$ vs. $82.94 \pm 5.90\%$, $n = 8$; of males: $50.65 \pm 11.38\%$, $n = 12$ vs. $42.11 \pm 7.68\%$, $n = 9$; $z = -0.43$, $p = 0.89$; table 1B and S2).”

Line 423: which one is males and which is females. Did these all die during/right after communal breeding, or just at some point in experiment? What are sources of mortality?

Thank you very much for the comment. We have now added to the number of dead females and males in our revised manuscript. We checked the survival of beetles after the second breeding event, and we suggest a high cost of reproduction may lead to mortality. We have now edited this sentence in our revised manuscript (Line 418-420):

“For both females and males, we found that individuals originating from communal groups suffered higher mortality after breeding than those from non-communal groups (19 out of 41 vs. 5 out of 27; table 1B).”

Line 448-452: There was no individual level – you measured reproductive success for the whole group, so you don’t know whether there were short-term effects on parental investment and reproductive success.

Thank you for the comment, and we agree with you that there was no individual level. We have now edited this in our revised manuscript (Line 447-449):

“We found that, at the group level, communal breeding appeared to have no short-term effects on parental investment and reproductive success, however we found carry-over effects on parental investment and reproduction.”

Line 467: “reproduce a majority of offspring”? Rephrase.

Thank you very much for the comment, and we have now rephrased this sentence in our revised manuscript (Line 469-470):

“Dominant individuals monopolizing a carcass could often reproduce and have a large proportion of offspring in communal groups [77].”

Line 469-474: seems to contradict conclusions of lines 455-458.

Thank you for the comment. This conclusion does not contradict previous sentences, because this conclusion and lines 455-458 discuss the benefits and evolution of communal breeding from two different points of view, including mutualistic benefit and mutual tolerance (i.e. a best-of-the-bad-job) hypotheses.

Line 487-489: you are contrasting reproduction and reproduction – first half and second half of sentence seem to be re-writes of each other. Rephrase.

Thank you very much for the comment, and we agree with you. We have now deleted the second half of sentence in our revised manuscript (Line 489-490):
“This may be due to individuals originating from communal breeding incur high reproduction and survival costs [57,62,90].”

Line 492: “deteriorate some opportunistic infections”? Implies the infection gets better because of microbe-rich soil. Rephrase.

Thank you very much for the comment, and we have now rephrased this in our revised manuscript (Line 493-495):
“In this scenario, a microbe-rich soil environment might increase the risk of infections associated with fight wounds, thereby increasing the rate of mortality [62].”

Line 494 and 495 – these two sentences are repeating each other.

Thank you for the comment, and we have now rephrased these two sentences in our revised manuscript (Line 495-497):
“For each individual, dominance status is closely associated with fecundity and nutritional state [35,63]. For example, in *N. vespilloides*, limited access to the carcass by subordinate females leads to low fecundity caused by nutritional deficiencies [63].”

Line 497-500: unclear and no literature cited for immunity?

Thank you for the comment, and we have added the references about immunity in our revised manuscript.

Line 509: “save more resources” – did they? You have measurements of weight change, state that here if you found this.

Thank you for the comment. In our study, we found that dominant males in communal breeding spent more time on the carcass than large males in non-communal breeding, whereas weight change between dominant males in communal and non-communal breeding did not differ. Thus, we suggest that dominant males may save more resources or gain more experience by spending more time on the carcass, despite the lack of a no significant difference in weight change.

Line 523: This sentence contradicts previous paragraph.

Thank you very much for the comment. This sentence does not contradict with the previous paragraph, because we suggest that the effects of an individual’s previous

breeding experience on its subsequent behaviour are not simply mediated by energetic benefits (i.e. weight gains). However, we could not excluded that males from communal breeding may save some types of parental resources towards future reproduction.

Line 537-539: wait so we already know about carry-over effects in communal breeders? Describe in intro.

Thank you for the comment, and we agree with you. We have now rephrased and added this to the introduction and discussion in our revised manuscript (Line 98-99 and Line 533-536):

“Moreover, previous breeding experience and changes in physiological states may have carry-over effects on fitness and reproductive performance for individuals [45,46].”

“This might mean that the carry-over effects of communal breeding on individual parenting behaviour and fitness is not simply determined by the parental investment to previous reproduction, but instead is associated with the individual’s previous experience [57,96].”

Line 589: edit

Thank you for the comment, and we have now edited this in our revised manuscript (Line 566-568):

“In burying beetles, although parents often cooperate to provide care towards their offspring, females always share the majority of parental care [107,108].”

Line 594: within carcasses?

Thank you for the comment, and we have now corrected this in our revised manuscript (Line 603-604):

“Males could likely gain compensatory benefits by spending more time on a carcass or consuming more of a carcass [86,116].”

Line 605: This sentence is confusing

Thank you for the comment, and we have now rephrased this in our revised manuscript (Line 614-618):

“Hence, we suggest that each individual adjusts its allocation of parental care between reproductive events in order to maximize its fitness benefits over its lifetime. Such an allocation of parental care is affected by the interplay of sexual conflict and intraspecific competition over resources and reproduction during communal breeding [49,77,82,97], for the following reasons.”

Line 614: I thought that they didn’t “save resources”, according to your results?

Thank you for the comment. In our study, we suggest that males may save more resources to allocate them to enhance future reproduction benefits, or gain a rich experience for future (e.g. fighting and parenting experience). However, ‘resources’ for individuals refer to not only to weight change but also to other types of resources, such as high immunity, improved physiological conditions and skilled prior experience.

Line 630-634: Both of the future directions seem to be the same project...?

Thank you for the comment. We agree with you that the future directions could be investigated in the same research project, and we have now revised this in our revised manuscript (Line 641-645):

“Future research on the evolution of social behaviour should investigate the importance of the carry-over effects of group breeding, and sex differences in these carry-over effects. Further study should investigate whether and how such carry-over effects on fitness select for variation in parental behaviour and reproductive strategy.”

===PREPARING YOUR MANUSCRIPT===

- one version identifying all the changes that have been made (for instance, in coloured highlight, in bold text, or tracked changes);
- a 'clean' version of the new manuscript that incorporates the changes made, but does not highlight them. This version will be used for typesetting if your manuscript is accepted.

If you have been asked to revise the written English in your submission as a condition of publication, you must do so, and you are expected to provide evidence that you have received language editing support. The journal would prefer that you use a professional language editing service and provide a certificate of editing, but a signed letter from a colleague who is a fluent speaker of English is acceptable. Note the journal has arranged a number of discounts for authors using professional language editing services (<https://royalsociety.org/journals/authors/benefits/language-editing/>).

===PREPARING YOUR REVISION IN SCHOLARONE===

-- Ensure that your data access statement meets the requirements

at <https://royalsociety.org/journals/authors/author-guidelines/#data>. You should ensure that you cite the dataset in your reference list. If you have deposited data etc in the Dryad repository, please include both the 'For publication' link and 'For review' link at this stage.

-- If you have uploaded ESM files, please ensure you follow the guidance

at <https://royalsociety.org/journals/authors/author-guidelines/#supplementary-material> to include a suitable title and informative caption. An example of appropriate titling and captioning may be found at

https://figshare.com/articles/Table_S2_from_Is_there_a_trade-off_between_peak_performance_and_performance_breadth_across_temperatures_for_aerobic_scope_in_teleost_fishes_/3843624.

At Step 7 'Review & submit', you must view the PDF proof of the manuscript before you will

be able to submit the revision. Note: if any parts of the electronic submission form have not been completed, these will be noted by red message boxes.

Appendix D

January 13, 2022

Dear Editors, Dr. Polly Campbell and Dr. Kevin Padian,

We would like to thank you and the reviewer for the constructive reviews of our manuscript entitled “*Sex-specific influence of communal breeding experience on parental performance and fitness in a burying beetle*”, and for the opportunity to resubmit a minor revised version. We have carefully read all comments, replied to each of them below and revised our manuscript accordingly. Also, we have amended the email addresses from all authors that are able to receive correspondence. We very much hope that our amendments, in the light of your comments, have improved the manuscript. We sincerely hope that you will find the current version acceptable for publication in *Royal Society Open Science*. Thank you very much for reconsidering our manuscript.

Yours sincerely,
Long Ma, on behalf of all authors

Dear Dr Ma

On behalf of the Editors, we are pleased to inform you that your Manuscript RSOS-211179.R2 "Sex-specific influence of communal breeding experience on parenting performance and fitness in a burying beetle" has been accepted for publication in Royal Society Open Science subject to minor revision in accordance with the referees' reports. Please find the referees' comments along with any feedback from the Editors below my signature.

Please also note that we require an active email address from all authors that is able to receive messages from the journal. At present 'm.hammers@rug.nl' is unable to receive correspondence - please can you ensure your colleague's email address is amended or communicated to the editorial office with your revised manuscript?

Please submit your revised manuscript and required files (see below) no later than 7 days from today's (ie 11-Jan-2022) date. Note: the ScholarOne system will 'lock' if submission of the revision is attempted 7 or more days after the deadline. If you do not think you will be able to meet this deadline please contact the editorial office immediately.

Thank you for submitting your manuscript to Royal Society Open Science and we look forward to receiving your revision. If you have any questions at all, please do not hesitate to

get in touch.

on behalf of Dr Polly Campbell (Associate Editor) and Kevin Padian (Subject Editor)
openscience@royalsociety.org

Associate Editor Comments to Author (Dr Polly Campbell):

Associate Editor

Comments to the Author:

The authors have, again, done a commendably thorough and thoughtful job on this second revision. Both readability and accurate use of terminology are greatly improved. Please resolve the following minor issues:

We very much appreciate your positive and constructive comments. We have addressed all these issues in our revised manuscript below.

L39 Please change that to and

Thank you for the comment, and we have corrected it in our revised manuscript.

L98-100 Please include common name of animal in cited study: "For example, in _____, experienced individuals" etc.

Thank you very much for the comment, and we have added the common name of animal cited in our revised manuscript (Line 98-99):

*"For example, in the northern goshawk (*Accipiter gentilis*), experienced individuals with previous breeding provided more parental care towards the current brood compared to inexperienced individuals [45]."*

L148-150 "It is also important to examine whether and how carry-over effects, concurrent with short-term effects, act on the selection for group breeding and its adaptation to different environments." Please revise for clarity. As written, it sounds like these effects act on selection that favors group breeding. This doesn't make any sense and it's unclear what adaptation to different environments refers to.

Thank you very much for the comment, and we have now rephrases this sentence in our revised manuscript (Line 148-150):

"It is also important to examine whether and how carry-over effects, concurrent with short-term effects, shape the evolution of group breeding and its adaptation to adverse and rapidly changing environments."

L159-160 "We predict that communal groups achieve higher reproductive success compared to non-communal groups." I know the authors have done a lot to modify their terminology in relation to fitness and that reproductive success was a reviewer suggestion. But reproductive success IS fitness and it doesn't make sense to talk about it as a single

metric for a group of unrelated individuals. It would be fine to predict that brood size will be larger in communal relative to non-communal groups.

Thank you very much for the comment, and we agree with you and have now rephrased this sentence in our revised manuscript (Line 159-160):

“We predict that communal groups produce larger brood sizes compared to non-communal groups.”

L250-255 The use of performance is a bit confusing. Suggest shortening to: “Therefore, we examined the timing of egg-laying and larval-hatching in communal and non-communal groups. Egg-laying and larval hatching times were recorded as the time from the start of the experiment until the onset of laying and hatching, respectively.”

Thank you very much for the comment, and we have rephrased this sentence according to your comment in our revised manuscript (Line 250-253):

“Therefore, we examined the timing of egg-laying and larvae-hatching in communal and non-communal groups. Egg-laying and larvae-hatching time was recorded as the time from the start of the experiment until the onset of egg laying and larvae hatching, respectively.”

L259 Please change success to output.

Changed.

L285-288 This is a confusing sentence. I think the intended meaning is that all males in the second experiment experience some paternity uncertainty because females mated in the first experiment but this shouldn't affect inferences about carry-over effects from different prior breeding experiences because 1) mating occurred in both pair and communal breeding groups, and 2) males dilute competitors' sperm by copulating a lot. Please try to get this across more clearly or remove from Methods as suggested by the reviewer and combine with the added text in Discussion (L516). Note that ALL females will have mated in the first experiment, not just the ones in the communal treatment as suggested on L519.

Thank you very much for the comment. Yes, the sentence means that all males in the second experiment may experience some paternity uncertainty because females mated in the first experiment. However, such paternity uncertainty does not affect inferences about carry-over effects from different previous breeding experiences in our study. We have clarified this in Methods (Line 283-286), and we have removed the associated explanation from Methods and have rephrased this in Discussion of our revised manuscript (Line 510-517). Also, we have noted that all females have mated in the first experiment (including communal and non-communal groups) in our revised manuscript (Line 514-517).

“For newly-formed breeding pairs in the second breeding event (without other conspecifics), all males may experience some paternity uncertainty because females mated in the first experiment. However, in the second experiment, the uncertainty of paternity was unlikely to have an effect on an individual's parental care [86-88].” (Line 283-286)

“When multiple males compete for one carcass, uncertain and reduced paternity may affect an individual's parental care. For example, when reproductive competition is high, burying beetle males spent more time providing parental care [86]. However, such paternity uncertainty does not affect inferences about carry-over effects from different previous breeding experiences in our study. Females have mated and may have stored sperm from

the first breeding event (i.e. communal and non-communal groups), with which she could fertilize eggs produced in the second breeding event (i.e. pair breeding). However, males largely prevent this and thus dilute the sperm of earlier competitors by copulating repeatedly with the same female [86-88].” (Line 510-517)

L332-334 “To further test whether the effects of previous breeding experience (communal versus non-communal breeding) on an individual’s parental performance are simply by-effects of the previous parental efforts” I think the authors over-interpreted what the reviewer was asking them to do. Please replace with the text from the prior version (“To further test whether the carry-over effects of communal and non-communal breeding on an individual’s parental performance in the future breeding events are manifested through the direct effect of the previous parental efforts”) and add potential before carry-over.

Thank you very much for the comment, and we have replaced with the text from the prior version in our revised manuscript (Line 328-330):

“To further test whether the potential carry-over effects of communal and non-communal breeding on an individual’s parental performance in the future breeding events are manifested through the direct effect of the previous parental investment, ...”

L372-373 Please just report that there was no effect of breeding treatment on brood size (and any other variable that reproductive success refers to).

Thank you very much for the comment, and we have rephrased this in our revised manuscript (Line 368-369):

“Breeding group (communal versus non-communal) had no effect on brood size, but had a significant effect on average larval weight (Table S1).”

L448-451 For the reasons mentioned above, it doesn’t make sense to talk about fitness/reproductive success at the group level and here, without knowing individual reproductive output for communal breeders in the first experiment it doesn’t make sense to talk about that either. Please modify as follows: “We found that communal breeding appeared to have no short-term effects of parental investment. However, we found...” etc. “Our results indicated that communally breeding groups had improved carcass burial performance and increased total time spent on parental behavior, whereas...” etc.

Thank you very much for the comment, and we agree with you that it does not make sense to talk about fitness at the group level because in the study we only test the fitness and reproductive success of communal and non-communal groups. We have modified this sentence in our revised manuscript (Line 442-446):

“We found that communal breeding appeared to have no short-term effects on parental investment and reproductive success. However, we found carry-over effects on parental investment and reproduction. Our results indicated that communally breeding in groups had improved carcass burial performance and increased total time spent on parental investment, whereas individual parental investment time was similar in communal and non-communal groups.”

L488-489 Grammar. Suggest changing incur high to incurring higher.

Thank you very much for the comment, and we have rephrased this sentence in our revised manuscript (Line 483-484):

“This may be due to individuals originating from communal breeding incurring higher reproduction and survival costs [57,62,90].”

Reviewer comments to Author:

===PREPARING YOUR MANUSCRIPT===

- one version should clearly identify all the changes that have been made (for instance, in coloured highlight, in bold text, or tracked changes);
- a 'clean' version of the new manuscript that incorporates the changes made, but does not highlight them. This version will be used for typesetting.

===PREPARING YOUR REVISION IN SCHOLARONE===

Attach your point-by-point response to referees and Editors at the 'View and respond to decision letter' step. This document should be uploaded in an editable file type (.doc or .docx are preferred). This is essential, and your manuscript will be returned to you if you

do not provide it.

- Your revised manuscript in editable file format (.doc, .docx, or .tex preferred). You should upload two versions:

- An individual file of each figure (EPS or print-quality PDF preferred [either format should be produced directly from original creation package], or original software format).

- An editable file of each table (.doc, .docx, .xls, .xlsx, or .csv).

- An editable file of all figure and table captions.

- Any electronic supplementary material (ESM).

- If you are requesting a discretionary waiver for the article processing charge, the waiver form must be included at this step.

- If you are providing image files for potential cover images, please upload these at this step, and inform the editorial office you have done so. You must hold the copyright to any image provided.

- A copy of your point-by-point response to referees and Editors. This will expedite the preparation of your proof.

- Ensure that your data access statement meets the requirements

at <https://royalsociety.org/journals/authors/author-guidelines/#data>. You should ensure that you cite the dataset in your reference list. If you have deposited data etc in the Dryad repository, please only include the 'For publication' link at this stage. You should remove the 'For review' link.

- If you are requesting an article processing charge waiver, you must select the relevant waiver option (if requesting a discretionary waiver, the form should have been uploaded, see 'File upload' above).

- If you have uploaded any electronic supplementary (ESM) files, please ensure you follow the guidance at <https://royalsociety.org/journals/authors/author-guidelines/#supplementary-material> to include a suitable title and informative caption. An example of appropriate titling and captioning may be found

at https://figshare.com/articles/Table_S2_from_Is_there_a_trade-off_between_peak_performance_and_performance_breadth_across_temperatures_for_aerobic_scope_in_teleost_fishes_/3843624.

At the 'Review & submit' step, you must view the PDF proof of the manuscript before you will be able to submit the revision. Note: if any parts of the electronic submission form have

not been completed, these will be noted by red message boxes - you will need to resolve these errors before you can submit the revision.